# DYNAMIC-LLaVA:
# EFFICIENT MULTIMODAL LARGE LANGUAGE MODELS VIA DYNAMIC VISION-LANGUAGE CONTEXT SPARSIFICATION

**Wenxuan Huang**[1][*]   **Zijie Zhai**[1][*]   **Yunhang Shen**[2]   **Shaosheng Cao**[3][✉]   **Fei Zhao**[4]
**Xiangfeng Xu**[1]   **Zheyu Ye**[3]   **Shaohui Lin**[1,5][✉]

[1]East China Normal University   [2]Xiamen University   [3]Xiaohongshu Inc.   [4]Nanjing University
[5]Key Laboratory of Advanced Theory and Application in Statistics and Data Science, MOE, China
`osilly0616@gmail.com, shaohuilin007@gmail.com`

## ABSTRACT

Multimodal Large Language Models (MLLMs) have achieved remarkable success in vision understanding, reasoning, and interaction. However, the inference computation and memory increase progressively with the generation of output tokens during decoding, directly affecting the efficacy of MLLMs. Existing methods attempt to reduce the vision context redundancy to achieve efficient MLLMs. Unfortunately, the efficiency benefits of the vision context reduction in the prefill stage gradually diminish during the decoding stage. To address this problem, we proposed a dynamic vision-language context sparsification framework **Dynamic-LLaVA**, which dynamically reduces the redundancy of vision context in the prefill stage and decreases the memory and computation overhead of the generated language context during decoding. Dynamic-LLaVA designs a tailored sparsification inference scheme for different inference modes, *i.e.*, prefill, decoding with and without KV cache, to achieve efficient inference of MLLMs. In practice, Dynamic-LLaVA can reduce computation consumption by ~**75%** in the prefill stage. Meanwhile, throughout the entire generation process of MLLMs, Dynamic-LLaVA reduces the ~**50%** computation consumption under decoding without KV cache, while saving ~**50%** GPU memory overhead when decoding with KV cache, due to the vision-language context sparsification. Extensive experiments also demonstrate that Dynamic-LLaVA achieves efficient inference for MLLMs with negligible understanding and generation ability degradation or even performance gains compared to the full-context inference baselines. Code is available at `https://github.com/Osilly/dynamic_llava`.

## 1 INTRODUCTION

Large Language Models (LLMs) have achieved outstanding performance and made a significant impact in real-world applications (Zheng et al., 2023; Team, 2023; Touvron et al., 2023a;b; Achiam et al., 2023; Jiang et al., 2024). In particular, within the vision-language multimodal fields, the LLaVA paradigm (Liu et al., 2024b;a; Bai et al., 2023; Zhu et al., 2023; Zhao et al., 2024) has emerged as the mainstream approach of Multimodal Large Language Models (MLLMs). This paradigm involves mapping visual data, through feature extractors and projecting embeddings, into the same feature distribution of LLMs for processing. It has shown notable success in enabling general capabilities in vision understanding, reasoning, and interaction.

However, LLMs often adopt the decoder-only Transformer as the base architecture and typically contain an extremely large number of parameters. As output tokens are generated during decoding, the computational consumption progressively increases, leading to a substantial computational burden (Vaswani, 2017). To alleviate this issue, modern LLMs frequently employ the KV cache

---

[*]Equal contribution.   [✉]Corresponding authors.

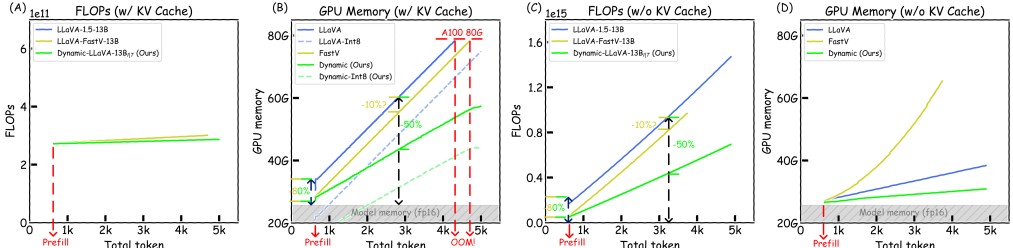

Figure 1: The entire generation process of MLLMs. As generation progresses, the primary resource overheads of MLLMs under decoding with and without KV cache modes are GPU memory overhead and computation consumption, respectively. Previous vision context sparsification methods achieved initial inference efficiency through vision context sparsification. However, these benefits gradually diminish as decoding continues. The results are measured in one A100 (80G) and the batch size is fixed to 8. "OOM" means the generation process has failed due to the out of GPU memory.

technique during decoding, which accelerates inference speed by storing previously computed KV activations and reusing them in subsequent decoding steps. It helps reduce redundant computations but introduces significant GPU memory overhead to store activated intermediate variables (Luohe et al., 2024; Radford et al., 2019). Moreover, the GPU memory overhead gradually increases in the decoding progresses. This problem becomes even more pronounced in MLLMs within the LLaVA paradigm, which is often combined with LLM, requiring significant resources for inference.

To achieve efficient MLLMs, recent works attempted to reduce the number of image tokens processed by LLMs, which decreases computation and memory overhead while maintaining model performance (Jin et al., 2024). One category of these methods (Chen et al., 2024a; Shang et al., 2024; Ye et al., 2024; Arif et al., 2024; Song et al., 2024) selects a subset of image tokens feeding to LLMs, *e.g.*, FastV (Chen et al., 2024a) uses the full-context attention matrix to determine which image tokens feed to LLMs. We refer to this approach as vision context sparsification. Another category (Li et al., 2024a; Kar et al., 2024; Li et al., 2024b) modifies the vision feature extractors or projectors to generate few number of image tokens. We refer to this strategy as efficient vision encoders/projectors. However, reducing image tokens primarily accelerates the prefill stage. We argue that ***the efficiency benefits of the image token reduction in the prefill stage gradually diminish during the decoding stage***. As shown in Fig. 1, the previous state-of-the-art (SoTA) methods with image token reduction, *e.g.*, FastV (Chen et al., 2024a), significantly reduce the number of image tokens during the prefill stage, leading to substantial resource savings. Unfortunately, as decoding progresses, the actual benefits of this reduction diminish. This is because, in the decoding stage, the computation cost increasingly shifts towards the autoregressive generation of the output text tokens, where the reduced image tokens have less impact on the overall efficiency. As a result, the initial speedup from fewer image tokens becomes less dominant as decoding progresses. This phenomenon is discussed in detail in Sec. 3.2.

To address the above problem, we propose a novel dynamic vision-language context sparsification framework, named ***Dynamic-LLaVA***. Specifically, we introduce two learnable predictors to sparsify the vision and language contexts in both prefill and decoding of MLLM, respectively (Sec. 3.3.1). The tailored sparsification inference scheme is designed for different inference modes within MLLMs, *i.e.*, prefill, decoding without KV cache and with KV cache (Sec. 3.3.2). During the training process, the proposed Dynamic-LLaVA employees masked softmax to isolate the influence of non-essential tokens on the important tokens and the Gumbel-Softmax (Jang et al., 2016) with Straight-Through Gradient Estimator (Bengio et al., 2013) to avoid the gradient flow problem, which enables end-to-end optimization (Sec. 3.3.3). Last but not least, we further develop the batch-parallel strategy for the sparsification inference of the Dynamic-LLaVA framework, which fully leverages GPU hardware advantage under batch parallel inference conditions (Sec. A.1). Our framework can be easily integrated into MLLMs (*e.g.*, LLaVA (Liu et al., 2024b;a)) and efficient vision encoder/projector methods for MLLMs (*e.g.*, TokenPacker (Li et al., 2024a)) to achieve more efficient inference.

Extensive experiments conducted on the vision understanding benchmarks and the generation ability evaluations demonstrate that the proposed Dynamic-LLaVA framework achieves efficient inference throughout the entire MLLM generation process with negligible performance degradation.In practice, the computation consumption of the prefill stage is reduced by ∼**75%** via the vision context

sparsification. Meanwhile, during the entire MLLM generation process, through the comprehensive vision-language context sparsification, the proposed Dynamic-LLaVA reduces computation cost by $\sim$**50%** when decoding without KV cache, while saving $\sim$**50%** GPU memory overhead with KV cache. To the best of our knowledge for efficient MLLMs, Dynamic-LLaVA is the **first** framework which attempts simultaneous sparsification of both vision and language contexts. We believe that our work can inspire new insights for the research community.

## 2 RELATED WORK

### 2.1 TOKEN REDUCTION FOR EFFICIENT MLLMS

The existing vision context sparsification (Chen et al., 2024a; Shang et al., 2024; Ye et al., 2024; Arif et al., 2024; Song et al., 2024) and efficient vision encoder/projector (Li et al., 2024a; Kar et al., 2024; Li et al., 2024b; Bai et al., 2023; Cha et al., 2024; Chen et al., 2024b; Chu et al., 2024) methods have attempted to reduce the image token to accelerate the prefill stage. However, with the decoding of MLLMs, the efficiency benefits gradually diminish during the decoding. Our proposed Dynamic-LLaVA framework achieves consistently efficient inference throughout the entire generation phase via the vision-language context sparsification. The detailed discussions are presented in Appendix A.3.1.

### 2.2 KV CACHE COMPRESSION FOR EFFICIENT LLMS

Some previous works (Zhang et al., 2024b; 2023; Li et al., 2024c; Xiao et al., 2023; Ge et al., 2023; Han et al., 2023; Yang et al., 2024; Zhang et al., 2024a) compress KV cache in order to reduce the GPU memory overhead of LLM inference during decoding, which has a similar goal to our Dynamic-LLaVA when considering decoding with KV cache. However, these methods typically necessitate using the subset of past generated KV cache to facilitate the selection of critical activations for participation during inference. In other words, it is imperative to obtaion the generated KV cache to make the decision for which activations should be removed during LLM inference. Dynamic-LLaVA, when decoding with KV cache, can be regarded as ***"online KV cache compression"***, *i.e.*, it utilizes the features of the current output text token to determine whether to retain the generated activations without relying on past KV cache. This capability is crucial, as it enables Dynamic-LLaVA to enhance efficiency during the stages of prefill and decoding without KV cache, which do not involve KV cache utilization. Moreover, the additional efficiency gained in prefill and decoding without KV cache modes is of comparable importance for MLLMs (Liu et al., 2024b; Li et al., 2024d; Chen et al., 2024a; Huang et al., 2024a; Cha et al., 2024; Leviathan et al., 2023; Chen et al., 2023a; Liu et al., 2023a). A detailed discussion of the proposed Dynamic-LLaVA framework and traditional LLM KV cache compression methods is provided in Appendix A.3.2.

## 3 METHOD

### 3.1 PRELIMINARIES AND NOTATIONS

Multimodal large language model (MLLM), *e.g.*, LLaVA (Liu et al., 2024b), continues the autoregressive model paradigm (Radford et al., 2019; Liu et al., 2024a) and typically includes prefill and decoding stages during inference. In the prefill stage, features from different modalities are mapped into the same feature distribution as the input embeddings of large language model (LLM). These multimodal features, in conjunction with text tokens, are simultaneously processed by LLM to generate the initial output text token. During the decoding stage, the tokens in the prefill stage, together with all subsequently generated output text tokens, are used in an autoregressive manner to predict the next output text token. In the context of LLM with $L$ Transformer decoder layers, we denote distinct index sets for various types of tokens processed across the stages of the model. Specifically, the index set for image tokens during prefill is denoted as $\mathcal{I}^I = \{1, 2, \cdots, N_l^I\}$, for text tokens during prefill as $\mathcal{I}^T = \{1, 2, \cdots, N_l^T\}$, while for output text tokens during decoding as $\mathcal{I}^{OT} = \{1, 2, \cdots, N_l^{OT}\}$, where $N_l^I$, $N_l^T$ and $N_l^{OT}$ are the counts of image tokens, text tokens and output text tokens of $l$-th decoder, respectively. The sets of image tokens, text tokens, and output text tokens processed by the $l$-th LLM decoder layer are denoted as $\mathcal{S}_l^I = \{t_{l,i}^I | \forall i \in \mathcal{I}^I\}$, $\mathcal{S}_l^T = \{t_{l,i}^T | \forall i \in \mathcal{I}^T\}$ and $\mathcal{S}_l^{OT} = \{t_{l,i}^{OT} | \forall i \in \mathcal{I}^{OT}\}$, respectively, where $t_{l,i}^I, t_{l,i}^T, t_{l,i}^{OT} \in \mathbb{R}^d$ denotes the $i$-th token of the

different token sets for the $l$-th decoder. The sets of all tokens and the corresponding size are defined as $\mathcal{S}_l = \mathcal{S}_l^I \cup \mathcal{S}_l^T \cup \mathcal{S}_l^{OT}$ and $N_l = N_l^I + N_l^T + N_l^{OT}$, respectively.

Considering the computation of the standard LLM (Touvron et al., 2023a), the prefill stage of the $l$-th decoder is simply described as follows:

$$\mathcal{S}_{l+1}^P = \mathrm{FFN}(\mathrm{MHA}(\mathcal{S}_l^P, \mathcal{S}_l^P, \mathcal{S}_l^P)), \tag{1}$$

where $\mathcal{S}_l^P = \mathcal{S}_l^I \cup \mathcal{S}_l^T$. $\mathrm{MHA}(\cdot, \cdot, \cdot)$ and $\mathrm{FFN}(\cdot)$ denote the Multi-Head Attention Block and the Feed-Forward Networks in $l$-th LLM decoder layer (Vaswani, 2017).

During the decoding stage, there are typically two methods available, *i.e.*, decoding without and with KV cache. First, we consider decoding without KV cache as an extension of the prefill stage. The one-pass decoding is computed as follows:

$$\mathcal{S}_{l+1}^P \cup \mathcal{S}_{l+1}^{OT} = \mathrm{FFN}(\mathrm{MHA}(\mathcal{S}_l^P \cup \mathcal{S}_l^{OT}, \mathcal{S}_l^P \cup \mathcal{S}_l^{OT}, \mathcal{S}_l^P \cup \mathcal{S}_l^{OT})). \tag{2}$$

For the decoding with KV cache, we further split the MHA operation as follows:

$$
\begin{aligned}
Q, K, V &= \{W_l^Q \mathcal{S}_{l,N_l^{OT}}^{OT}\}, \{W_l^K \mathcal{S}_{l,N_l^{OT}}^{OT}\}, \{W_l^V \mathcal{S}_{l,N_l^{OT}}^{OT}\}, \\
\mathcal{S}_l^K &= \mathcal{S}_l^K \cup K, \ \mathcal{S}_l^V = \mathcal{S}_l^V \cup V, \\
O &= W_O \, \mathrm{Attention}(Q, \mathcal{S}_l^K, \mathcal{S}_l^V), \\
\mathcal{S}_{l+1,N_{l+1}^{OT}}^{OT} &= \mathrm{FFN}(O),
\end{aligned}
\tag{3}
$$

where $W_l^Q$, $W_l^K$, $W_l^V$ and $W_l^O$ are the linear layers to obtain the activation sets $Q$, $K$, $V$ and $O$ in $l$-th LLM decoder layer. $\mathrm{Attention}(\cdot, \cdot, \cdot)$ is the Scaled Dot-Product Attention operation (Vaswani, 2017). And $\mathcal{S}_{l,N_l^{OT}}^{OT}$ is the last output text token ($N_l^{OT}$-th output text token) in the $l$-th decoder layer during decoding, while $\mathcal{S}_l^K$ and $\mathcal{S}_l^V$ are KV cache of the $l$-th decoder layer.

The primary computation cost of the prefill and decoding without KV cache operations involve processing various sets of tokens and most overhead is in the activation of linear layers. Specifically, the cost of $l$-th deocder layer during the prefill stage is dictated by the combined size of the image and text token sets, denoted as $\mathrm{Computation}(\mathrm{Prefill})_l \propto |\mathcal{S}_l^I \cup \mathcal{S}_l^T|$, where $|\cdot|$ denotes the number of tokens. For the decoding stage without KV cache, the computation overhead is determined by the sum of the aforementioned components plus the size of the generated output text tokens, represented as $\mathrm{Computation}(\mathrm{Decoding}_{\mathrm{w/o\,cache}})_l \propto |\mathcal{S}_l^I \cup \mathcal{S}_l^T \cup \mathcal{S}_l^{OT}|$. Unlike the previous two methods, decoding with KV cache operation requires only the last output text token to be activated. This decoding method yields lower computation overhead compared to decoding without KV cache. However, it needs additional GPU memory to store the the activated intermediate variables, thereby introducing a higher GPU memory cost (Luohe et al., 2024). The GPU memory overhead of $l$-th KV cache is also determined by the quantity of the token sets, defined as $\mathrm{Memory}(\mathrm{Decoding}_{\mathrm{w/\,cache}})_l \propto |\mathcal{S}_l^I \cup \mathcal{S}_l^T \cup \mathcal{S}_l^{OT}|$. Given a $l$-th LLM decoder layer, reducing the sizes $|\mathcal{S}_l^I|$, $|\mathcal{S}_l^T|$ and $|\mathcal{S}_l^{OT}|$ results in a corresponding reduction in the number of tokens for layers beyond the $l$-th decoder layer (Huang et al., 2024b). This reduction decreases the computation consumption and GPU memory overhead during the prefill and decoding stage of LLM.

## 3.2 MOTIVATION

As mentioned above, the token number affects the computation consumption and GPU memory overhead, while the previous works (Chen et al., 2024a; Shang et al., 2024; Li et al., 2024a; Ye et al., 2024; Arif et al., 2024; Song et al., 2024) focus on reducing the image token set $\mathcal{S}_l^I$. However, the reduction in the number of image tokens often primarily accelerates the prefill stage and gradually diminishes in benefit during the decoding stage.

$$
\begin{aligned}
&\mathrm{Computation}(\mathrm{Prefill})_l \propto |\mathcal{S}_l^I \cup \mathcal{S}_l^T|, \\
&\mathrm{Computation}(\mathrm{Decoding}_{\mathrm{w/o\,cache}})_l \propto |\mathcal{S}_l^I \cup \mathcal{S}_l^T \cup \mathcal{S}_l^{OT}| \approx |\mathcal{S}_l^{OT}|, \\
&\mathrm{Memory}(\mathrm{Decoding}_{\mathrm{w/\,cache}})_l \propto |\mathcal{S}_l^I \cup \mathcal{S}_l^T \cup \mathcal{S}_l^{OT}| \approx |\mathcal{S}_l^{OT}|, \\
&\mathrm{where}, \quad |\mathcal{S}_l^{OT}| \to \infty.
\end{aligned}
\tag{4}
$$

As shown in Eq. 4 and Fig. 1, with the generation of output text tokens during the decoding stage, the computation consumption and GPU memory overhead increase progressively. Meanwhile, the prefill

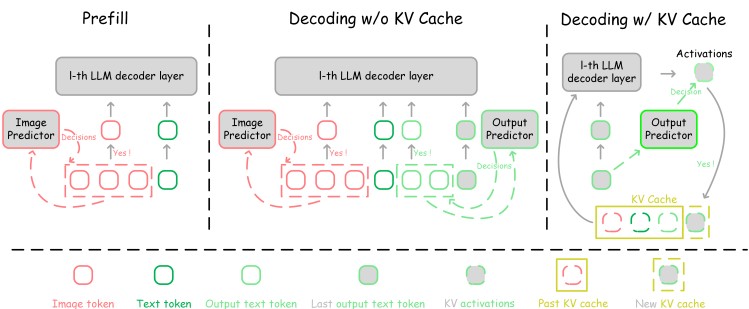

Figure 2: The sparsification inference modes for MLLMs. In the prefill stage, only image tokens are dropped based on the decisions of the learnable image predictor. For decoding without KV cache, we reduce both the image tokens and output text tokens to maintain consistent inference efficiency. When decoding with KV cache, the output predictor determines whether the activations generated by the current output text token should be added to KV cache and thereby discard part of the activations to reduce the size of KV cache. Note that the decision regarding the activations of the current output text token will be shared across all subsequent layers beyond the $l$-th layer. Meanwhile, the "Yes" branch means the decision to keep the token or its activations to participate in subsequent calculations.

stage typically computes only once during the generation process of LLM. Therefore, the benefits gained from reducing the number of image tokens diminish over the course of the entire generation process, as their impact becomes less significant during the later stages of decoding. Inspired by this, we propose the ***Dynamic-LLaVA*** framework to sparsify both vision context $\mathcal{S}_l^I$ and language context $\mathcal{S}_l^{OT}$ involved in LLM generation to achieve the consistently **Efficient MLLM**.

### 3.3 DYNAMIC VISON-LANGUAGE CONTEXT SPARSIFICATION

#### 3.3.1 OVERVIEW

To compress the image token set and output token set, we use two binary masks $\mathcal{M}^I$ and $\mathcal{M}^{OT}$ to determine whether their tokens should be retained or discarded in $l$-th decoder layer, where $\mathcal{M}^I = \{m_i | m_i \in \{0, 1\} \wedge \forall i \in \mathcal{I}^I\}$ and $\mathcal{M}^{OT} = \{m_i | m_i \in \{0, 1\} \wedge \forall i \in \mathcal{I}^{OT}\}$. The reduced image token set and output text token set are defined as $\mathcal{S}_l^{I*} = \{\mathcal{S}_{l,i} | \mathcal{M}_i^I = 1 \wedge \forall i \in \mathcal{I}^I\}$ and $\mathcal{S}_l^{OT*} = \{\mathcal{S}_{l,i} | \mathcal{M}_i^{OT} = 1 \wedge \forall i \in \mathcal{I}^{OT}\}$, respectively, where $|\mathcal{S}_l^{I*}| \leq |\mathcal{S}_l^I|$ and $|\mathcal{S}_l^{OT*}| \leq |\mathcal{S}_l^{OT}|$. In this way, the computation consumption and GPU memory overhead in both the prefill and decoding stages, as described in Eq. 4, are simultaneously reduced. Such optimization contributes to efficient resource utilization during the entire generation process of LLM. In particular, Dynamic-LLaVA uses two learnable predictors $P^I$ and $P^{OT}$ to generate the binary masks $\mathcal{M}^I$ and $\mathcal{M}^{OT}$, respectively. The learnable predictors are applied only once, after the $l$-th decoder layer of the MLLM. Specifically, once the predictor makes the decisions on tokens, this decisions are shared across all subsequent layers. This one-time token reduction strategy circumvents the need for complex adjustments to sparsity ratios of MLLM decoders. In practice, we follow the work (Chen et al., 2024a) and set $l$ to 2. The ablation study for $l$ is presented in Tab. 6. Note that the learnable predictors consist of lightweight, multi-layer neural networks that introduce only marginal additional computation costs (less than 1% of the total). The detailed real GPU latency and memory of the Dynamic-LLaVA framework are shown in Tab. 4 and Tab. 14. Detailed architectures are shown in Appendix A.4.1.

In the following sections, we introduce how to perform the sparsification inference in the prefill and decoding stages of MLLMs in Sec. 3.3.2. We also introduce the end-to-end training for the learnable predictors in Sec. 3.3.3.

#### 3.3.2 SPARSIFICATION INFERENCE

As shown in Fig. 2, we apply different sparsification strategies tailored to specific generation stages of MLLMs. For the token sets $\mathcal{S}_l^I$ and $\mathcal{S}_l^T$ processed by $l$-th decoder layer in the prefill stage, we consider the reduction of the image tokens. The image predictor leverages the features of image tokens and predicts the binary mask to select the tokens that participate in the prefill computation.

Specifically, we consider the set $\mathcal{S}_l^I \in \mathbb{R}^{N_l^I \times d}$ as 2D matrices and the pipeline is defined as:

$$
\begin{aligned}
\mathcal{D}^I &= P^I(\mathcal{S}_l^I) \in \mathbb{R}^{N_l^I \times 2}, \\
\mathcal{M}^I &= \mathrm{argmax_j}(\mathcal{D}^I), \\
\mathcal{S}_l^{I*} &= \{\mathcal{S}_{l,i}^I | \mathcal{M}_i^I = 1 \wedge \forall i \in \mathcal{I}^I\},
\end{aligned}
\tag{5}
$$

where $P^I(\cdot)$ is the forward propagation of the image predictor, $\mathcal{D}^I$ is a feature sampling decision generated by $P^I$, and $\mathrm{argmax_j}$ indicates that we sample values of $\mathcal{D}^I$ along the second dimension. This indicates that a token should be kept if the second value is greater than the first value in the second dimension of $\mathcal{D}^I$. Then we replace the reduced token set $\mathcal{S}_l^{P*} = \mathcal{S}_l^{I*} \cup \mathcal{S}_l^T$ with $\mathcal{S}_l^P$ in Eq. 1 to perform the prefill stage.

For the decoding stage, we first consider decoding without KV cache. This operation is similar to the prefill stage except that the last output token generates the next output token during decoding. Thus we use the pipeline in Eq. 5 with $S_l^{OT}$ and $P^{OT}$, while keeping $\mathcal{M}_{N_l^{OT}}^{OT} = 1$ to obtain $S_l^{OT*}$, where $\mathcal{M}_{N_l^{OT}}^{OT}$ is the $N_l^{OT}$-th value of the mask $\mathcal{M}^{OT}$. We further replace the above $\mathcal{S}_l^{P*}$ and $\mathcal{S}_l^{OT*}$ with $\mathcal{S}_l^P$ and $\mathcal{S}_l^{OT}$ in Eq. 2 to conduct the decoding stage. Note that in both the prefill and decoding without KV cache stages, after the predictors reduce the token set $\mathcal{S}_l$ to $\mathcal{S}_l^*$, all subsequent layers use this reduced token set for computation, *i.e.*, the length of computed token sets is smaller to introduce the computation efficiency.

For decoding with KV cache, our primary goal is to reduce the sizes of KV cache, specifically $\mathcal{S}_l^K$ and $\mathcal{S}_l^V$, to achieve GPU memory savings during the decoding process. We use the last output token $\mathcal{S}_{l,N_l^{OT}}^{OT}$ and $P^{OT}$ to get a binary decision $\mathcal{M}_{N_l^{OT}}^{OT} \in \{0, 1\}$ by Eq. 5 and the decoding operation in Eq. 3 is modified to the following operation:

$$
\begin{aligned}
Q, K, V &= \{W_l^Q \mathcal{S}_{l,N_l^{OT}}^{OT}\}, \{W_l^K \mathcal{S}_{l,N_l^{OT}}^{OT}\}, \{W_l^V \mathcal{S}_{l,N_l^{OT}}^{OT}\}, \\
O &= W_O \, \mathrm{Attention}(Q, \mathcal{S}_l^K \cup K, \mathcal{S}_l^V \cup V), \\
&\begin{cases}
\mathcal{S}_l^K = \mathcal{S}_l^K \cup K, \, \mathcal{S}_l^V = \mathcal{S}_l^V \cup V, & \text{if} \quad \mathcal{M}_{N_l^{OT}}^{OT} = 1, \\
\mathcal{S}_l^K = \mathcal{S}_l^K \cup \emptyset, \, \mathcal{S}_l^V = \mathcal{S}_l^V \cup \emptyset, & \text{otherwise},
\end{cases} \\
\mathcal{S}_{l+1,N_{l+1}^{OT}}^{OT} &= \mathrm{FFN}(O),
\end{aligned}
\tag{6}
$$

where the binary mask $\mathcal{M}_{N_l^{OT}}^{OT}$ determines whether KV activations of the current output text token should be added to KV cache. Meanwhile, the decision to add the current token's activations to the KV cache is also propagated to subsequent layers, and the size of the KV cache for subsequent layers is reduced accordingly. Note that we still include this token in the current attention computation, regardless of whether it will be discarded in subsequent calculations. This ensures that this token contributes to the attention mechanism immediately, even if it does not persist in KV cache for future decoding steps.

### 3.3.3 END-TO-END SPARSIFICATION TRAINING

Unlike the sparsification inference phase, where the token sets are directly reduced, we employ the full-token sets along with binary masks to optimize the predictors and LLM in an end-to-end training process, as inspired by (Veit & Belongie, 2018; Herrmann et al., 2020; Rao et al., 2021; Lin et al., 2023). This approach ensures the model dynamically adjusts and learns which tokens are essential during training while maintaining the full set of tokens for comprehensive optimization. The details of the end-to-end sparsification training are presented in Fig. 3 and Appendix A.2.

Given the binary masks of the token sets, we need to use these masks to isolate the influence of non-essential tokens on the important tokens during the LLM training computation. One native idea is to directly set the values of the unnecessary tokens to zero vectors. However, this method introduces a problem: when we sparsify output text tokens during the parallel training of LLM, discarding the value of an output text token will prevent it from autoregressively predicting the next output text token, making it impossible to compute the loss. This "hard training" result is presented in Tab. 7. To address this, we apply the masks to the $\mathrm{Softmax}(\cdot)$ operation in $\mathrm{Attention}(\cdot)$ during training. Specifically, we firstly obtain a binary mask of the full-token set $\mathcal{M} = \mathcal{M}^I \cup \{1\}^{N_l^T} \cup \mathcal{M}^{OT}$.

Then we use the full-token set mask to get the binary mask matrix $\mathbb{G} = \{\mathcal{M}\}^{N_l} \in \mathbb{R}^{N_l \times N_l}$ and let $\text{diag}(\mathbb{G}) = 1$. The $\text{Softmax}(\cdot)$ is changed to $\text{MaskedSoftmax}(\cdot, \cdot)$ operation during training:

$$\text{MaskedSoftmax}(\mathbb{X}_{i,j}, \mathbb{G}) = \frac{\exp(\mathbb{X}_{i,j})\mathbb{G}_{i,j}}{\sum_{k=1}^{N_l} \exp(\mathbb{X}_{i,k})\mathbb{G}_{i,k}}, \tag{7}$$

where $\mathbb{X} \in \mathbb{R}^{N_l \times N_l}$ is the input of $\text{Softmax}(\cdot)$ operation. Eq. 7 allows our framework to maintain parallelism in training while ensuring that the non-essential tokens do not influence the output, without breaking the autoregressive process essential for loss calculation. Note that this form of masking and predictors maintains uniformity between training and inference. Specifically, during training, the presence of the causal attention mask (Brown, 2020) ensures that each token focuses only on prior context information, while the use of MLP in the output predictor $P^I$ ensures that decisions are based solely on its own features, consistent with the process used during decoding.

In addition to the issues mentioned above, we have the gradient flow problem in the backward propagation of training. The $\arg\max(\cdot)$ operation we performed to obtain $\mathcal{M}^I$ and $\mathcal{M}^{OT}$ is non-differentiable and impedes end-to-end training. Thus we use the Gumbel-Softmax (Jang et al., 2016) with Straight-Through Gradient Estimator (Bengio et al., 2013) to avoid the gradient flow problem. Taking the reduction of the image token set as an example and the $\arg\max(\cdot)$ in Eq. 5 is modified to the differentiable operation during training. The forward propagation is formulated as follows:

$$\mathcal{D}^{I\dagger} = \text{GumbelSoftmax}_j(\mathcal{D}^I, \tau),$$
$$\mathcal{M}^I = \arg\max_j(\mathcal{D}^{I\dagger}), \tag{8}$$

where $\tau$ is the temperature of the Gumbel-Softmax function. When $\tau$ becomes smaller, $\mathcal{D}^{I\dagger}$ smoothly approaches the discrete distribution. Following the work (Lin et al., 2023), Dynamic-LLaVA employ the exponential decay for $\tau$ from 1 to 0.1 during training. Meanwhile, in the backward propagation, we propagate the gradient flow from $\mathcal{M}^I$ to $\mathcal{D}^{I\dagger}$:

$$\frac{\partial \mathcal{L}}{\partial \mathcal{D}^{I\dagger}} = \frac{\partial \mathcal{L}}{\partial \mathcal{M}^I}, \tag{9}$$

where $\mathcal{L}$ is the objective loss function.

Furthermore, we incorporate a constraint regularization term to constrain binary masks according to a pre-defined keep rate for token sets. This constraint ensures that the predictors retains a certain proportion of tokens, adhering to the desired sparsification strategy. It is worth noting that we found that applying sparsification to $\mathcal{S}_l^{OT}$ on samples with short output tokens during training lead to instability training and performance degradation. To avoid this problem, we apply $\mathcal{S}_l^{OT}$ sparsification only to samples where output text tokens exceed a pre-defined length $\text{LEN}^{OT} \in \mathbb{N}_0$, where $\mathbb{N}_0$ denotes the set of non-negative integers. The constraint regularization term $\mathcal{R}$ can be defined as:

$$\mathcal{R} = \|\text{sum}(\mathcal{M}^I)/|\mathcal{S}_l^I| - r^I\|_F + \begin{cases} \|\text{sum}(\mathcal{M}^{OT})/|\mathcal{S}_l^{OT}| - r^{OT}\|_F, & \text{if } |\mathcal{S}_l^{OT}| \geq \text{LEN}^{OT}, \\ 0, & \text{otherwise}, \end{cases} \tag{10}$$

where $\text{sum}(\cdot)$ is the summation operation and $\|\cdot\|_F$ denotes the Frobenius norm. $r^I$ and $r^{OT}$ are the pre-defined keep rates of $\mathcal{S}_l^I$ and $\mathcal{S}_l^{OT}$, respectively. We directly add this term to the original objective loss function, with a hyper-parameter $\lambda$ used to control its weight. The trade-off analysis of vision context ($r^I$) and language context ($r^{OT}$) are presented in Tab. 5, while the ablation study of the sample used for training should have a minimum output text token length ($\text{LEN}^{OT}$) and the weight of the regularization term ($\lambda$) are shown in Tab. 6.

## 4 EXPERIMENTS

### 4.1 EXPERIMENTAL SETTINGS

**Datasets, Benchmarks, and Metrics.** During training, we use the instruction-tuning dataset 656K Mixture Dataset, as in LLaVA-1.5 (Liu et al., 2024a). For vision understanding evaluations, we use the commonly used vision understanding benchmarks to evaluate the performance similar as LLaVA-1.5, such as VQAv2 (Goyal et al., 2017), GQA (Hudson & Manning, 2019), VizWiz (Gurari et al., 2018), SciQA (Lu et al., 2022), TextVQA (Singh et al., 2019), POPE (Li et al., 2023b), MMBench (en) (Liu

Table 1: Comparison with SoTA vision context sparsification methods on vision understanding benchmarks. The best results are **bolded** and the second best results are underlined in all following tables. The sign "$I$" and "$I|T$" mean only the vision sparsification and the vision-language sparsification of Dynamic-LLaVA in all following tables, respectively. "TFLOPs" is only the computation consumption of the image tokens. Dynamic-LLaVA achieves the best performance in most benchmarks using a similar number of image tokens. The "Free" column indicates whether a method is training-free.

| Method | Free | Image (prefill) Token | TFLOPs | VQAv2 | GQA | SciQA | TextVQA | POPE | MME | MMBench | SEED | MMVP | RealWorldQA | CVBench-2D |
|---|---|---|---|---|---|---|---|---|---|---|---|---|---|---|
| LLaVA-1.5-7B | – | 576 | 10.1 | 78.5 | 62.0 | 66.8 | 58.2 | 85.9 | 1510.7 | 64.3 | 66.1 | 29.3 | 53.7 | 58.5 |
| (Arxiv24) LLaVA-PruMerge+ | ✗ | 146 (-75%) | 2.5 (-75%) | 76.8 (-1.7) | – | 68.3 (+1.5) | **57.1 (-1.1)** | 84.0 (-1.9) | 1462.4 (-48.3) | 64.9 (+0.6) | – | 24.0 (-5.3) | 53.7 (-0.0) | 56.7 (-1.9) |
| (ECCV24) LLaVA-FastV$_{k=3,r=0.75}$ | ✓ | 144 (-75%) | 3.2 (-68%) | 75.1 (-3.4) | 57.5 (-4.5) | 68.7 (+1.9) | 56.2 (-2.0) | 81.0 (-4.9) | 1458.9 (-51.8) | 63.5 (-0.8) | 62.8 (-3.3) | 24.0 (-5.3) | 53.7 (-0.0) | 56.7 (-1.9) |
| †(ECCV24) LLaVA-FastV$_{k=3,r=0.75}$ | ✗ | 144 (-75%) | 3.2 (-68%) | 74.2 (-4.3) | 56.6 (-5.4) | 64.0 (-2.8) | 54.3 (-3.9) | 82.7 (-3.2) | 1292.2 (-218.5) | 58.6 (-5.7) | 58.4 (-7.7) | 16.7 (-12.6) | 49.8 (-3.9) | 44.7 (-13.8) |
| (Arxiv24) VoCo-LLaMA | ✗ | 128 (-78%) | 2.2 (-78%) | 76.9 (-1.6) | 59.8 (-2.2) | – | – | – | – | 61.0 (-3.3) | – | – | – | – |
| (Arxiv24) LLaVA-HiRED | ✓ | 115 (-80%) | 2.0 (-80%) | 74.7 (-3.8) | – | 66.4 (-0.4) | 44.2 (-14.0) | – | – | – | 59.1 (-7.0) | – | – | – |
| Dynamic-LLaVA-7B$_I$ (Ours) | ✗ | 115 (-80%) | 2.5 (-75%) | **78.0 (-0.5)** | **61.4 (-0.6)** | **69.1 (+2.3)** | 57.0 (-1.2) | 85.0 (-0.9) | 1479.8 (-30.9) | **65.4 (+1.1)** | 64.6 (-1.5) | – | – | – |
| Dynamic-LLaVA-7B$_{I|T}$ (Ours) | ✗ | 115 (-80%) | 2.5 (-75%) | 77.9 (-0.6) | 61.3 (-0.7) | 68.6 (+1.8) | 56.5 (-1.7) | **85.9 (+0.0)** | **1501.0 (-9.7)** | 64.1 (-0.2) | **65.0 (-1.1)** | **26.3 (-3.0)** | **57.0 (+3.3)** | **58.3 (-0.2)** |
| ‡LLaVA-1.5-7B+H2O$_{r=0.5}$ | ✓ | 576 | 10.1 | – | – | – | 11.6 (-46.6) | 50.5 (-30.4) | 500.5 (-1010.2) | – | – | – | – | – |
| ‡LLaVA-1.5-7B+H2O$_{k=10,r=0.5}$ | ✓ | 576 | 10.1 | 77.9 (-0.6) | 61.0 (-1.0) | 41.9 (-16.3) | 55.9 (-2.3) | 86.9 (+1.0) | 1458.4 (-52.3) | 64.1 (-0.2) | 26.8 (-39.3) | 0 (-29.3) | 42.3 (-11.4) | 49.7 (-8.8) |
| LLaVA-1.5-13B | – | 576 | 19.6 | 80.0 | 63.3 | 71.6 | 61.3 | 85.9 | 1531.3 | 67.7 | 68.2 | 30.7 | 55.3 | 62.3 |
| (Arxiv24) LLaVA-PruMerge+ | ✗ | 146 (-75%) | 4.9 (-75%) | 77.8 (-2.2) | – | 71.0 (-0.6) | 58.6 (-2.7) | 84.4 (-1.5) | 1485.5 (-45.8) | 65.7 (-2.0) | – | 28.0 (-2.7) | 52.8 (-2.5) | 58.9 (-3.4) |
| (ECCV24) LLaVA-FastV$_{k=3,r=0.75}$ | ✓ | 144 (-75%) | 6.0 (-69%) | 77.0 (-3.0) | 60.1 (-3.2) | **72.8 (+1.2)** | 59.0 (-2.3) | 83.2 (-2.7) | 1470.3 (-61.0) | 66.9 (-0.8) | 65.4 (-2.8) | 28.0 (-2.7) | 52.8 (-2.5) | 58.9 (-3.4) |
| Dynamic-LLaVA-13B$_I$ (Ours) | ✗ | 115 (-80%) | 4.7 (-76%) | **79.1 (-0.9)** | **62.7 (-0.6)** | 72.2 (+0.6) | 59.5 (-1.8) | **86.8 (+0.9)** | 1554.1 (+22.8) | **68.3 (+0.6)** | 66.6 (-1.6) | – | – | – |
| Dynamic-LLaVA-13B$_{I|T}$ (Ours) | ✗ | 115 (-80%) | 4.7 (-76%) | 78.8 (-1.2) | 62.5 (-0.8) | 72.4 (+0.8) | **59.6 (-1.7)** | 86.5 (+0.6) | **1563.3 (+32.0)** | 66.9 (-0.8) | 66.5 (-1.7) | **31.3 (+0.6)** | **53.3 (-2.0)** | **60.7 (-1.6)** |
| ‡LLaVA-1.5-13b+H2O$_{k=10,r=0.5}$ | ✓ | 576 | 19.6 | 78.9 (-1.1) | 62.3 (-1.0) | 48.5 (-23.1) | 57.5 (-3.8) | 87.3 (+1.4) | 1448.3 (-83.0) | 4.7 (-59.6) | 32.3 (35.9) | 0 (-30.7) | 43.9 (-11.4) | 50.8 (-11.5) |

† It means that we apply one additional epoch sparsification training for FastV. For a fair comparison, the settings are the same for Dynamic-LLaVA. The details are presented in Appendix A.4.2.
‡ It means that we directly apply the LLM KV cache compression method H2O to MLLMs. **Note that H2O cannot provide the significant computation efficiency in the prefill stage of MLLMs.** The details are presented in Appendix A.4.2.

Table 2: Comparison with SoTA efficient vision projector methods on vision understanding benchmarks. Dynamic-LLaVA uses the original vision projector and achieves competitive performance.

| Method | Projector | Image (prefill) Res. | Token | TFLOPs | MMBench | MM-Vet | VQAv2 | GQA | POPE | VizWiz | Avg. |
|---|---|---|---|---|---|---|---|---|---|---|---|
| LLaVA-1.5-7B | MLP | 336 | 576 | 10.1 | 64.3 | 31.1 | 78.5 | 62.0 | 85.9 | 50.0 | 62.0 |
| (Arxiv23) LLaVA-Resampler-7B | Resampler | 336 | 144 | 2.5 | 63.1 | 29.2 | 75.1 | 58.4 | 84.7 | **51.9** | 60.4 |
| (CVPR24) LLaVA-C-Abstractor-7B | C-Abstractor | 336 | 144 | 2.5 | 63.1 | 29.4 | 74.6 | 59.2 | 84.6 | 49.2 | 60.0 |
| (Arxiv24) LLaVA-Pixel-Shuffle-7B | Pixel-Shuffle | 336 | 144 | 2.5 | 64.0 | 29.7 | 76.2 | 60.1 | 85.9 | 48.8 | 60.8 |
| (Arxiv24) LLaVA-LDP-v2-7B | LDP-v2 | 336 | 144 | 2.5 | **66.2** | 28.7 | 77.3 | 61.1 | 86.1 | 47.6 | 61.2 |
| Dynamic-LLaVA-7B$_I$ (Ours) | MLP | 336 | 115 | 2.5 | 65.4 | 29.5 | **78.0** | **61.4** | **86.2** | 50.2 | 61.8 |
| Dynamic-LLaVA-7B$_{I|T}$ (Ours) | MLP | 336 | 115 | 2.5 | 64.1 | **32.2** | 77.9 | 61.3 | 85.9 | 51.2 | **62.1** |
| LLaVA-TokenPacker-7B-144Token | TokenPacker | 336 | 144 | 2.5 | 65.1 | 31.7 | 77.9 | 61.8 | 87.0 | 52.0 | 62.6 |
| (Arxiv24) LLaVA-TokenPacker-7B-64Token | TokenPacker | 336 | 64 | 1.1 (-56%) | 64.1 | – | 77.2 | 61.1 | **86.3** | 50.7 | – |
| (ECCV24) LLaVA-TokenPacker-FastV-7B$_{k=3,r=0.5}$ | TokenPacker | 336 | 72 | 1.4 (-44%) | 65.0 | 32.0 | 76.6 | 60.8 | 85.2 | 51.4 | 61.8 |
| Dynamic-LLaVA-TokenPacker-7B$_I$ (Ours) | TokenPacker | 336 | 57 | 1.1 (-56%) | **65.1** | **32.1** | **77.8** | **61.9** | 85.9 | 51.2 | **62.3** |
| Dynamic-LLaVA-TokenPacker-7B$_{I|T}$ (Ours) | TokenPacker | 336 | 57 | 1.1 (-56%) | 63.5 | 31.9 | 77.6 | 61.7 | 86.0 | 51.2 | 62.0 |
| LLaVA-TokenPacker-13B-144Token | TokenPacker | 336 | 144 | 4.9 | 68.0 | 34.6 | 78.3 | 62.5 | 87.4 | 55.6 | 64.5 |
| (Arxiv24) LLaVA-TokenPacker-13B-64Token | TokenPacker | 336 | 64 | 2.2 (-55%) | 66.2 | – | 78.1 | 62.0 | **87.3** | 52.9 | – |
| (ECCV24) LLaVA-TokenPacker-FastV-13B$_{k=3,r=0.5}$ | TokenPacker | 336 | 72 | 2.6 (-47%) | 66.8 | 34.5 | 77.5 | 61.5 | 86.3 | 54.9 | 63.6 |
| Dynamic-LLaVA-TokenPacker-13B$_I$ (Ours) | TokenPacker | 336 | 57 | 2.1 (-57%) | **67.4** | 34.8 | **78.3** | **62.5** | 86.7 | **56.4** | **64.4** |
| Dynamic-LLaVA-TokenPacker-13B$_{I|T}$ (Ours) | TokenPacker | 336 | 57 | 2.1 (-57%) | 66.8 | **37.1** | 78.3 | 62.1 | 86.3 | 54.8 | 64.2 |

et al., 2023b), SEED (image) (Li et al., 2023a) and MM-Vet (Yu et al., 2023). Furthermore, we also use the vision-centric vision understanding benchmarks, such as MMVP (Tong et al., 2024b), RealWorldQA (xAI, 2024) and CVBench-2D (Tong et al., 2024a). We employ the same evaluation metrics of the above benchmarks to assess the results.

Specially, to evaluate the generation ability of MLLM before and after the generated language context sparsification. We constructed LVIS-VQA single- and multi-round Benchmarks based on the subset of LVIS-Instruct4V (Wang et al., 2023). Meanwhile, we constructed a single-round long generation text benchmark ShareGPT4V-VQA by ShareGPT4V dataset (Chen et al., 2023b). Details of these datasets are presented in Appendix A.4.3 and A.4.4. To ensure a fair comparison, none of the images selected for our benchmarks were included in the training set of the base MLLMs and Dynamic-LLaVA. We evaluate the fluency of the generated responses of MLLMs using the Perplexity Metric (PPL) (Jelinek et al., 1977), and the similarity of the generated answers to the standard answers using the Metric for Evaluation of Translation with Explicit ORdering (METEOR) (Banerjee & Lavie, 2005).

**Implementations.** All of the methods are trained on 8 NVIDIA A100 (80G) using Pytorch (Paszke et al., 2019). For fair comparisons, we utilize the open-source MLLMs weights available in the official GitHub repositories and continue one-epoch instruction-tuning for MLLMs and learnable predictors. The hyper-parameters of which decoder layer to sparsify the tokens ($l$), the sample used for training should have a minimum output text token length ($\text{LEN}^{OT}$) and the weight of the regularization term ($\lambda$) are set to 2, 50 and 100 in all experiments, respectively. The details are presented in Appendix A.4.2.

## 4.2 MAIN RESULTS

**Vision understanding.** As demonstrated in Tab. 1, our proposed Dynamic-LLaVA framework achieves superior performance compared to SoTA vision context sparsification methods in most benchmarks while reducing image tokens by approximately 80%. Additionally, the Dynamic-LLaVA framework reduces FLOPs by more than 70% for official LLaVA-1.5 base-

Table 3: Comparison of generation ability benchmarks. "Total→Computing" means the total generated token number and the maximum number of tokens involved in decoding. "a/b" means the results in experiments of PPL/METEOR. "TFLOPs" and "Mem." in "Text (decoding)" are measured under decoding w/o and w/ KV cache, respectively, while "Mem." means maximum GPU memory of KV cache. Dynamic-LLaVA achieves significant inference efficiency during the decoding stage with negligible generation ability drop when sparsifying both vision and language.

| Benchmark | Method | Image (prefill) TFLOPs | Text (decoding) Total→Computing | Text (decoding) TFLOPs | Text (decoding) Mem. (M) | PPL↓ | METEOR↑ |
|---|---|---|---|---|---|---|---|
| LVIS-VQA (single-round) | LLaVA-1.5-7B | 10.1 | 159/173→159/173 | 2.75/2.99 | 103/91 | 4.59 | 0.3103 |
| | LLaVA-FastV-7B$_{k=3,r=0.75}$ | 3.2 (-68%) | 159/172→159/172 | 2.75/2.99 | 103/89 | 4.67 | 0.3074 |
| | Dynamic-LLaVA-7B$_I$ (Ours) | 2.5 (-75%) | 159/174→159/174 | 2.75/3.02 | 102/89 | 4.70 | 0.3099 |
| | Random | 10.1 | 159/570→79/285 | 2.67/4.98 | 62 (-40%)/157 | 11.52 | 0.2080 |
| | Structure | 10.1 | 159/603→79/302 | 2.68/5.72 | 63 (-39%)/166 | 9.69 | 0.1865 |
| | H2O$_{r=0.5}$ | 10.1 | 159/589→79/295 | – | 62 (-40%)/158 | 78.95 | 0.0388 |
| | †FastV-7B$_{k=3,r=0.75}$+H2O$_{r=0.5}$ | 3.2 (-68%) | 159/168→79/84 | – | 61 (-41%)/58 (-36%) | 5.62 | 0.3071 |
| | Dynamic-LLaVA-7B$_{I|T}$ (Ours) | 2.5 (-75%) | 159/181→84/90 | 1.52 (-45%)/1.57 (-47%) | 63 (-39%)/46 (-49%) | 4.90 | 0.3108 |
| | LLaVA-1.5-13B | 19.6 | 159/174→159/174 | 5.36/5.88 | 149/143 | 4.39 | 0.3148 |
| | LLaVA-FastV-13B$_{k=3,r=0.75}$ | 6.0 (-69%) | 159/176→159/176 | 5.36/5.88 | 152/143 | 4.46 | 0.3118 |
| | Dynamic-LLaVA-13B$_I$ (Ours) | 4.7 (-76%) | 159/178→159/178 | 5.36/6.01 | 148/144 | 4.55 | 0.3148 |
| | Random | 19.6 | 159/262→79/131 | 2.67 (-50%)/4.43 | 88 (-41%)/113 | 10.47 | 0.2774 |
| | Structure | 19.6 | 159/330→79/165 | 2.68 (-50%)/5.60 | 88 (-41%)/145 | 8.39 | 0.2393 |
| | H2O$_{r=0.5}$ | 19.6 | 159/152→79/76 | – | 86 (-43%)/62 (-57%) | 41.44 | 0.0428 |
| | †FastV-13B$_{k=3,r=0.75}$+H2O$_{r=0.5}$ | 6.0 (-68%) | 159/168→79/84 | – | 84 (-44%)/89 (-38%) | 5.30 | 0.3151 |
| | Dynamic-LLaVA-13B$_{I|T}$ (Ours) | 4.7 (-76%) | 159/179→84/88 | 2.95 (-44%)/3.06 (-48%) | 90 (-40%)/72 (-50%) | 4.76 | 0.3151 |
| LVIS-VQA (multi-round) | LLaVA-1.5-7B | 10.1 | 351/461→351/461 | 6.12/8.08 | 222/236 | 2.97 | 0.4227 |
| | LLaVA-FastV-7B$_{k=3,r=0.75}$ | 3.2 (-68%) | 351/463→351/463 | 6.12/8.12 | 227/242 | 3.06 | 0.4159 |
| | Dynamic-LLaVA-7B$_I$ (Ours) | 2.5 (-75%) | 351/491→351/491 | 6.12/8.61 | 223/251 | 3.08 | 0.4222 |
| | Random | 10.1 | 351/1160→176/580 | 3.04 (-50%)/10.35 | 141 (-38%)/328 | 5.96 | 0.3290 |
| | Structure | 10.1 | 351/951→176/475 | 3.04 (-50%)/8.47 | 141 (-38%)/270 | 5.40 | 0.2820 |
| | H2O$_{r=0.5}$ | 10.1 | 351/503→99/175 | – | 133 (-41%)/149 (-38%) | 42.6 | 0.0892 |
| | †FastV-7B$_{k=3,r=0.75}$+H2O$_{r=0.5}$ | 3.2 (-68%) | 351/482→175/240 | 3.04 (-50%)/4.19 (-48%) | 140 (-38%)/128 (-47%) | 3.57 | 0.4049 |
| | Dynamic-LLaVA-7B$_{I|T}$ (Ours) | 2.5 (-75%) | 351/522→182/260 | 3.33 (-45%)/4.38 (-46%) | 144 (-35%)/140 (-41%) | 3.17 | 0.4251 |
| | LLaVA-1.5-13B | 19.6 | 351/497→351/497 | 11.92/16.97 | 331/405 | 2.88 | 0.4240 |
| | LLaVA-FastV-13B$_{k=3,r=0.75}$ | 6.0 (-69%) | 351/501→351/501 | 11.92/17.11 | 326/399 | 2.96 | 0.4193 |
| | Dynamic-LLaVA-13B$_I$ (Ours) | 4.7 (-76%) | 351/507→351/507 | 11.92/17.30 | 328/405 | 2.98 | 0.4221 |
| | Random | 19.6 | 351/600→175/300 | 5.92 (-50%)/10.23 | 195 (-41%)/265 (-35%) | 5.40 | 0.3900 |
| | Structure | 19.6 | 351/671→176/335 | 5.93 (-50%)/11.48 | 194 (-41%)/303 (-25%) | 4.67 | 0.3627 |
| | H2O$_{r=0.5}$ | 19.6 | 351/144→99/72 | – | 183 (-45%)/283 (-30%) | 18.62 | 0.0493 |
| | †FastV-13B$_{k=3,r=0.75}$+H2O$_{r=0.5}$ | 6.0 (-69%) | 351/493→175/247 | – | 186 (-44%)/197 (-51%) | 3.45 | 0.4127 |
| | Dynamic-LLaVA-13B$_{I|T}$ (Ours) | 4.7 (-76%) | 351/494→181/242 | 6.40 (-46%)/8.27 (-51%) | 196 (-41%)/202 (-50%) | 3.07 | 0.4243 |
| ShareGPT4V-VQA (single-round) | LLaVA-1.5-7B | 10.1 | 1555/100→1555/100 | 28.70/1.75 | 1024/50 | 2.52 | ‡0.0540 |
| | Random | 10.1 | 1555/450→779/224 | 13.97 (-51%)/3.95 | 664 (-35%)/120 | 6.07 | ‡0.0560 |
| | Structure | 10.1 | 1555/531→777/266 | 13.93 (-51%)/4.68 | 644 (-37%)/144 | 30.7 | ‡0.0484 |
| | H2O$_{r=0.5}$ | 10.1 | 1555/392→777/196 | – | 607 (-41%)/107 | 25.54 | ‡0.0185 |
| | †FastV-7B$_{k=3,r=0.75}$+H2O$_{r=0.5}$ | 3.2 (-68%) | 1555/112→777/56 | – | 604 (-41%)/27 (-46%) | 7.56 | ‡0.0537 |
| | Dynamic-LLaVA-7B$_{I|T}$ (Ours) | 2.5 (-75%) | 1555/102→784/51 | 14.97 (-47%)/0.95 (-46%) | 625 (-39%)/28 (-44%) | 3.14 | ‡0.0551 |
| | LLaVA-1.5-13B | 19.6 | 1555/103→1555/103 | 55.40/3.49 | 1489/83 | 2.43 | ‡0.0523 |
| | Random | 19.6 | 1555/192→777/97 | 27.01 (-51%)/3.30 | 875 (-41%)/265 | 5.32 | ‡0.0589 |
| | Structure | 19.6 | 1555/285→777/142 | 27.04 (-51%)/4.88 | 876 (-41%)/126 | 21.10 | ‡0.0489 |
| | H2O$_{r=0.5}$ | 19.6 | 1555/72→777/36 | – | 840 (-43%)/29 | 10.51 | ‡0.0131 |
| | †FastV-13B$_{k=3,r=0.75}$+H2O$_{r=0.5}$ | 6.0 (-79%) | 1555/115→777/57 | – | 835 (-44%)/45 (-46%) | 3.41 | ‡0.0507 |
| | Dynamic-LLaVA-13B$_{I|T}$ (Ours) | 4.76 (-76%) | 1555/100→763/50 | 27.96 (-49%)/1.78 (-49%) | 854 (-43%)/44 (-46%) | 3.10 | ‡0.0524 |

† It means that we use FastV to reduce the token set of prefill, while using H2O to only compress generated KV cache during decoding. The details are presented in Appendix A.4.2.
‡ In long text generation tasks, the substantial discrepancy between lengths of text generated by MLLMs and the label text leads to a significant degradation of the METEOR metric.

Table 4: Total time and maximum GPU memory overhead during MLLMs' generation. "2K/4K" means the number of the generated output text tokens. "×" means the generation process has failed due to the OOM. The results are measured in one A100 (80G) and the batch size is fixed to 8.

| Method | Prefill Time | Decoding w/o KV cache 1K Time | Decoding w/o KV cache 2K Time | Decoding w/o KV cache 4K Time | Decoding w/ KV cache 1K Mem. | Decoding w/ KV cache 2K Mem. | Decoding w/ KV cache 4K Mem. |
|---|---|---|---|---|---|---|---|
| LLaVA-1.5-13B | 0.83s | 1453s | 4117s | 13368s | 46G | 58G | × |
| LLaVA-FastV-13B$_{k=3,r=0.75}$ | 0.43s | 1079s | 3462s | × | 41G | 53G | 77G |
| Dynamic-LLaVA-13B$_{I|T}$ | 0.37s | 838s | 2382s | 6184s | 35G | 42G | 56G |

† We apply the Parallel Sparsification Inference for FastV to achieve the batch-parallel inference.

lines in the prefill stage, with only minimal performance degradation across most benchmarks. Especially, for the LLaVA-1.5 with 7B and 13B parameters, the Dynamic-LLaVA even shows performance improvements on the SciQA, POPE, MME, and MMBench benchmarks. For instance, on the SciQA benchmark, the 7B and 13B Dynamic-LLaVA models achieve performance gains of +2.3% and +0.8%, respectively, compared to the original LLaVA-1.5. More comparison of SoTA vision context sparsification methods are presented in Appendix A.5.1.

Table 5: Trade-off of the keep rates of vision context ($r^I$) and language context ($r^{OT}$).

| Context | Rate | Token Vision | Token Language | VQAv2 | GQA | MMVP | LVIS (single) PPL↓ | LVIS (single) MET.↑ |
|---|---|---|---|---|---|---|---|---|
| Vision | 20% | 115 | 84/90 | 77.9 | 61.3 | 26.3 | 4.90 | 0.3108 |
| | 50% | 288 | 83/86 | 78.8 | 62.3 | 26.3 | 4.88 | 0.3107 |
| | 80% | 461 | 83/85 | 78.9 | 62.5 | 26.2 | 4.88 | 0.3090 |
| Language | 20% | 115 | 41/39 | 77.6 | 61.4 | 26.0 | 5.53 | 0.2592 |
| | 50% | 115 | 84/90 | 77.9 | 61.3 | 26.3 | 4.90 | 0.3108 |
| | 80% | 115 | 129/138 | 78.0 | 61.6 | 26.0 | 4.76 | 0.3116 |
| LLaVA-7B | 100% | 576 | 159/173 | 78.5 | 62.0 | 29.3 | 4.59 | 0.3103 |

As shown in Tab. 2, for efficient vision projector methods for MLLMs, Dynamic-LLaVA exclusively utilizes the original MLP of LLaVA-1.5 as the vision projector. This approach surpasses those that modify the vision projector across most benchmarks, achieving an average performance that exceeds the best alternative methods by 0.9%. Furthermore, our Dynamic-LLaVA framework can be integrated with other efficient vision projector methods. For instance, our combination with the TokenPacker projector, Dynamic-LLaVA-TokenPacker, achieves significant reductions in vision tokens by an additional 60% on top of the already reduced count by efficient vision projector methods, with only a minimal loss in performance. Compared to the official LLaVA-TokenPacker method, our Dynamic-LLaVA-TokenPacker incurs only a minimal performance loss of 0.3% and 0.1% (for 7B and 13B models) while using fewer vision tokens (57 vs. 144). Moreover, we observe that the sparsification of the output text tokens does not impede the comprehension ability of MLLMs. This finding is supported by comparisons between Dynamic-LLaVA$_I$ and Dynamic-LLaVA$_{I|T}$, as detailed in last two rows of Tab. 1 and Tab. 2.

**Generation ability.** To evaluate the impact of vision-language context sparsification on generation ability, we utilize the LVIS-VQA (single-round) and LVIS-VQA (multi-round) as benchmarks.

To demonstrate the advantages of Dynamic-LLaVA in dynamically sparsifying the language context, we compare it against traditional static methods, "Random" and "Structure". "Random" refers to randomly discarding ∼50% output text tokens, while "Structure" involves discarding every alternate output text token.

Table 6: Hyper-parameter ablation for $l$, $\text{LEN}^{OT}$ and $\lambda$.

| Hyper | Value | Benchmark | | |
|---|---|---|---|---|
| | | VQAv2 | GQA | POPE |
| $l$ | 1 | 77.6 | 61.3 | 85.7 |
| | **2** | 77.9 | 61.3 | 85.9 |
| | 4 | 77.9 | 61.4 | 85.8 |
| $\text{LEN}^{OT}$ | 0 | 77.4 | 61.0 | 85.5 |
| | **50** | 77.9 | 61.3 | 85.9 |
| | 100 | 77.8 | 61.2 | 86.1 |
| $\lambda$ | 10 | 77.9 | 61.2 | 85.8 |
| | **100** | 77.9 | 61.3 | 85.9 |
| | 1000 | 77.7 | 61.3 | 85.7 |

As shown in Tab. 3, Dynamic-LLaVA maintains nearly unchanged generation fluency (by PPL) and quality (by METEOR) when only the vision context is sparsified. When sparsifying both vision and language contexts, Dynamic-LLaVA exhibits a slight increase in PPL (less than 0.3) and a minor improvement in METEOR score compared to full-context baselines, while also reducing ∼50% FLOPs (decoding without KV cache) and GPU memory overhead (decoding with KV cache) of the output text tokens. In comparison with the traditional static methods under the same condition of discarding ∼50% output text tokens, both "Random" and "Structure" methods significantly underperform in terms of fluency and quality of generation compared to Dynamic-LLaVA's language context sparsification. Furthermore, compared to methods only focusing on the image token reduction, Dynamic-LLaVA achieves additional inference efficiency during the decoding stage while maintaining nearly consistent generation ability. Meanwhile, the LLM KV cache compression method H2O, directly discarding historical KV cache does not adapt well to mixed-modality contexts, causing a significant performance degradation. We enhance the generation ability by integrating FastV with H2O. Despite these reductions, Dynamic-LLaVA still maintains a slight advantage in generation ability.

**Practical inference efficiency.** As presented in Tab. 4, we measured the practical inference time and GPU memory usage of MLLMs. The proposed Dynamic-LLaVA framework can perform parallel inference with mini-batch samples, due to the Parallel Sparsification Inference strategy in Appendix A.1. Vision token reduction methods provide noticeable inference benefits during the prefill stage and play a role in subsequent generations. However, the proportion of this efficiency gradually decreases throughout the entire generation process. Dynamic-LLaVA not only achieves significant inference speed improvements during the prefill stage but also reduces generation time by ∼50% when decoding without KV cache and significantly lowers GPU memory overhead when decoding with KV cache. Moreover, we further report the practical inference lantency of the decoding stage with KV cache, which are presented in Appendix A.5.2.

### 4.3 ABLATION STUDY

**Trade-off.** We respectively adjust the keep rates of vision context ($r^I$) and language context ($r^{OT}$) during training, as mentioned in Sec. 3.3.3. As in Tab. 5, we observe that increasing the value of $r^I$, i.e., retaining more image tokens during inference, significantly enhances the vision understanding performance of Dynamic-LLaVA. On the other hand, adjusting the value of $r^{OT}$ has negligible effects on vision understanding but impacts the generation ability of Dynamic-LLaVA. By tuning the values of $r^I$ and $r^{OT}$ during training, it is possible to achieve a balance between performance and efficiency tailored to specific requirements. More detailed results are presented in Appendix A.5.4.

**Hyper-parameters.** Tab. 6 provides an analysis of which decoder layer to sparsify the tokens ($l$) and the sample used for training should have a minimum output text token length ($\text{LEN}^{OT}$), as mentioned in Sec. 3.3. This analysis enables the Dynamic-LLaVA framework to be easily integrated into various MLLMs. For $l$ and $\lambda$, Dynamic-LLaVA shows relative insensitivity in vision understanding performance. We follow the work (Chen et al., 2024a) and let $l = 2$, while $\lambda = 100$. For $\text{LEN}^{OT}$, sparsifying output text tokens of all samples significantly reduces the understanding performance of Dynamic-LLaVA. Incrementally increasing the threshold of $\text{LEN}^{OT}$ results in diminishing returns.

## 5 CONCLUSION

In this paper, we proposed a dynamic vision-language context sparsification framework, termed Dynamic-LLaVA. This framework is designed with tailored sparsification inference strategies for the inference modes of MLLMs and can be integrated into MLLMs' training in an end-to-end manner.

ACKNOWLEDGMENTS

This work is partly supported by the National Natural Science Foundation of China (NO. 62102151), the Open Research Fund of Key Laboratory of Advanced Theory and Application in Statistics and Data Science, Ministry of Education (KLATASDS2305), the Fundamental Research Funds for the Central Universities.

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

CONTENTS OF APPENDIX

# A  APPENDIX

## A.1  PARALLEL SPARSIFICATION INFERENCE OPTIMIZATION

In this section, we introduce the parallel inference optimization for the sparsification inference described in Sec. 3.3.2. We denote the mini-batch form of $\mathcal{S}_l^P$ and $\mathcal{S}_l^{OT}$ in the $l$-th decoder layer as $\mathbb{S}_l^P$ and $\mathbb{S}_l^{OT}$, respectively. We treat these as matrices, thus $\mathbb{S}_l^P = \{\mathcal{S}_l^{P(b)} | \forall b \in \{1, 2, \cdots, B\}\} \in \mathbb{R}^{B \times (N_l^I + N_l^T) \times d}$ and $\mathbb{S}_l^{OT} = \{\mathcal{S}_l^{OT(b)} | \forall b \in \{1, 2, \cdots, B\}\} \in \mathbb{R}^{B \times N_l^{OT} \times d}$, where $B$ represents the mini-batch size, and $\mathcal{S}_l^{P(b)}$ and $\mathcal{S}_l^{OT(b)}$ refer to the token set for the $b$-th sample within the batch.

For the prefill stage, we pad the indefinite length mini-batch image token sets and use one-pass parallel inference of the predictor for the padded mini-batch image token set $\mathbb{S}_l^I = \{\text{LPadding}(\mathcal{S}_l^{I(b)}) | \forall b \in \{1, 2, \cdots, B\}\} \in \mathbb{R}^{B \times \max(\mathbb{N}_l^I) \times d}$ to obtaion the reduced token set $\mathbb{S}_l^{P*}$ for prefill, where $\mathbb{N}_l^I$ is the sizes of the mini-batch image token set, $\max(\mathbb{N}_l^I)$ denotes the maximum size of the image token sets within the mini-batch and $\text{LPadding}(\cdot)$ represents the operation of padding zero values in the left of the input token set, extending its size to match the maximum size $\max(\mathbb{N}_l^I)$. Considering the computation parallelization, we directly select the tokens with high predictor scores to retain, and the pipeline in Eq. 5 is modified as:

$$
\begin{aligned}
\mathbb{D}^I &= P^I(\mathbb{S}_l^I) \in \mathbb{R}^{B \times \max(\mathbb{N}_l^I) \times 2}, \\
\mathcal{S}_l^{I*(b)} &= \{\mathbb{S}_{l,i}^{I(b)} | \forall i \in \text{TopkArgmax}_{\lfloor r^I | \mathcal{S}_l^{I(b)} | \rfloor}(\mathbb{D}_{*,2}^{I(b)})\}, \\
\mathbb{S}_l^{P*} &= \{\text{LPadding}(\mathcal{S}_l^{I*(b)} \cup \mathcal{S}_l^{T(b)}) | \forall b \in \{1, 2, \cdots, B\}\} \in \mathbb{R}^{B \times \max(\mathbb{N}_l^{P*}) \times d},
\end{aligned}
\tag{11}
$$

where $\text{TopkArgmax}_k(\cdot)$ the top argmax opeartion with the number of $k$ tokens and $\lfloor \cdot \rfloor$ is a floor function. $\mathbb{D}_{*,2}^{I(b)}$ represents the extraction of all the second values along the last dimension of $\mathbb{D}^{I(b)}$, serving as the predictor score, with $\mathbb{D}_{*,2}^{I(b)} \in \mathbb{R}^{\max(\mathbb{N}_l^I)}$. It allows us to retain a fixed proportion $r^I$ of the image tokens for each batch's image token set, while simultaneously enabling batch-parallel processing for both the predictor's predictions and the subsequent computations within LLM.

For the decoding without KV cache, we use $\mathbb{S}_l^{OT}$ to get $\mathbb{S}_l^{OT*}$ for computation parallelization:

$$
\begin{aligned}
\mathbb{D}^{OT} &= P^{OT}(\mathbb{S}_l^{OT}) \in \mathbb{R}^{B \times \max(\mathbb{N}_l^{OT}) \times 2}, \quad \mathcal{M}^{OT(b)} = \text{argmax}_j(\mathbb{D}^{OT(b)}), \\
\mathcal{S}_l^{OT*(b)} &= \{\mathbb{S}_{l,i}^{OT(b)} | \mathcal{M}^{OT(b)} = 1 \wedge \forall i \in \mathcal{I}^{OT}\}, \\
\mathbb{S}_l^{OT*} &= \{\text{LPadding}(\mathcal{S}_l^{OT*(b)}) | \forall b \in \{1, 2, \cdots, B\}\} \in \mathbb{R}^{B \times \max(\mathbb{N}_l^{OT*}) \times d}.
\end{aligned}
\tag{12}
$$

Meanwhile, considering the decoding with KV cache, we store a KV cache for the each batch token set and the mini-batch KV cache set can be define as $\{\{\mathcal{S}_l^{K(b)}, \mathcal{S}_l^{V(b)}\} | \forall b \in \{1, 2, \cdots, B\}\}$. We apply a similar operation as in Eq. 12 to each last output text token $\mathcal{S}_{l,N_l^{OT}}^{OT(b)}$, resulting in the batch-wise binary decision $\mathcal{M}_{N_l^{OT}}^{OT(b)} \in \{0, 1\}$, which determines whether to add the activations to $\{\mathcal{S}_l^{K(b)}, \mathcal{S}_l^{V(b)}\}$ as outlined in Eq. 6. For the computation in the $\text{Attention}(\cdot, \cdot, \cdot)$ operation, we utilize the padded KV cache sets $\{\mathbb{S}_l^K, \mathbb{S}_l^V\} = \{\{\text{LPadding}(\mathbb{S}_l^{K(b)}) | \forall b \in \{1, 2, \cdots, B\}\}, \{\text{LPadding}(\mathbb{S}_l^{V(b)}) | \forall b \in \{1, 2, \cdots, B\}\}\}$. In this way, we reduce the activations stored in KV cache, and use these reduced activations to participate in the computation of $\text{Attention}(\cdot, \cdot, \cdot)$. Note that $|\mathbb{S}_l^{OT*}| = \max(\mathbb{N}_l^{OT*}) \approx r^{OT} |\mathcal{S}_l^{OT}|$ due to we use Eq. 10 to constrain the number of the output text token set during training, and ensures that each batch $\mathbb{S}_l^{OT*(b)}$ adheres to a close keep rate of $r^{OT}$ during inference.

## A.2  DETAILED PIPELINE OF END-TO-END SPARSIFICATION TRAINING

The mask generation pipeline of the MaskedSoftmax operation is presented in the above figure of Fig. 3. Specifically, we use the mask generated by the predictors to create a matrix for the MaskedSoftmax operation, somewhat analogous to the attention mask in the Multi-Head Attention Block. When values in this matrix are zero, they engage the MaskedSoftmax to set corresponding attention scores in the attention matrix produced by $Q$ and $K$ to zero. This effectively isolates the influence of non-essential tokens on essential tokens during training.

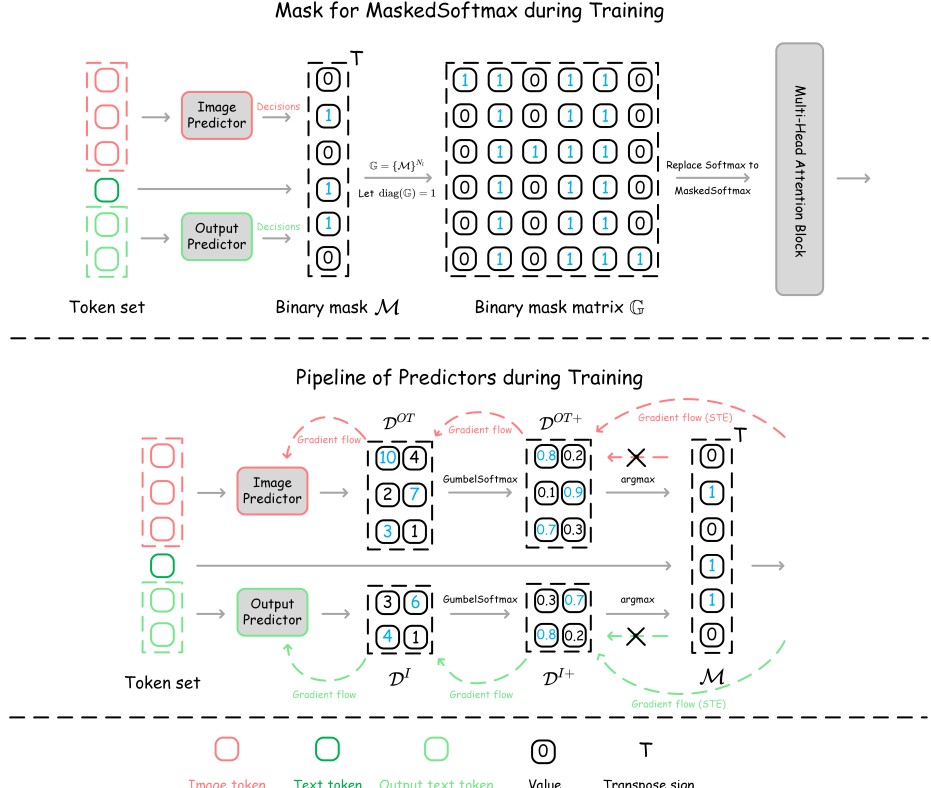

Figure 3: The detailed training pipeline of Dynamic-LLaVA. **Above Figure:** the mask for Masked-Softmax operation during training. We utilize the predictors to generate the binary mask $\mathcal{M}$ and subsequently form a binary mask matrix $\mathbb{G}$. This generated binary mask matrix is employed in the Multi-Head Attention Block within the MaskedSoftmax operation to isolate the influence of non-essential tokens on essential tokens during training. **Bottom Figure:** the pipeline of predictors during training. In the forward propagation, we use GumbelSoftmax function to relax the decision matrix $D^I$ and $D^{OT}$ to obtain $D^{I\dagger}$ and $D^{OT\dagger}$, respectively. Then, we use argmax operation to generate the binary mask $\mathcal{M}$ for the token set. During back propagation, we utilize the STE technique Bengio et al. (2013) to directly estimate the gradient of $D^I$ and $D^{OT}$ through the binary mask $\mathcal{M}$, bypassing the non-differentiable argmax operation to avoid the gradient flow problem.

Table 7: Effect on the MaskedSoftmax operation.

| Method | POPE | VQAv2 | GQA |
|---|---|---|---|
| LLaVA-1.5-7B | 85.9 | 78.5 | 62.0 |
| Dynamic-LLaVA-7B$_{I|T}$ (Ours) | 85.9 | 77.8 | 61.3 |
| Dynamic-LLaVA-7B$_{I|T}$ w/o MaskedSoftmax | 84.5 | 76.7 | 59.8 |

As shown in the bottom figure of Fig. 3, we display the pipeline of predictors during training. Simply put, during the forward propagation of training, we relax the decision matrix generated by the predictors. In the backward propagation, we employ the Straight-Through Estimator (STE) (Bengio et al., 2013) technique to circumvent the gradient problem, thus enabling end-to-end training of the predictors.

Furthermore, as shown in Tab. 7, we analyzed the rationale for using the MaskedSoftmax operation instead of directly employing selective approaches during training. It is evident that directly setting the values of unnecessary tokens to zero vectors leads to a significant performance degradation, as observed in VQAv2, GQA, and POPE, where there was a performance loss of over 1%.

Table 8: Benchmark statistics of average token length (rows 2-5) and the total number of tokens that participated in computation during inference (the last 3 rows).

| Dataset | VQAv2 | VizWiz | SciQA | LVIS-VQA (single) | LVIS-VQA (multi) | ShareGPT4V-VQA (single) |
|---|---|---|---|---|---|---|
| Avg. image token length | 576 | 576 | 576 | 576 | 576 | 576 |
| Avg. text token length | 8 | 10 | 15 | 56 | 205 | 51 |
| Avg. output text token length | 2 | 3 | 6 | 159 | 351 | 1555 |
| Avg. token length | 586 | 589 | 597 | 794 | 1132 | 2182 |
| LLaVA-1.5-13B | 586 | 589 | 597 | 794 | 1132 | 2182 |
| LLaVA-13B-FastV$_{k=3,r=0.75}$ | 154(-74%) | 157 (-73%) | 165 (-72%) | 359 (-55%) | 700 (-38%) | 1750 (-20%) |
| Dynamic-LLaVA-13B$_{I|T}$ | 125 (-79%) | 128 (-78%) | 136 (-77%) | 255 (-68%) | 501 (-56%) | 929 (-57%) |

## A.3 MORE DETAILED DISCUSSION

### A.3.1 INFERENCE EFFICIENCY OF DYNAMIC-LLAVA AND TOKEN REDUCTION METHODS

To further quantify the improvements of inference efficiency that Dynamic-LLaVA brings by the output text token lengthens, we have presented statistics on the token length for three vision understanding benchmarks and generation ability benchmarks in Table 8. Additionally, we compare the total token lengths of Dynamic-LLaVA and FastV, including both the prefill and decoding stages.

The results indicate that as the output length increases, Dynamic-LLaVA progressively exhibits a significant advantage in terms of token reduction percentage compared to FastV, which only reduces image tokens during the prefill stage.

It should be noted that the vision understanding benchmarks (VQAv2, VizWiz, SciQA) generally need the model responding to multiple-choice questions or providing brief answers. However, real-world scenarios often require MLLMs to provide more detailed and extensive responses, which aligns with our constructed LVIS-VQA and ShareGPT4V-VQA benchmarks. Therefore, the improvement in inference efficiency that Dynamic-LLaVA provides is particularly significant on these two generation ability benchmarks compared to previous MLLM token reduction methods (*e.g.*, FastV Chen et al. (2024a)), which have longer output text lengths. Additionally, Dynamic-LLaVA also demonstrates superior performance in terms of generation fluency and quality.

### A.3.2 DISCUSSION FOR LLM KV CACHE COMPRESSION METHODS AND DYNAMIC-LLAVA

We further discuss the core distinctions between Dynamic-LLaVA and LLM KV cache compression methods as follows.

First, considering the complete generation process of MLLMs with KV cache. The key distinction between Dynamic-LLaVA and other LLM KV cache compression methods lies in its approach. Dynamic-LLaVA implements a "online" decision-making process to determine whether to add KV activations of the current token to KV cache, rather than removing KV activations from the historical KV cache. As presented in Fig. 4, we show the differences between a commonly used LLM KV cache compression method, H2O (Zhang et al., 2023), and our method. A significant distinction is highlighted between the two methods. In the left figure, H2O computes the attention scores between the current token's query $Q$ and all past KV cache, removing useless KV cache based on their attention scores (e.g., KV cache corresponding to tokens with an attention score of 0.05). While our method (in the right figure) does not decide how to retain historical KV cache. Instead, it calculates KV activations (by $W_k$ and $W_v$) for the current token and applies an output predictor (on the current token's embedding) to decide whether to add the current token's corresponding KV activations into KV cache (as shown in the "Yes" branch) or not to add them (as shown in the "No" branch).

Second, Dynamic-LLaVA is an MLLM inference acceleration framework that considers the distinct properties of different modalities and incorporates tailored sparsification strategies accordingly. We have implemented the H2O method in conjunction with LLaVA, and the results are presented in Tab. 1 and Tab. 3. We configured the hyperparameters of H2O to retain 50% KV cache in each of the prefill and decoding stages. However, H2O, performs poorly in multimodal scenarios involving vision and language contexts. Our analysis suggests that H2O's strategy of discarding historical KV cache based on attention scores does not adapt well to mixed-modality contexts. To get more comparable results, in the vision understanding benchmarks of Tab. 1, we modified the layer configuration for H2O's KV cache compression to enhance the performance. The first 10 layers do not conduct KV cache compression, and H2O is applied only beyond the 10-th layer. In the generation ability tasks presented

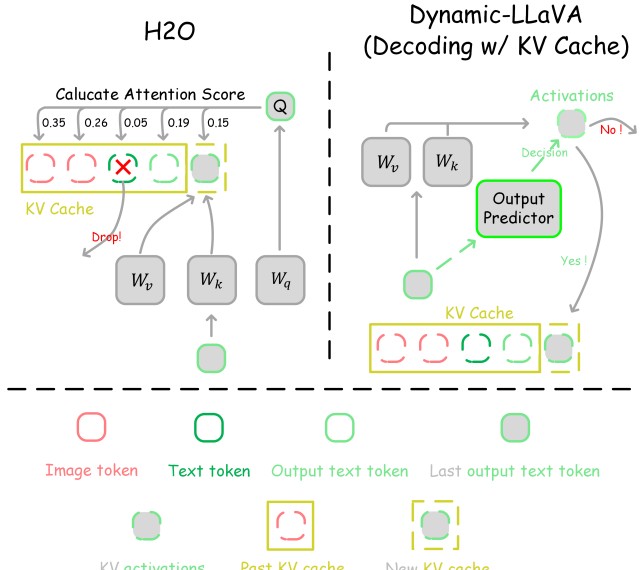

Figure 4: KV cache compression pipeline (H2O (Zhang et al., 2023) vs. Dynamic-LLaVA (when decoding with KV cache). **Left Figure:** the KV cache compression pipeline of H2O involves calculating the attention score between the current $Q$ and past KV cache during the decoding stage. The KV activations corresponding to the minimal attention score is subsequently dropped from historical KV cache. **Right Figure:** The workflow of Dynamic-LLaVA when decoding with KV cache. Our approach evaluates each current token's features by an output predictor to determine **whether its activations which through $W_K$ and $W_V$ should be added to the KV cache**.

in Tab. 3, we combined H2O with FastV (Chen et al., 2024a). The enhanced H2O implementation on MLLM shows some performance improvements. However, its performance still falls short compared to Dynamic-LLaVA.

Third, Dynamic-LLaVA introduces a tailored sparsification inference scheme specific to various inference modes. In the above discussion, we have extensively discussed the core design of our method and its distinctions from other approaches in the context of decoding with KV cache. However, it is important to emphasize that we have designed efficient inference methods tailored for different scenarios, i.e., prefill, decoding with KV cache, and decoding without KV cache. Decoding with KV cache can be seen as "online KV cache compression". For the scenarios of prefill and decoding without KV cache, although KV cache activations are not involved, Dynamic-LLaVA can still substantially enhance the computational efficiency of MLLMs. This enhancement is an advantage that traditional KV cache compression methods do not provide, demonstrating the broader applicability and effectiveness of our method across various inference modes within MLLMs.

Moreover, enhancing computational efficiency during the prefill and decoding without KV cache stages is equally critical for MLLMs Liu et al. (2024b); Li et al. (2024d); Chen et al. (2024a); Huang et al. (2024a); Cha et al. (2024); Leviathan et al. (2023); Chen et al. (2023a); Liu et al. (2023a).

There is a positive correlation between the vision understanding performance of MLLMs and image resolution Liu et al. (2024b); Li et al. (2024d). While achieving improved performance, the increased number of image tokens significantly adds to the computation burden during the MLLM's prefill stage, which includes a rise in computation budgets such as longer image token sequences and increased inference latency Chen et al. (2024a); Huang et al. (2024a); Cha et al. (2024). This challenge echoes the importance of structural information compression in graph representation learning, where methods like GraRep Cao et al. (2015) reduce complexity via low-dimensional embeddings. Similar graph-based approaches, such as compressing image tokens via graph sparsity Jiang et al. (2025), further enable efficient MLLM acceleration. Therefore, reducing the number of image tokens to accelerate MLLMs is crucial. However, existing LLM KV cache compression methods typically do not accelerate the computation in the MLLM prefill stage, thus limiting their applicability in MLLMs.

Table 9: Results of Dynamic-LLaVA with text context sparsification during the prefill stage on 3 vision understanding benchmarks.

| Method | POPE | VQAv2 | GQA |
|---|---|---|---|
| LLaVA-1.5-7B | 85.9 | 78.5 | 62.0 |
| Dynamic-LLaVA-7B$_{I|T}$ (Ours) | 85.9 | 77.8 | 61.3 |
| Dynamic-LLaVA-7B$_{I|T}$ +30% text context sparsification | 85.1 | 75.3 | 60.2 |

For decoding without KV cache, this inference mode still retains advantages over those using KV cache Vaswani (2017). For instance, decoding without KV cache does not require the storage of extensive KV activations for subsequent decoding use, which significantly reduces the memory overhead during the MLLM inference process. Accelerating the inference speed of decoding without KV cache also has potential applications in commonly used speculative sampling strategies Leviathan et al. (2023); Chen et al. (2023a); Liu et al. (2023a). These strategies utilize preliminary decoding by smaller LLMs to parallelize the autoregressive decoding of larger LLMs, also enhancing the efficiency of MLLMs in practical deployments. Our Dynamic-LLaVA facilitates this by reducing the number of tokens processed in parallel during decoding, thereby accelerating the parallel decoding of large MLLMs and further improving the inference benefits of speculative sampling strategies.

### A.3.3 ADDITIONAL DISCUSSION FOR SPARSIFYING TEXT CONTEXT DURING PREFILL

In the current design of Dynamic-LLaVA, we only apply sparsification to the image tokens during the prefill stage. Naturally, it is also feasible to perform sparsification on text tokens during the prefill stage. Therefore, we also conduct experiments with sparsifying text tokens during the prefill stage, reducing 30% text tokens. The results on 3 vision understanding benchmarks are presented in Tab. 9. Unfortunately, sparsifying text tokens during the prefill stage resulted in a noticeable performance degradation. Across the three benchmarks presented in the table, the performance dropped by an average of 1.45% compared to sparsifying only the image tokens during the prefill stage. These results imply that text tokens during the prefill stage are crucial in the current multimodal scenarios, necessitating a more refined sparsification design. This will also serve as an important direction for our future works.

### A.4 IMPLEMENTATION DETAILS

#### A.4.1 DETAILS OF PREDICTOR ARCHITECTURE

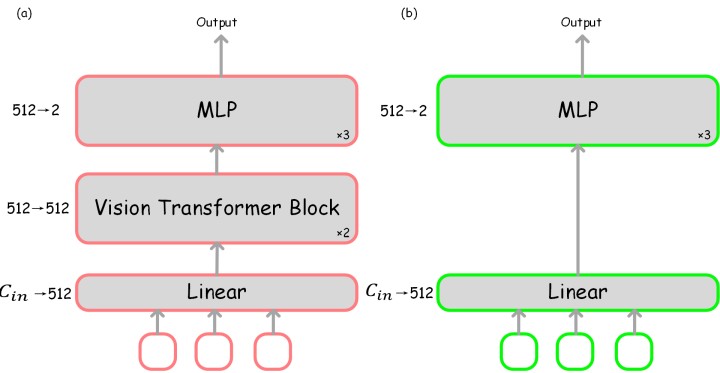

Figure 5: Overviews of the image predictor (a) and the output text predictor (b).

The architectues of the image predictor and the output text predictor are presented in Fig. 5. Both the image predictor and the output text predictor employ a linear layer to reduce the input dimension from MLLMs to 512, thereby decreasing computational demands. The image predictor utilizes two vision transformer blocks (Dosovitskiy, 2020; Touvron et al., 2021) and a dimension-reducing three-layer MLP ($512 \rightarrow 256 \rightarrow 128 \rightarrow 2$). The structure of the output text predictor is similar to that of the image predictor, except it does not use vision transformer blocks. This design choice ensures that

Table 10: The detailed calculation formula for FLOPs.

| Model Name | Calculation |
|---|---|
| LLaVA-1.5-7B | $32 \times (32 \times 1 \times 576 \times 4096 \times 4096 \times 1 + 4 \times 1 \times 576 \times 4096 \times 4096) = 10.1T$ |
| LLaVA-PruMerge+ | $32 \times (32 \times 1 \times 146 \times 4096 \times 4096 \times 1 + 4 \times 1 \times 146 \times 146 \times 4096) = 2.5T$ |
| LLaVA-FastV | $3 \times (32 \times 1 \times 576 \times 4096 \times 4096 \times 1 + 4 \times 1 \times 576 \times 576 \times 4096) + 29 \times (32 \times 1 \times 144 \times 4096 \times 4096 + 4 \times 1 \times 144 \times 144 \times 4096) = 3.2T$ |
| VoCo-LLAMA | $32 \times (32 \times 1 \times 128 \times 4096 \times 4096 \times 1 + 4 \times 1 \times 128 \times 128 \times 4096) = 2.2T$ |
| LLaVA-HiRED | $32 \times (32 \times 1 \times 115 \times 4096 \times 4096 \times 1 + 4 \times 1 \times 115 \times 115 \times 4096) = 2.07T$ |
| Dynamic-LLaVA | $2 \times (32 \times 1 \times 576 \times 4096 \times 4096 + 4 \times 1 \times 576 \times 576 \times 4096) + 30 \times (32 \times 1 \times 115 \times 4096 \times 4096 + 4 \times 1 \times 115 \times 115 \times 4096) = 2.5T$ |
| LLaVA-1.5-13B | $40 \times (32 \times 1 \times 576 \times 5120 \times 5120 + 4 \times 1 \times 576 \times 576 \times 5120) = 19.6T$ |
| LLaVA-PruMerge+ | $40 \times (32 \times 1 \times 146 \times 5120 \times 5120 + 4 \times 1 \times 146 \times 146 \times 5120) = 4.9T$ |
| LLaVA-FastV | $3 \times (32 \times 1 \times 576 \times 5120 \times 5120 + 4 \times 1 \times 576 \times 576 \times 5120) + 37 \times (32 \times 1 \times 144 \times 5120 \times 5120 + 4 \times 1 \times 144 \times 144 \times 5120) = 6.0T$ |
| VoCo-LLaMA | $40 \times (32 \times 1 \times 128 \times 5120 \times 5120 + 4 \times 1 \times 128 \times 128 \times 5120) = 4.3T$ |
| LLaVA-HiRED | $40 \times (32 \times 1 \times 115 \times 5120 \times 5120 + 4 \times 1 \times 115 \times 115 \times 5120) = 3.9T$ |
| Dynamic-LLaVA | $2 \times (32 \times 1 \times 576 \times 5120 \times 5120 + 4 \times 1 \times 576 \times 576 \times 5120) + 38 \times (32 \times 1 \times 115 \times 5120 \times 5120 + 4 \times 1 \times 115 \times 115 \times 5120) = 4.7T$ |
| LLaVA-TokenPacker-7B-144Token | $32 \times (32 \times 1 \times 144 \times 4096 \times 4096 + 4 \times 1 \times 144 \times 144 \times 4096) = 2.5T$ |
| LLaVA-TokenPacker-7B-64Token | $32 \times (32 \times 1 \times 64 \times 4096 \times 4096 + 4 \times 1 \times 64 \times 64 \times 4096) = 1.1T$ |
| LLaVA-TokenPacker-FastV-7B | $3 \times (32 \times 1 \times 144 \times 4096 \times 4096 + 4 \times 1 \times 144 \times 144 \times 4096) + 29 \times (32 \times 1 \times 72 \times 4096 \times 4096 + 4 \times 1 \times 72 \times 72 \times 4096) = 1.4T$ |
| Dynamic-LLaVA-TokenPacker-7B | $2 \times (32 \times 1 \times 144 \times 4096 \times 4096 + 4 \times 1 \times 144 \times 144 \times 4096) + 30 \times (32 \times 1 \times 57 \times 4096 \times 4096 + 4 \times 1 \times 57 \times 57 \times 4096) = 1.1T$ |
| LLaVA-TokenPacker-13B-144Token | $40 \times (32 \times 1 \times 144 \times 5120 \times 5120 + 4 \times 1 \times 144 \times 144 \times 5120) = 4.9T$ |
| LLaVA-TokenPacker-13B-64Token | $40 \times (32 \times 1 \times 64 \times 5120 \times 5120 + 4 \times 1 \times 64 \times 64 \times 5120) = 2.2T$ |
| LLaVA-TokenPacker-FastV-13B | $3 \times (32 \times 1 \times 144 \times 5120 \times 5120 + 4 \times 1 \times 144 \times 144 \times 5120) + 37 \times (32 \times 1 \times 72 \times 5120 \times 5120 + 4 \times 1 \times 72 \times 72 \times 5120) = 2.6T$ |
| Dynamic-LLaVA-TokenPacker-13B | $2 \times (32 \times 1 \times 144 \times 5120 \times 5120 + 4 \times 1 \times 144 \times 144 \times 5120) + 38 \times (32 \times 1 \times 57 \times 5120 \times 5120 + 4 \times 1 \times 57 \times 57 \times 5120) = 2.1T$ |

Table 11: Comparison with more SoTA vision context sparsification methods on vision understanding benchmarks.

| Method | Free | Image (prefill) | | VQAv2 | GQA | SciQA | TextVQA | POPE | MME | MMBench | SEED | MMVP | RealWorldQA | CVBench-2D |
|---|---|---|---|---|---|---|---|---|---|---|---|---|---|---|
| | | Token | TFLOPs | | | | | | | | | | | |
| LLaVA-1.5-7B | – | 576 | 10.1 | 78.5 | 62.0 | 66.8 | 58.2 | 85.9 | 1510.7 | 64.3 | 66.1 | 29.3 | 53.7 | 58.5 |
| (Arxiv24) LLaVA-PruMerge+ | ✗ | 146 (-75%) | 2.5 (-75%) | 76.8 (-1.7) | – | 68.3 (+1.5) | 57.1 (-1.1) | 84.0 (-1.9) | 1462.4 (-48.3) | 64.9 (+0.6) | – | – | – | – |
| (ECCV24) LLaVA-FastV$_{k=3,r=0.75}$ | ✓ | 144 (-75%) | 3.2 (-68%) | 75.1 (-3.4) | 57.5 (-4.5) | 68.7 (+1.9) | 56.2 (-2.0) | 81.0 (-4.9) | 1458.9 (-51.8) | 63.5 (-0.8) | 62.8 (-3.3) | 24.0 (-5.3) | 53.7 (-0.0) | 56.7 (-1.9) |
| (Arxiv24) VoCo-LLaMA | ✗ | 128 (-78%) | 2.2 (-78%) | 76.9 (-1.6) | 59.8 (-2.2) | – | – | – | – | 61.0 (-3.3) | 59.1 (-7.0) | – | – | – |
| (ECCV24) IVTP | ✗ | – | 4.7 (-53%) | 77.8 (-0.7) | 60.4 (-1.6) | 67.8 (+1.0) | 58.2 (+0.0) | 85.7 (-0.2) | – | 66.1 (+1.8) | 54.6 (-11.5) | – | – | – |
| (Arxiv24) TRIM | ✗ | 455 (-21%) | – | 76.4 (-2.1) | 61.4 (-0.6) | 48.1 (-18.7) | 53.7 (-4.5) | 85.3 (-0.6) | 1461.3 (-49.4) | 67.4 (+3.1) | – | – | – | – |
| (Arxiv24) SparseVLM | ✓ | 449 (-22%) | – | 73.8 (-4.7) | 56.0 (-6.0) | 67.1 (+0.3) | 54.9 (-3.3) | 80.5 (-5.4) | 1696 (+185.3) | 60.0 (-4.3) | – | – | – | – |
| (Arxiv24) LLaVA-HiRED | ✓ | 115 (-80%) | 2.0 (-80%) | 74.7 (-3.8) | – | 66.4 (-0.4) | 44.2 (-14.0) | – | – | – | – | – | – | – |
| Dynamic-LLaVA-7B$_I$ (Ours) | ✗ | 115 (-80%) | 2.5 (-75%) | 78.0 (-0.5) | 61.4 (-0.6) | 69.1 (+2.3) | 57.0 (-1.2) | 85.0 (-0.9) | 1479.8 (-30.9) | 65.4 (+1.1) | 64.6 (-1.5) | – | – | – |
| Dynamic-LLaVA-7B$_{I|T}$ (Ours) | ✗ | 115 (-80%) | 2.5 (-75%) | 77.9 (-0.6) | 61.3 (-0.7) | 68.6 (+1.8) | 56.5 (-1.7) | 85.9 (+0.0) | 1501.0 (-9.7) | 64.1 (-0.2) | 65.0 (-1.1) | 26.3 (-3.0) | 57.0 (+3.3) | 58.3 (-0.2) |
| LLaVA-1.5-7b+H2O$_{r=0.5,k=10}$ | ✓ | – | 10.1 | 77.9 (-0.6) | 61.0 (-1.0) | 41.9 (-16.3) | 55.9 (-2.3) | 86.9 (+1.0) | 1458.4 (-52.3) | 1.4 (-62.9) | 26.8 (-39.3) | 0 (-29.3) | 42.3 (-11.4) | 49.7 (-8.8) |

decisions during decoding with KV cache are solely based on the features from the current output text token.

## A.4.2 DETAILS OF SETTINGS

For Dynamic-LLaVA presented in Tab. 1, we use LLaVA-1.5 (Liu et al., 2024a) as the base model and performing an additional one-epoch of instruction-tuning on its open-source weights. During training, we freeze the vision encoder and projector, updating only the parameters of LLM and predictors. The initial learning rates for LLM and predictors are set at 5e-6 and 2e-4, respectively, with a fixed global batch size of 64. We use the same 656K Mixture Dataset of LLaVA-1.5 for instruction-tuning, exclusively employing data containing images to train the predictors. All the other settings remain consistent with LLaVA-1.5.

For Dynamic-LLaVA-Tokenpacker presented in Tab. 2, we utilize the open-source weights of LLaVA-TokenPacker (Jin et al., 2024) as the base model. All settings remain identical to those described for Dynamic-LLaVA above.

We make adjustments only to the learning rates of the original MLLMs, applying uniform learning rates across all weights and the experiments are conducted on 8 NVIDIA A100 (80G).

For LLaVA-FastV$_{k=3,r=0.75}$ presented in Tab. 1 and Tab. 3, we use FastV to perform token pruning starting from the 3-rd layer of the model, applying FastV to tokens only during the prefill stage.

For $^\dagger$LLaVA-FastV$_{k=3,r=0.75}$ presented in Tab. 1, we further train LLaVA-FastV$_{k=3,r=0.75}$ for 1 epoch based on the open-source codes. The learning rate and batch size are same as Dynamic-LLaVA, and the discarded tokens are set to 0.

Table 12: More comparison with SoTA vision context sparsification methods on vision understanding benchmarks.

| Method | LLM | Image (prefill) | | | VizWiz | LLaVA-Wild | MM-Vet |
|---|---|---|---|---|---|---|---|
| | | Res. | Token | TFLOPs | | | |
| LLaVA-1.5-7B | Vicuna-7B | 336 | 576 | 10.1 | 50.0 | 65.8 | 31.7 |
| (Arxiv24) LLaVA-PruMerge+ | Vicuna-7B | 336 | 146 (-75%) | 2.5 (-75%) | – | – | – |
| (ECCV24) LLaVA-FastV$_{k=3,r=0.75}$ | Vicuna-7B | 336 | 144 (-75%) | 3.2 (-68%) | 51.9 (+1.9) | 66.7 (+0.9) | 28.1 (-3.6) |
| (Arxiv24) VoCo-LLaMA | Vicuna-7B | 336 | 128 (-78%) | 2.2 (-78%) | – | – | – |
| (Arxiv24) LLaVA-HiRED | Vicuna-7B | 336 | 115 (-80%) | 2.0 (-80%) | – | – | – |
| Dynamic-LLaVA-7B$_I$ (Ours) | Vicuna-7B | 336 | 115 (-80%) | 2.5+0.01 (-75%) | 50.2 (+0.2) | 67.3 (+1.5) | 29.5 (-2.2) |
| Dynamic-LLaVA-7B$_{I|T}$ (Ours) | Vicuna-7B | 336 | 115 (-80%) | 2.5+0.01 (-75%) | 51.2 (+1.2) | 68.7 (+2.9) | 32.2 (+0.5) |
| LLaVA-1.5-13B | Vicuna-13B | 336 | 576 | 19.6 | 53.6 | 72.5 | 36.6 |
| (Arxiv24) LLaVA-PruMerge+ | Vicuna-13B | 336 | 146 (-75%) | 4.9 (-75%) | – | – | – |
| (ECCV24) LLaVA-FastV$_{k=3,r=0.75}$ | Vicuna-13B | 336 | 144 (-75%) | 6.0 (-69%) | 54.7 (+1.1) | 74.2 (+1.7) | 34.5 (-2.1) |
| Dynamic-LLaVA-13B$_I$ (Ours) | Vicuna-13B | 336 | 115 (-80%) | 4.7+0.01 (-76%) | 53.3 (-0.3) | 73.4 (+0.9) | 37.3 (+0.7) |
| Dynamic-LLaVA-13B$_{I|T}$ (Ours) | Vicuna-13B | 336 | 115 (-80%) | 4.7+0.01 (-76%) | 53.0 (-0.6) | 70.1 (-2.5) | 34.6 (-2.0) |

Table 13: Additional results of Dynamic-LLaVA with the TokenPacker projector on generation ability benchmarks. Dynamic-LLaVA achieves significant inference efficiency during the decoding stage with negligible generation ability drop when both sparsify vision and language.

| Benchmark | Method | Image (prefill) | Text (decoding) | | | PPL↓ | METEOR↑ |
|---|---|---|---|---|---|---|---|
| | | TFLOPs | Total→Computing | TFLOPs | Mem. (M) | | |
| LVIS-VQA (single-round) | LLaVA-TokenPacker-7B-144Token | 2.5 | 159/175→159/175 | 2.75/3.03 | 102/89 | 4.60 | 0.3114 |
| | Dynamic-LLaVA-TokenPacker-7B$_I$ (Ours) | 1.1 (-56%) | 159/178→159/178 | 2.75/3.08 | 111/101 | 4.72 | 0.3127 |
| | Dynamic-LLaVA-TokenPacker-7B$_{I|T}$ (Ours) | 1.1 (-56%) | 159/182→83/90 | 1.51/1.59 | 68 (-33%)/54 (-39%) | 4.91 | 0.3120 |
| | LLaVA-TokenPacker-13B-144Token | 4.9 | 159/176→159/176 | 5.36/5.94 | 148/142 | 4.40 | 0.3166 |
| | Dynamic-LLaVA-TokenPacker-13B$_I$ (Ours) | 2.1 (-57%) | 159/178→159/178 | 5.36/6.03 | 152/149 | 4.55 | 0.3149 |
| | Dynamic-LLaVA-TokenPacker-13B$_{I|T}$ (Ours) | 2.1 (-57%) | 159/178→84/88 | 2.95/3.08 | 87 (-41%)/73 (-49%) | 4.78 | 0.3131 |
| LVIS-VQA (multi-round) | LLaVA-TokenPacker-7B-144Token | 2.5 | 351/499→351/499 | 6.12/8.76 | 222/257 | 2.97 | 0.4220 |
| | Dynamic-LLaVA-TokenPacker-7B$_I$ (Ours) | 1.1 (-56%) | 351/507→351/507 | 6.12/8.90 | 245/270 | 3.09 | 0.4223 |
| | Dynamic-LLaVA-TokenPacker-7B$_{I|T}$ (Ours) | 1.1 (-56%) | 351/519→181/258 | 3.32/4.48 | 142 (-36%)/145 (-44%) | 3.20 | 0.4245 |
| | LLaVA-TokenPacker-13B-144Token | 4.9 | 351/514→351/514 | 11.92/17.57 | 321/409 | 2.89 | 0.4232 |
| | Dynamic-LLaVA-TokenPacker-13B$_I$ (Ours) | 2.1 (-57%) | 351/514→351/514 | 11.92/17.90 | 346/426 | 2.97 | 0.4222 |
| | Dynamic-LLaVA-TokenPacker-13B$_{I|T}$ (Ours) | 2.1 (-57%) | 351/517→179/253 | 6.33/8.54 | 189(-41%)/207 (-49%) | 3.07 | 0.4230 |

For H2O in vision understanding benchmarks presented in Tab. 1, LLaVA-1.5-7/13B+H2O$_{r=0.5}$ in Tab. 1 means that we directly implement H2O on LLaVA, which retains 50% KV cache in each of the prefill and decoding stages. And LLaVA-1.5-7/13B+H2O$_{k=10,r=0.5}$ means that we still retain 50% KV cache in each of the prefill and decoding stages, but the first 10 layers do not conduct KV cache compression, and H2O is applied only beyond the 10-th layer.

For H2O in generation ability benchmarks presented in Tab. 3. H2O$_{r=0.5}$ means that we directly implement H2O on LLaVA, which retains 50% KV cache in each of the prefill and decoding stages. Fastv-7/13B$_{k=3,r=0.5}$+H2O$_{r=0.5}$ means that we use FastV to reduce tokens of the prefill stage, and use H2O to reduce KV cache (retain ratio = 50%) only for the decoding stage.

### A.4.3  DETAILS OF LVIS-VQA BENCHMARKS

We use the LVIS-Instruct4V Dataset (Wang et al., 2023) to build the generation ability benchmarks.

The LVIS-Instruct4V Dataset contains 220,000 visually aligned and context-aware instructions generated by prompting the advanced GPT-4V model with images sourced from the LVIS database. We created the LVIS-VQA (single-round) Benchmark by selecting 1,000 instances from the LVIS-Instruct4V dataset, specifically focusing on single-round Visual Question Answering (VQA) scenarios where the answer lengths exceed 100 words. Additionally, we constructed the LVIS-VQA (multi-round) Benchmark from another subset of 1,000 multi-round VQA instances. The average answer length in this benchmark exceeds 300 words, with interactions spanning more than seven rounds. Examples from the LVIS-VQA benchmark we constructed can be found in Fig. 6 and Fig. 7.

### A.4.4  DETAILS OF SHAREGPT4V-VQA BENCHMARKS

To evaluate the generation ability of the model in long-text context scenarios, we use the ShareGPT4V dataset (Chen et al., 2023b), extracting VQA samples with long-text caption to construct our ShareGPT4V-VQA benchmark. The original ShareGPT4V dataset consists of 100K VQA samples, with an average caption length of 900 characters. To construct the long generation text benchmark

Table 14: The latency of the model during the entire generation stage with KV cache. For decoding, we report the average latency per token when batch size = 1 and the number of total generation tokens = 1000. The results are measured in one A100 (80G).

| Method | Prefill latency (ms) | Decoding latency (ms) |
|---|---|---|
| LLaVA-1.5-13B | 124.52 | 28.42 |
| †LLaVA-1.5-13B+H2O$_{r=0.5}$ | 111.84 | 37.71 |
| Dynamic-LLaVA-13B$_{I|T}$ | 72.51 | 26.85 |

† H2O requires an additional attention computation process to obtain the attention scores, due to the efficient inference operator implicitly handling attention operations.

Table 15: Training time of Dynamic-LLaVA on 8 A100 (80G).

| Method | Training time (h) |
|---|---|
| Dynamic-LLaVA-7B$_{I|T}$ | ∼13 |
| Dynamic-LLaVA-13B$_{I|T}$ | ∼24 |

and obtain the benchmark ShareGPT4V-VQA of the ShareGPT4V dataset (total of 178 samples), we selected the samples with captions containing no less than 300 words, while the average output text length of this benchmark is more than 1,000 tokens.

### A.4.5 DETAILS OF CALCULATION EQUATION OF FLOPS

For the prefill FLOPs presented in Tab. 1, Tab. 2 and Tab. 3, We adopt the equation below to calculate:

$$\text{FLOPs} = 32BNC^2 + 4BN^2C, \tag{13}$$

where $B$, $N$ and $C$ denote the input batch size, the number of image tokens, and the dimensionality of MLLMs (Liu et al., 2024b;a) utilizing the LLaMA architecture (Touvron et al., 2023a;b; Llama, 2024), respectively. The first term in Eq. 13 represents the computation consumption of the MHA layer, while the second term corresponds to the overhead of the MLP layer. Specific calculations are detailed in Tab. 10. Note that we only calculate the FLOPs of the image tokens for prefill stage of MLLMs.

### A.5 ADDITIONAL RESULTS

### A.5.1 ADDITIONAL BENCHMARK RESULTS

**Vision understanding.** Due to constraints on the length of the main text, we have included the remaining experimental results for the three vision understanding benchmarks in Tab. 12. Dynamic-LLaVA achieved competitive results on these three benchmarks. In comparison with other methods that only sparsify vision tokens, Dynamic-LLaVA is capable of significantly reducing the number of tokens involved in the computation process.

We further compare the vision understanding performance with more current SoTA and recent token reduction methods, including IVTP (Huang et al., 2024a), TRIM (Song et al., 2024), and SparseVLM (Zhang et al., 2024c). The results in Tab. 11 are based on evaluations conducted with LLaVA-1.5-7B. Compared to other token reduction methods, Dynamic-LLaVA$_{I|T}$ not only reduces 80% of image tokens in the prefill stage but also decreases 50% of output text tokens during decoding, consistently outperforming both training-free and training-required methods in most cases.

**Generation ability of Dynamic-LLaVA with TokenPacker projector.** To further demonstrate the comprehensiveness of our proposed method, we present the results of Dynamic-LLaVA using TokenPacker as the vision projector on the LVIS-VQA benchmarks in Tab. 13. As shown in Tab. 13, compared to LLaVA-TokenPacker, Dynamic-LLaVA that sparsifies both vision and language contexts, significantly reduces computational costs with only a slight compromise in performance. Specifically, Dynamic-LLaVA-TokenPacker-7B/13B$_{I|T}$ can reduce up to ∼50% TFLOPs and 37.75% GPU memory overhead on average, while the PPL increases by a maximum of 0.38 and on average only by 0.275; METEOR decreases by a maximum of 0.0035, and in most cases, the results are improved.

Table 16: More trade-off results of the rates of vision context sparsification and language context sparsification.

| Context | Rate | Benchmark | | | | | |
|---|---|---|---|---|---|---|---|
| | | SciQA | TextVQA | POPE | MME | MMBench | SEED |
| Vision | **20%** | 68.6 | 56.5 | 85.9 | 1501.0 | 64.1 | 65.0 |
| | 50% | 68.8 | 57.2 | 85.5 | 1487.0 | 65.7 | 65.5 |
| | 80% | 69.7 | 57.3 | 86.1 | 1469.3 | 65.9 | 65.9 |
| Language | 20% | 68.7 | 54.9 | 86.3 | 1483.3 | 64.6 | 64.5 |
| | **50%** | 68.6 | 56.5 | 85.9 | 1501.0 | 64.1 | 65.0 |
| | 80% | 69.4 | 56.9 | 86.6 | 1473.9 | 64.8 | 64.4 |
| baseline | 100% | 66.8 | 58.2 | 85.9 | 1510.7 | 64.3 | 66.1 |

Table 17: Trade-off of the rates of vision context sparsification and language context sparsification on generation ability benchmarks. Baseline indicates LLaVA-1.5-7B. Note that the definition of "Total→Computing", "TFLOPs", "Mem. (M)" and "a/b" are same as Tab. 3.

| Context | Rate | LVIS (single-round) | | | | | LVIS (multi-round) | | | | |
|---|---|---|---|---|---|---|---|---|---|---|---|
| | | Total→Computing | TFLOPs | Mem. (M) | PPL↓ | MET.↑ | Total→Computing | TFLOPs | Mem. (M) | PPL↓ | MET.↑ |
| Vision | **20%** | 159/181→84/90 | 1.52/1.57 | 63/46 | 4.90 | 0.3108 | 351/522→182/260 | 3.33/4.38 | 144/140 | 3.17 | 0.4251 |
| | 50% | 159/173→83/86 | 1.52/1.53 | 65/202 | 4.88 | 0.3107 | 351/478→178/234 | 3.27/4.34 | 138/125 | 3.15 | 0.4264 |
| | 80% | 159/171→83/85 | 1.51/1.56 | 66/284 | 4.88 | 0.3090 | 351/480→179/238 | 3.29/4.40 | 139/124 | 3.16 | 0.4258 |
| Language | 20% | 159/170→41/39 | 0.84/0.82 | 44/22 | 5.53 | 0.2592 | 351/487→84/108 | 1.74/2.28 | 95/69 | 3.47 | 0.4236 |
| | **50%** | 159/181→84/90 | 1.52/1.57 | 63/46 | 4.90 | 0.3108 | 351/522→182/260 | 3.33/4.38 | 144/140 | 3.17 | 0.4251 |
| | 80% | 159/172→129/138 | 2.26/2.43 | 85/70 | 4.76 | 0.3116 | 351/484→281/387 | 4.98/6.86 | 190/203 | 3.10 | 0.4226 |
| baseline | 100% | 159/173→159/173 | 2.75/2.99 | 103/91 | 4.59 | 0.3103 | 351/461→351/461 | 6.12/8.08 | 222/236 | 2.97 | 0.4227 |

### A.5.2 ADDITIONAL PRACTICAL INFERENCE EFFICIENCY ANALYSIS

We further report the practical inference latency with KV cache in Tab. 14. We measure the average generation latency per token when the generation length is 1000. The proposed Dynamic-LLaVA framework exhibits an average improvement of 1.57 ms per token in generation latency compared to LLaVA when decoding with KV cache. This result demonstrates that, the learnable lightweight predictors add a negligible increase to inference latency (less than 1%). Meanwhile, for traditional "attention-based" KV cache compression methods (e.g., H2O), the requirement for attention scores during decoding to perform KV cache compression poses a challenge in practical engineering implementations. In many cases, the attention operations of efficient inference operators are implicit, thus requiring an additional computation step to obtain attention scores during inference. This can impact inference speed, especially when dealing with excessively long KV caches. In contrast, Dynamic-LLaVA relies solely on the features of the current token for prediction when decoding with KV cache, thereby avoiding this issue.

### A.5.3 TRAINING TIME

Same as LLaVA-PruMerge (Shang et al., 2024), Dynamic-LLaVA requires one epoch instruction-tuning based on pretrained LLaVA-1.5. We report the training time of Dynamic-LLaVA in Tab. 15, our training time is similar to the original LLaVA-1.5. Notably, Dynamic-LLaVA achieves efficient inference with superior performance compared to other token sparsification methods that require training (Shang et al., 2024; Song et al., 2024; Huang et al., 2024a; Ye et al., 2024).

### A.5.4 ADDITIONAL TRADE-OFF ANALYSIS

**Trade-off analysis on vision understanding tasks.** We further illustrate in Tab. 16 the performance across more vision understanding tasks when adjusting the keep rates of vision context ($r^I$) and language context ($r^{OT}$) during training. Across most datasets, setting higher keep rates leads to better results but also entails greater computational expense. Under the current settings, i.e., $r^I = 20\%, r^{OT} = 50\%$, Dynamic-LLaVA can achieve a balance between computational costs and performance. Additionally, we did not observe sharp fluctuations in performance when adjusting $r^I$ and $r^{OT}$, suggesting that Dynamic-LLaVA has robustness to the keep rates variations.

**Trade-off analysis of generation ability.** Same as vision understanding tasks, we also analysis the trade-off of $r^I$ and $r^{OT}$ on LVIS-VQA (single-round) and LVIS-VQA (multi-round).As shown in

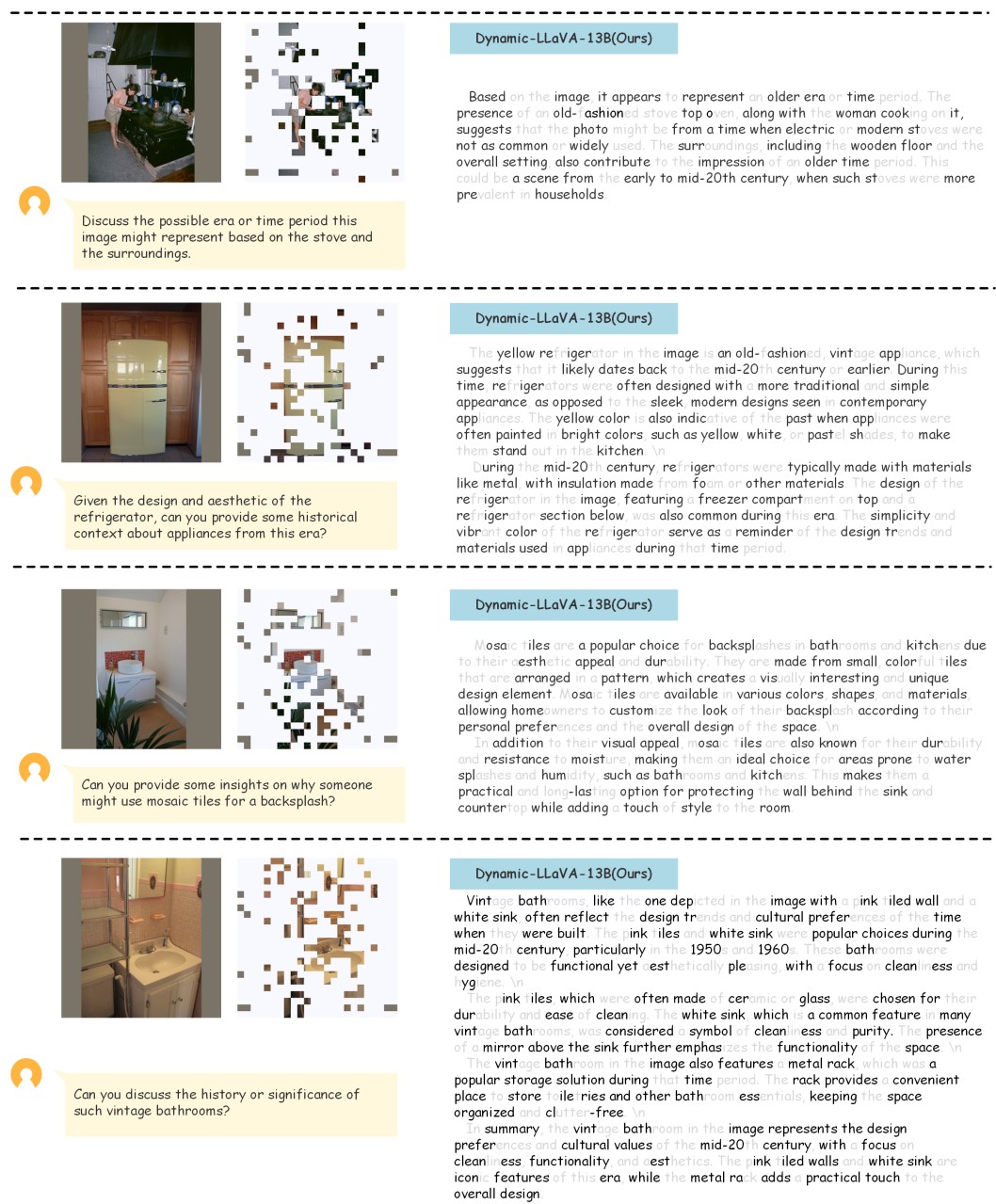

Figure 6: Visual representation of dynamic token reduction for LVIS-VQA (single-round). The gray color means the contexts reduced by Dynamic-LLaVA-13B$_{I|T}$. **Note that the reduction of the language context does not imply the texts are not generated. Rather, it refers to the subsequent computational exclusion of these output text tokens to improve inference efficiency.** Dynamic-LLaVA-13B$_{I|T}$ is able to reduce the vision and language contexts that are not crucial for generating the next token.

Table 17, when adjusting $r^I$, the performance and computational costs of Dynamic-LLaVA remain consistent; whereas, when adjusting $r^{OT}$, a small $r^{OT}$ leads to a significant decrease in computational expenses, along with a decline in generative ability. Setting $r^{OT} = 50$ achieves a balance between performance and computation costs. Although a smaller $r^{OT}$ significantly lowers computational costs, it concurrently yields a considerable decrease in the model's generative ability. While a larger $r^{OT}$ leads to higher computational costs, it only results in a slight improvement in performance.

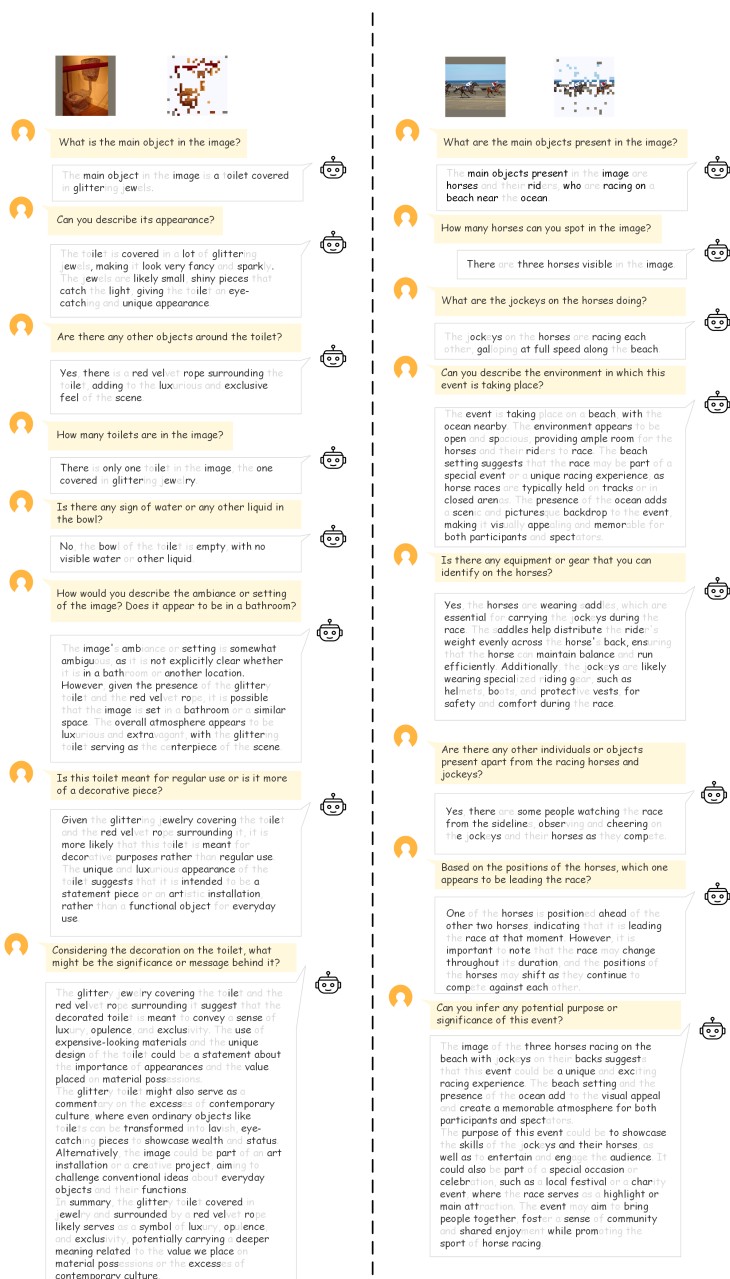

Figure 7: Visual representation of dynamic token reduction for LVIS-VQA (multi-round). The gray color means the contexts reduced by Dynamic-LLaVA-13B$_{I|T}$. Dynamic-LLaVA-13B$_{I|T}$ is able to reduce the vision and language contexts that are not crucial for generating the next token. **Note that the reduction of the language context does not imply the texts are not generated. Rather, it refers to the subsequent computational exclusion of these output text tokens to improve inference efficiency.** Dynamic-LLaVA-13B$_{I|T}$ is able to reduce the vision and language contexts that are not crucial for generating the next token.

## A.6 VISUALIZATION

**Visualization of LVIS-VQA.** Fig. 6 shows the dynamic token reduction process of Dynamic-LLaVA on LVIS-VQA (single-round), while Fig. 7 illustrates the same process in LVIS-VQA (multi-round). In Fig. 6, a user asks a question to Dynamic-LLaVA, and the associated input image is shown, with regions marked in gray indicating the dynamically reduced token patches determined by Dynamic-LLaVA. Drawing from the identified mask and the provided input information, the model produces a

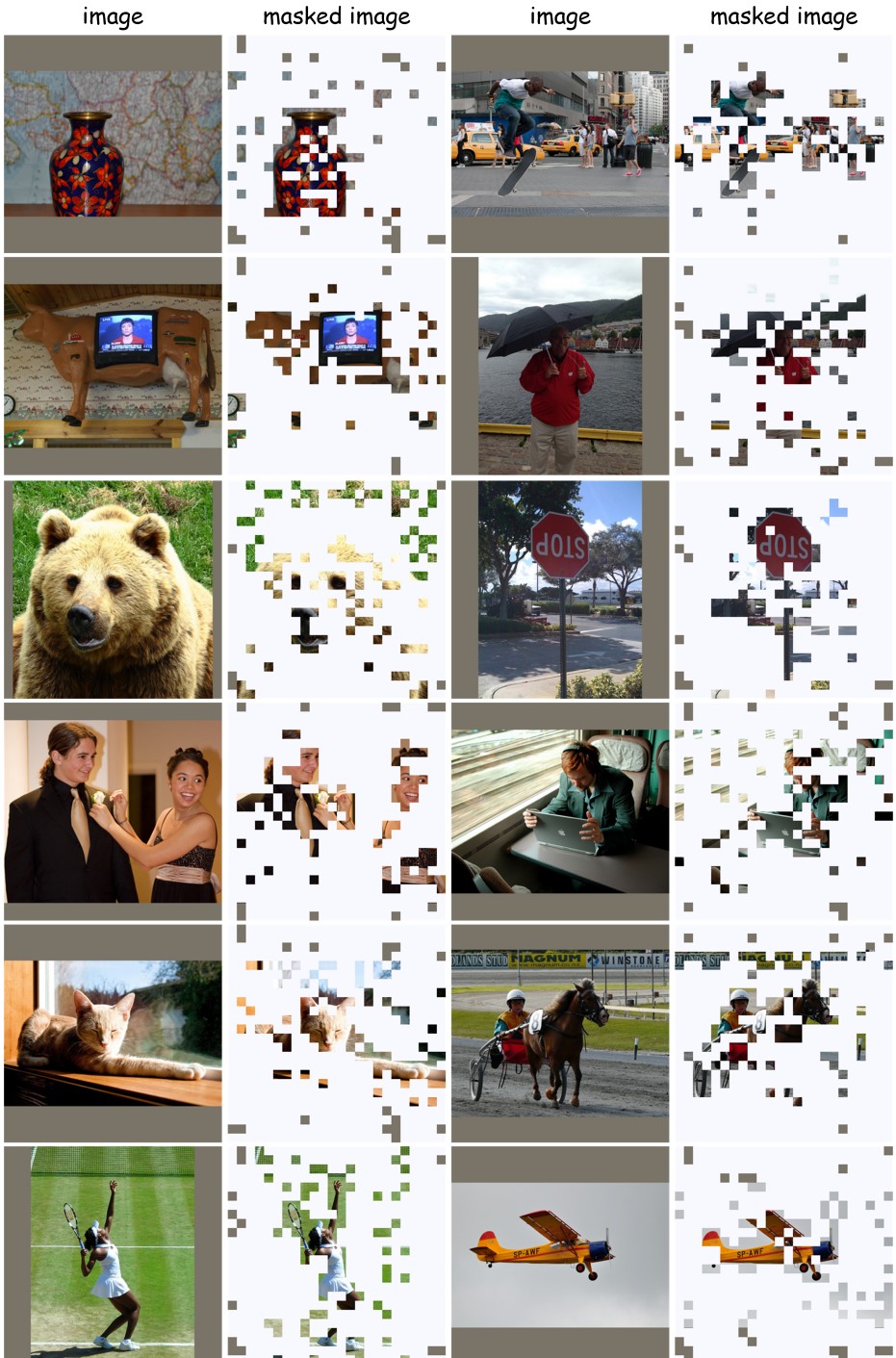

Figure 8: Visualization of the vision token patches for COCO. The gray color means the contexts reduced by Dynamic-LLaVA-13B$_{I|T}$. The first and third columns are the original images, and the second and fourth columns are the reduced vision token patches by Dynamic-LLaVA-13B$_{I|T}$.

textual mask, with the obscured words or affixes similarly exhibited in gray. This procedure boosts the inference efficiency of the model and ensures effective focus on relevant information for a fast response generation.

Fig. 7 further shows this process by illustrating the example of multi-round dialogue interaction in LVIS-VQA (multi-round). In this scenario, the model not only considers the current user query and

image but also incorporates context from previous rounds of dialogues. The dynamically masked areas in both the image and text are both shown in gray, emphasizing how the model dynamically adjusts its focus based on the ongoing interaction. This enables the model to maintain coherence and context across multiple exchanges while boosting response generation speed.

**Visualization of COCO.** In Fig. 8, we show the visualization results of the vision token patches in COCO Dataset (Lin et al., 2014). We set the instruction as "Describe this image" for Dynamic-LLaVA. Notably, the patches focus primarily on the foreground elements of the images, effectively highlighting important features and discarding irrelevant background details. This demonstrates the capability of Dynamic-LLaVA to isolate and retain essential visual features for further process.

