# OpenReview forum: "Dynamic-LLaVA: Efficient Multimodal Large Language Models via Dynamic Vision-language Context Sparsification"
_ICLR.cc/2025/Conference — ICLR 2025 Poster_

### Official Review · Reviewer_FzrR · 2024-10-15

**Soundness:** 2
**Presentation:** 3
**Contribution:** 2
**Rating:** 6
**Confidence:** 2

**Summary:**

The paper introduces Dynamic-LLaVA, a dynamic vision-language context sparsification framework aimed at enhancing the inference efficiency of mllms. The main challenge addressed is the increasing computational and memory overhead during inference, particularly as output tokens are generated, which diminishes efficiency benefits of prior vision context reduction methods. Dynamic-LLaVA reduces redundancy in both the vision and language contexts during the prefill and decoding stages, with a specialized sparsification strategy tailored for different inference modes, including decoding with and without KV cache. Experimental results demonstrate that Dynamic-LLaVA significantly reduces computational consumption—by about 75% during the prefill stage and 50% during decoding without KV cache—and saves approximately 50% of GPU memory with KV cache, all while maintaining or even improving model performance. This framework is the first to attempt simultaneous sparsification of both vision and language contexts, achieving efficient inference with negligible performance degradation.

**Strengths:**

1. As an article focusing on efficiency, it grasps the key point very well: solving efficiency problems from the perspective of efficiency. From the numerical results, Dynamic-LLaVA performs well in reducing computation and memory consumption, especially in the pre-filling and decoding stages, reducing the computation by about 75% and the GPU memory overhead by about 50%.

2. The **SPARSIFICATION INFERENCE** part achieves dynamic sparsification of visual and language context by introducing learned predictors, effectively reducing the computational and memory burden in different reasoning modes, making the entire generation process more efficient. I think this is a highlight. Sparsification of tokens during test time is a very useful method, and I personally believe that less is more. These sparse parts should actually improve the representation ability.

3. MaskedSoftmax is a good point to note. From my previous fine-tuning experience, since the generated $logp(y|x)$ is calculated by logits under the corresponding label $y$, the appearance of 0 value in the whole chain process is absolutely harmful. This soft labeling method is very thoughtful.

**Weaknesses:**

1. In the method section, the description of sparse reasoning and end-to-end training relies too much on symbolic expressions and lacks intuitive explanations. It could be aided by adding some simplified explanations or visual flowcharts to help a wider readership understand the complex technical details.
2. You can consider adding some evaluations. I personally feel that the current evaluation benchmarks are not comprehensive enough to explain the performance of MLLMs. And you should analyze as thoroughly as possible. You can refer to the Cambrian-1 [1] standard or try to apply the method to the Cambrian model.
3. In the results of the experimental section, the impact of sparsification on complex tasks (such as generating long texts or multi-round conversations) is not sufficiently verified.

[1] Tong, S., Brown, E., Wu, P., Woo, S., Middepogu, M., Akula, S. C., ... & Xie, S. (2024). Cambrian-1: A fully open, vision-centric exploration of multimodal llms. arXiv preprint arXiv:2406.16860.

**Questions:**

1. I am curious why it is not tested on the latest versions of llava such as llava-one-vision, llava-next, etc.
2. I don’t quite understand why Table 5 uses GQA and VQAv2 as vision evaluations. I think a more vision-centric dataset such as MMVP is needed.
3. Paper mentions that different sparsification strategies are used in the pre-filling stage and the decoding stage, but the specific differences between the two and the motivation for this design are not clear. In particular, why both visual and language context are sparsified in the decoding stage, while only visual tokens are sparsified in the pre-filling stage, what is the design consideration behind this?
4. In order to avoid unstable training when there are few samples, paper only sparses the samples whose text length exceeds a certain threshold. However, does this strategy ensure the stability of the model when facing inputs of different lengths? In actual use, will the model cause significant performance fluctuations due to changes in input length? If a model behaves unstable for certain input lengths, this may significantly limit its applicability in real-world applications.

---

> ### Author Response · Authors · 2024-11-14
> **Any additional concerns or questions**
>
> Dear Reviewer FzrR,
>
> Thank you very much for your effort on our paper and the valuable feedback!
>
> We noticed that you adjusted your scores (333->232, 5->3) before we submitted our rebuttal. We are currently preparing a detailed rebuttal and would appreciate knowing if you have any additional concerns or questions regarding our paper. Your insights are invaluable to us as we work to improve it. We are more than willing to respond to any inquiries and address any feedback. Thank you very much!
>
> Sincerely Yours,
>
> The Authors

---

> > ### Comment · Reviewer_FzrR · 2024-11-14
> >
> > I am sorry to inform you that I am not particularly familiar with the field of your paper, and during the bidding stage, I selected 'unwilling.' Therefore, I am unable to provide comprehensive suggestions. My feedback will change in accordance with the overall opinions of the other reviewers. If you address the other key issues, I believe I will give a score that aligns with acceptance.
> >
> > Best regards.

---

> ### Author Response · Authors · 2024-11-22
> **[2024/11/22 AOE] Detailed Rebuttal (1/2)**
>
> Dear Reviewer FzrR,
>
> Thank you for your valuable suggestions! We have conducted extensive additional experiments and made revisions to our paper. Detailed responses are provided in the following rebuttal.
>
> Sincerely Yours,
>
> The Authors
>
>
>
> > **W1. Writing.**
>
> We recognize that excessive notation may hinder broader reader comprehension. In the revised paper, we are incorporating more intuitive visualizations and adding simplified descriptions to improve the readability.
>
> In the revised version of our paper, we have supplemented the ''Method'' section with simplified explanations (Sec. 3.3), additional details (Appendix A.2), and visual flowcharts (Fig. 3 and Fig. 4).
>
>
>
> > **W2. Additional vision understanding benchmarks.**
>
> To more comprehensively evaluate Dynamic-LLaVA's performance in vision understanding tasks, we incorporated three vision-centric vision understanding benchmarks following the Cambrian-1 [1], including MMVP [2], RealWorldQA [3], and CVBench-2D [1]. The results are presented in the table below.
>
> |             Methods             |       MMVP        |    RealWorldQA    |    CVBench-2D     |
> | :-----------------------------: | :---------------: | :---------------: | :---------------: |
> |          LLaVA-1.5-7B           |       29.3        |       53.7        |       58.5        |
> |  LLaVA-7B-FastV$_{k=3,r=0.75}$  |    24.0 (-5.3)    |    53.7 (-0.0)    |    56.7 (-1.9)    |
> | ***Dynamic-LLaVA-7B$_{I/T}$***  | ***26.3 (-3.0)*** | ***57.0 (+3.3)*** | ***58.3 (-0.2)*** |
> |          LLaVA-1.5-13B          |       30.7        |       55.3        |       62.3        |
> | LLaVA-13B-FastV$_{k=3,r=0.75}$  |    28.0 (-2.7)    |    52.8 (-2.5)    |    58.9 (-3.4)    |
> | ***Dynamic-LLaVA-13B$_{I/T}$*** | ***31.3 (+0.6)*** | ***53.3 (-2.0)*** | ***60.7 (-1.6)*** |
>
> On these three new vision-centric Vision Understanding benchmarks, Dynamic-LLaVA not only leads in performance but also significantly reduces the inference resource cost of MLLMs.
>
> *Detailed results can be found in Tab. 1 of the revised paper.*
>
>
>
> > **W3. Additional complex generation ability benchmarks.**
>
> Thank you for your suggestion. We have added experiments on a complex generation ability benchmark, ShareGPT4V-VQA (single-round) [4], which is designed for generating long texts. The average output text length of this benchmark exceeds 1,000 tokens. *The detailed introduction of ShareGPT4V-VQA (single-round) is presented in Appendix A.4.4.*  The results are presented in the table below.
>
> |                                 | Image (profill) |       Text (decoding)       | Text (decoding) | Text (decoding) |                  |
> | :-----------------------------: | :-------------: | :-------------------------: | :-------------: | :-------------: | :--------------: |
> |                                 |     TFLOPs      | Total$\rightarrow$Computing |     TFLOPs      |    Mem. (M)     | PPL $\downarrow$ |
> |          LLaVA-1.5-7B           |      10.1       |    1555$\rightarrow$1555    |      28.7       |      1024       |       2.52       |
> | ***Dynamic-LLaVA-7B$_{I/T}$***  |   2.5 (-75%)    |    1555$\rightarrow$763     |  14.97 (-47%)   |   625 (-39%)    |       3.14       |
> |          LLaVA-1.5-13B          |      19.6       |    1555$\rightarrow$1555    |      55.40      |      1489       |       2.43       |
> | ***Dynamic-LLaVA-13B$_{I/T}$*** |   4.76 (-76%)   |    1555$\rightarrow$763     |  27.96 (-49%)   |   854 (-43%)    |       3.10       |
>
> On this generation ability benchmark with long output texts, Dynamic-LLaVA has significantly enhanced inference efficiency while having minimal impact on the text generation abilityof MLLMs.
>
>
> *Detailed results can be found in Tab. 3 of the revised paper.*
>
> > **Q1. Baseline  choice.**
>
> We primarily use LLaVA-1.5 as our main baseline, employing the same 12 benchmarks utilized in LLaVA-1.5. This choice is motivated by the fact that most works [5-11] in the field validate their results on these benchmarks for LLaVA-1.5, allowing for a fair comparison by adopting the same settings.
>
> For the latest version of LLaVA, such as llava-next [12], due to its incomplete open-source (only weights are released, without open-sourcing datasets and training codes), it is challenging for our method and other methods in the field, especially those that are training-required, to conduct experiment. We will include additional baselines in the future revision due to the limited rebuttal time.
>
> You might be interested in the results presented in our results in Tab. 2, where we validated our method on a recent Efficient MLLM LLaVA-tokenpacker [13]. This model serves as a challenging baseline since it already uses fewer image tokens to accelerate inference. The results demonstrate that our method can further enhance inference efficiency with minimal impact on performance.

---

> ### Author Response · Authors · 2024-11-22
> **[2024/11/22 AOE] Detailed Rebuttal (2/2)**
>
> > **Q2-1. Benchmark settings for Tab. 5.**
>
> We apologize for the confusion. The main purpose of Tab. 5 is to analyze the impact of different sparsification ratios on performance, which is why we selected two sub-benchmarks from the main benchmarks for experimental analysis. The choice of VQAv2 and GQA was due to their prominent positions in the main benchmarks, with VQAv2 also being a widely recognized large-scale VQA benchmark.
>
>
>
> > **Q2-2. Additional vision-centric benchmarks for Tab. 5.**
>
> Thanks for your suggestions. We have included the results for the MMVP benchmark in Tab. 5 of the revised paper. The results for MMVP in Tab. 5 are as follows.
>
>
> |                     |   Vision   | Vision | Vision | Langugage |  Language  | Language |
> | ------------------- | :--------: | :----: | :----: | :-------: | :--------: | :------: |
> | Sparsification rate | ***20%***  |  50%   |  80%   |    20%    | ***50%***  |   80%    |
> | MMVP performance    | ***26.3*** |  26.3  |  26.2  |   26.0    | ***26.3*** |   26.0   |
>
>
>
> > **Q3. Distinctions and motivation of different sparsification strategies in prefill and decoding stages.**
>
> We apologize for the confusion.
>
> Dynamic-LLaVA sparsifies both image tokens and output text tokens during inference. However, since MLLM has not yet started decoding to generate output text tokens during the prefill stage, only the number of image tokens is reduced at this stage. During the decoding stage of MLLM, since both image tokens and output text tokens are generated, we sparsify them simultaneously.
>
> Additionally, we conducted an experiment of sparsifying text tokens (instructions/prompts) during the prefill stage. *You can find the specific results in the  "comment for Q4 of reviewer UyMq."*
>
>
>
> > **Q4. Whether changes in input length contribute to unstable performance.**
>
> First, we need to clarify that during training, Dynamic-LLaVA sparsifies output text tokens that exceeding a certain length ($\text{LEN}^{OT}$) and optimizes the output predictor. This process does not affect the input text tokens (since we do not sparsify the input text tokens).
>
> We have conducted extensive experiments on a variety of benchmarks with diverse input lengths, as shown in Tab. 1, Tab. 2, and Tab. 3 of our paper. Dynamic-LLaVA maintains excellent performance across these benchmarks, ensuring that there are no significant performance fluctuations due to changes in input length.
>
> Additionally, regarding the sparsification ratio of output text tokens (which is similar to the ratio during training), the results in Tab. 3 indicate that on benchmarks with output text tokens of varying lengths, Dynamic-LLaVA still achieves an overall 50% reduction of output text tokens while maintaining stable generation ability. *Ablation studies of $\text{LEN}^{OT}$ can be found in Tab. 6 of our paper.*
>
>
>
> > **Reference**
>
> [1] Cambrian-1: A fully open, vision-centric exploration of multimodal llms. Arxiv2406.
>
> [2] Eyes wide shut? exploring the visual shortcomings of multimodal llms. CVPR2024.
>
> [3] Grok. 2024.
>
> [4] Sharegpt4v: Improving large multi-modal models with better captions. ECCV2024.
>
> [5] IVTP: Instruction-guided Visual Token Pruning for Large Vision-Language Models. ECCV2024.
>
> [6] Llava-prumerge: Adaptive token reduction for efficient large multimodal models. Arxiv2403.
>
> [7] VoCo-LLaMA: Towards Vision Compression with Large Language Models. Arxiv2406.
>
> [8] Less is More: A Simple yet Effective Token Reduction Method for Efficient Multi-modal LLMs. Arxiv2409.
>
> [9] An Image is Worth 1/2 Tokens After Layer 2: Plug-and-Play Inference Acceleration for Large Vision-Language Models. ECCV2024.
>
> [10] HiRED: Attention-Guided Token Dropping for Efficient Inference of High-Resolution Vision-Language Models in Resource-Constrained Environments. Arxiv2408.
>
> [11] SparseVLM: Visual Token Sparsification for Efficient Vision-Language Model Inference. Arxiv2410.
>
> [12] LLaVA-NeXT: Improved reasoning, OCR, and world knowledge. 2024.
>
> [13] Tokenpacker: Efficient visual projector for multimodal llm. Arxiv2407.

---

> ### Author Response · Authors · 2024-11-25
> **Any additional concerns or questions**
>
> Dear Reviewer FzrR,
>
> Thank you once again for reviewing our paper and providing valuable feedback. We have carefully considered your suggestions and made multiple revisions to enhance the clarity, depth, and contribution of the paper. Your constructive insights and feedback have played a significant role in the process of improving our paper.
>
> We sincerely hope you will continue to engage in the discussion. Should you have further questions or concerns, we are more than willing to provide additional explanations or supporting materials. Your insights are critical to refining our research and ensuring its relevance and impact.
>
> Furthermore, we hope these revisions and clarifications will encourage you to reassess your evaluation, as these updates directly address your constructive comments. If you have any additional questions or concerns, please feel free to reach out to us. We are committed to ensuring that all issues are thoroughly addressed.
>
> Sincerely Yours,
>
> The Authors

---

> > ### Comment · Reviewer_FzrR · 2024-11-25
> >
> > I think your rebuttal content is solid which tackled my most of concerns, I will change the score.

---

> > > ### Author Response · Authors · 2024-11-25
> > >
> > > Dear Reviewer FzrR,
> > >
> > > Thank you again for your invaluable time and effort on our paper. Thank you very much for approving our work!
> > >
> > > Sincerely Yours,
> > >
> > > The Authors

---

### Official Review · Reviewer_xRRX · 2024-11-02

**Soundness:** 3
**Presentation:** 3
**Contribution:** 3
**Rating:** 6
**Confidence:** 4

**Summary:**

Dynamic-LLaVA is a framework for efficient inference in multimodal large language models, dynamically reducing vision context redundancy during the prefill and decoding stages, cutting computation by ~75% in prefill, and saving ~50% computation and memory during decoding, without significant loss in performance.  It also designs the specific training and inference strategy for parallel sparsification, making their approach more efficient.

**Strengths:**

1. The article has a rigorous logical structure, with precise formulations for each method and complete notation. Readers who read carefully can clearly understand the methods explained in the text.

2. The experimental results are excellent; it appears that inference speed and computational cost are significantly reduced with minimal loss in performance.

3. The method design is ingenious; selective approaches are often challenging to train, but the paper provides an effective end-to-end training method that successfully alleviates the gradient flow problem.

**Weaknesses:**

Although the writing is logically clear and the formulas are complete, this level of detail makes it somewhat challenging for readers.  There are too many formulas here. Perhaps reduce them a bit and add more diagrams to aid understanding. Some high-level schematic diagrams could help; Figure 2 aids understanding to some extent, but it’s still not sufficient.

**Questions:**

There aren’t many issues; the method in the paper is logically clear, the experiments are comprehensive, and the performance is excellent.

---

> ### Author Response · Authors · 2024-11-22
> **[2024/11/22 AOE] Detailed Rebuttal (1/1)**
>
> Dear Reviewer xRRX,
>
> We greatly appreciate your high regard and assessment of our work, as well as your valuable suggestions regarding the presentation style of our paper! We recognize that our current writing overly relies on formulaic expressions. To address this, we have made concerted efforts to include more detailed explanations and illustrative diagrams. We have implemented the following modifications to enhance the readability of method part in our revised paper.
>
> Sincerely Yours,
>
> The Authors
>
>
>
> > **W. Specific modifications to ''Method'' section of the paper.**
>
> * We have moved the ''Parallel Sparsification Inference Optimization'' section focused on engineering optimizations to the Appendix A.1.
> * In Sec. 3.3.1, we have incorporated a more detailed explanation of the overall framework, including the specific mechanisms by which learnable predictors operate within the model and the details of the layers affected (L248-L252).
> * In Sec. 3.3.2, we further explain how subsequent layers of the model compute after token sparsification at $l$-layer during the prefill and decoding stages, to clarify the rationale behind our reduction of computational overhead (L280-283). Meanwhile, we have also detailed the processing by the model in subsequent layers under the scenario of decoding with KV cache (L296-299).
> * In Section 3.3.3, we provide an explanation of the consistency in masking forms during training and inference. Additionally, we further discuss the pipeline of the end-to-end sparsification training comprehensively in Appendix A.2 and add a more intuitive figure to facilitate understanding (Fig. 3).
> * In Appendix A.3.2, we provide discussions about the differences between Dynamic-LLaVA and other KV cache compression methods, and add Fig. 4, which visually shows the distinctions between Dynamic-LLaVA (when decoding with KV cache) and a KV cache compression method (H2O[1]).
>
>
> > **Reference**
>
> [1] H2O: Heavy-Hitter Oracle for Efficient Generative Inference of Large Language Models. NeurIPS2023.

---

> > ### Comment · Reviewer_xRRX · 2024-12-02
> >
> > I would like to thank the authors for their reply and their answers to my questions. I apprechiate that they were incorporated into the main text and I maintain my score.

---

> > > ### Author Response · Authors · 2024-12-02
> > >
> > > Dear Reviewer xRRX,
> > >
> > > Thank you again for your invaluable time and effort on our paper. Thank you very much for approving our work!
> > >
> > > Sincerely Yours,
> > >
> > > The Authors

---

> ### Author Response · Authors · 2024-11-25
> **Any additional concerns or questions**
>
> Dear Reviewer xRRX,
>
> Thank you once again for reviewing our paper and providing valuable feedback. We have carefully considered your suggestions and made multiple revisions to enhance the clarity, depth, and contribution of the paper. Your constructive insights and feedback have played a significant role in the process of improving our paper.
>
> We sincerely hope you will continue to engage in the discussion. Should you have further questions or concerns, we are more than willing to provide additional explanations or supporting materials. Your insights are critical to refining our research and ensuring its relevance and impact.
>
> Furthermore, we hope these revisions and clarifications will encourage you to reassess your evaluation, as these updates directly address your constructive comments. If you have any additional questions or concerns, please feel free to reach out to us. We are committed to ensuring that all issues are thoroughly addressed.
>
> Sincerely Yours,
>
> The Authors

---

### Official Review · Reviewer_UyMq · 2024-11-04

**Soundness:** 3
**Presentation:** 2
**Contribution:** 3
**Rating:** 6
**Confidence:** 5

**Summary:**

This paper proposes a novel method, Dynamic-LLaVA, which dynamically sparsifies visual context during the prefill stage and vision-text context during the decoding stage of Visual Language Model inference. Specifically, two learnable predictors are designed to identify important visual and language tokens. Additionally, masked softmax is applied during training, and Gumbel-softmax with a Straight-Through Gradient Estimator is used to address gradient flow issues. Furthermore, a batch-parallel strategy is introduced to enable efficient sparsification during inference in the Dynamic-LLaVA framework. Empirical experiments demonstrate that this method can compress approximately 75% of visual context and save around 50% of GPU memory for the KV cache.

**Strengths:**

This paper introduces a novel design for two token predictors, formulating visual and textual context compression as an end-to-end optimization problem. Techniques such as Gumbel-Softmax and the Straight-Through Gradient Estimator are employed to enhance training stability. In addition, parallel sparsification inference optimization would provide an effective solution for model serving.

**Weaknesses:**

1. It is unclear whether the two learnable predictors are applied to each decoder layer. If they are, do they share the same parameters or are they independently parameterized?  If not all layers, please clarify which specific decoder layers the predictors are applied to.
2. Compressing generated text is primarily beneficial for long-generation tasks. Would you please provide specific statistics on the average output length for each task they evaluated? And discuss how the benefits of their method scale with increasing output length.
3. Please quantify the additional training time and computational resources required for this method compared to direct KV cache compression approaches, as well as additional prediction overhead compared normal decoding. This would help readers better understand the tradeoffs.
4. More recent related work about KV Cache compression method should be included, such as TidalDecode (https://arxiv.org/abs/2410.05076) focusing on decoding compression.
5. A few mathematical representations are not appropriate. For example, in section 3.1, $\mathcal{I}^I = $\{ $1, 2, \cdots, N_{l}^I $}, layer index $l$ should be set for $\mathcal{I}^I $; And in section 3.3.1, all layers share the same binary masks $\mathcal{M}^I and \mathcal{M}^{OT}$? since there are no layer index $l$.
6. The writing could be improved to make the narrative more concise and clear.

**Questions:**

1. Previous studies have shown that deeper layers exhibit higher attention sparsity. How is sparsification strength adjusted for different decoder layers?
2. Running predictors introduces additional latency. However, the proportion of this latency relative to overall prefill and decoding time is not disclosed. Additionally, in standard LLaVA decoding, previous KV cache is fetched, and byproduct attention scores are accumulated to assess token importance, which is quick and efficient. However, for "Online KV Cache Compression," running predictor incurs additional latency. And only the current token is used to predict all previous tokens, do you think solely performing prediction on a current token provide enough information? Have you analyzed the wrongly evicted tokens?
3. Can this method be applied to language models that utilize Group Query Attention?
4. Have you considered scenarios where long-text context is present during the prefill stage?

---

> ### Author Response · Authors · 2024-11-22
> **[2024/11/22 AOE] Detailed Rebuttal (1/4)**
>
> Dear Reviewer UyMq,
>
> Thank you for your valuable suggestions and thoughtful evaluation of our work! We recognize that some parts of the current paper may lead to misunderstandings or confusion. To address this, we have made detailed revisions to the paper, adding more explanations and illustrative figures to clarify the overall working mechanism. Additionally, we have conducted extensive experiments and provided detailed explanations to address your concerns.
>
> Sincerely Yours,
>
> The Authors
>
>
>
> > **W1. Predictor mechanism.**
>
> We apologize for any confusion. We clarify the predictor mechanism as follows.
>
> The learnable predictors are applied only once, after the $l$-th decoder layer of the MLLM. Specifically, once the predictor makes the decisions on tokens, this decisions are shared across all subsequent layers. In both the prefill and decoding without KV cache stages, after the predictor reduces the token set $S_l$ to $S_l^*$, all subsequent layers use this reduced token set $S_l^*$ for computation. In decoding with KV cache, the decision of whether to add the generated activations of the current token to the KV cache is likewise propagated to subsequent layers. This one-time token reduction strategy circumvents the need for complex adjustments to sparsity ratios of MLLM decoders. In practice, we follow FastV and set $l$ to 2. The ablation study for $l$ is presented in Tab. 6.
>
> *We have added more descriptions of the predictor mechanism in Sec. 3.3.1 and Sec. 3.3.2 of the revised paper.*

---

> ### Author Response · Authors · 2024-11-22
> **[2024/11/22 AOE] Detailed Rebuttal (2/4)**
>
> > **W2. Benchmark statistics and inference benefit discussion.**
>
> Thanks for your suggestions!
>
> First, we provide statistics for some of the vision understanding and generation ability benchmarks in the table below.
>
> |                               | VQA$^{v2}$ | VizWiz | SciQA | LVIS-VQA (single) | LVIS-VQA (multi) | ShareGPT4V-VQA (single) |
> | ----------------------------- | ---------- | ------ | ----- | ----------------- | ---------------- | ----------------------- |
> | Avg. image token length       | 576        | 576    | 576   | 576               | 576              | 576                     |
> | Avg. text token length        | 8          | 10     | 15    | 56                | 205              | 51                      |
> | Avg. output text token length | 2          | 3      | 6     | 159               | 351              | 1555                    |
> | Avg. token length             | 586        | 589    | 597   | 794               | 1132             | 2182                    |
>
> As can be seen, vision understanding benchmarks often require outputs in the form of choice answers such as "A," "B," "C," or single words or phrases to accurately assess an MLLM's understanding performance. However, in real-world scenarios, MLLMs are typically required to perform both understanding and generation, necessitating more detailed and extensive responses. This aligns with our constructed LVIS-VQA and ShareGPT4V-VQA [1] (which were added following the suggestion of Reviewer FzrR) benchmarks.
>
> The total token length (prefill and decoding) of LLaVA-1.5, FastV, and Dynamic-LLaVA across multiple benchmarks is presented in the table below.
>
> |                                 | VQA$^{v2}$       | VizWiz           | SciQA            | LVIS-VQA (single) | LVIS-VQA (multi) | ShareGPT4V-VQA (single) |
> | ------------------------------- | ---------------- | ---------------- | ---------------- | ----------------- | ---------------- | ----------------------- |
> | LLaVA-1.5-13B                   | 586              | 589              | 597              | 794               | 1132             | 2182                    |
> | LLaVA-13B-FastV$_{k=3,r=0.75}$  | 154 (-74%)       | 157 (-73%)       | 165 (-72%)       | 359 (-55%)        | 700 (-38%)       | 1750 (-20%)             |
> | ***Dynamic-LLaVA-13B$_{I/T}$*** | ***125 (-79%)*** | ***128 (-78%)*** | ***136 (-77%)*** | ***255 (-68%)***  | ***501 (-56%)*** | ***929 (-57%)***        |
>
> The improvement in inference efficiency that Dynamic-LLaVA provides is particularly significant on these three generation ability benchmarks compared to previous MLLM token reduction methods (e.g., FastV), which have longer output text lengths. **The results indicate that as the output length increases, Dynamic-LLaVA progressively exhibits a significant advantage in terms of token reduction percentage compared to FastV, as FastV only reduces image tokens during the prefill stage.**
>
> For instance, when the average output text token length is 2 (VQAv2), the efficiency gains achieved by FastV are similar to those of Dynamic-LLaVA (-74% vs. -79%). However, when the average output text token length is 1555 (ShareGPT4V-VQA-single), the efficiency gains achieved by FastV are significantly lower than those of Dynamic-LLaVA (-20% vs. -57%). Additionally, Dynamic-LLaVA also demonstrates superior performance.
>
> *Detailed discussions and results are presented in Appendix A.3.1 and Tab. 8 of the revised paper.*

---

> ### Author Response · Authors · 2024-11-22
> **[2024/11/22 AOE] Detailed Rebuttal (3/4)**
>
> > **W3. Training time and lantency.**
>
> We apologize for not providing this part in our paper.
>
> In the table below, we report the additional training time for Dynamic-LLaVA (measured on 8 A100 80G) and the inference latency when using KV cache (measured on one A100 80G, with the prefill time and the average decoding latency per token for generating 1000 tokens and the fixed batch size of 1).
>
> Additional training time:
>
> | Method                    | Training time (h) |
> | ------------------------- | ----------------- |
> | Dynamic-LLaVA-7B$_{I/T}$  | $\sim$13          |
> | Dynamic-LLaVA-13B$_{I/T}$ | $\sim$24          |
>
>
> Inference latency when using KV cache:
>
> | Method                          | Prefill latency (ms) | Decoding latency (ms) |
> | ------------------------------- | -------------------- | --------------------- |
> | LLaVA-1.5-13B                   | 124.52               | 28.42                 |
> | ***Dynamic-LLaVA-13B$_{I/T}$*** | ***72.51***          | ***26.95***           |
>
> **The latency results measured on GPU indicate that the learnable lightweight predictors contribute a negligible increase to inference latency (in practice, less than 1%).** Meanwhile, the proposed Dynamic-LLaVA framework exhibits an average reduction of 1.57 ms per token in generation latency compared to LLaVA-1.5 when decoding with KV cache. This improvement in inference speed is due to Dynamic-LLaVA significantly reducing the length of KV cache, which speeds up the attention operations.
>
> *Detailed discussions and results are presented in Appendix A.5.2, Tab. 14 and Tab. 15 of the revised paper.*
>
>
>
> > **W4. More recent related works about KV Cache compression.**
>
> Thank you for your suggestion!  *We have added the latest KV Cache compression methods and a detailed discussion in Sec. 2.2 and Appendix A.3.2.*
>
>
>
> > **W5. Mathematical representations.**
>
> We apologize for the confusion. As shown in the "response to W1", we only use the predictors once to generate a mask, so to simplify the notations, we did not use the layer index $l$.
>
>
>
> > **W6. Writing.**
>
> Thank you for your suggestions! We have revised the presentation of the paper to add simplified explanations, visual flowcharts, and more discussions. Our changes to the paper are highlighted in blue.
>
>
>
> > **Q1. Sparsification strength adjustment strategy.**
>
> Specifically, we first adopted the recommendation from previous work (FastV) and set the sparsification to occur after the second layer. Meanwhile, based on our empirical observations, the layer after which Dynamic-LLaVA performs sparsification is not a sensitive parameter, which enhances its adaptability. *Detailed ablation results regarding the choice of layer $l$ can be found in Tab. 6.*
>
>
>
> > **Q2-1. Lantency of decoding with KV cache.**
>
> *Please refer to the "response for W3".* In short, the predictors introduce negligible inference overhead, accounting for less than 1% (compared to the total computation cost of MLLM).
>
>
>
> > **Q2-2. Efferency of our ''online'' KV cache compression method and traditional ''attention-based'' KV cache compression method.**
>
> *Please refer to the "response for W3", the additional inference overhead introduced by Dynamic-LLaVA is practically negligible in inference.*  Additionally, for traditional "attention-based" KV cache compression methods (such as H2O[2]), which require attention scores for KV cache compression during decoding, the attention operations of efficient inference operators are often implicit (e.g., FlashAttention[3, 4] and `torch.nn.functional.scaled_dot_product_attention`). This can necessitate an additional computation of attention mechanism to obtain attention scores, which may slightly impact inference speed in scenarios with long KV cache length. This phenomenon is shown in the table below, using the `torch.nn.functional.scaled_dot_product_attention` operator.
>
> | Method                      | Prefill latency (ms) | Decoding latency (ms) |
> | --------------------------- | -------------------- | --------------------- |
> | LLaVA-1.5-13B               | 124.52               | 28.42                 |
> | LLaVA-1.5-13B+H2O$_{r=0.5}$ | 111.84               | 37.71                 |
>
> **However, Dynamic-LLaVA relies solely on the features of the current token for prediction when decoding with KV cache, thus avoiding this issue.**
>
> *Detailed discussions and results are provided in Appendix A.5.2 and Tab. 14 of the revised paper.*

---

> ### Author Response · Authors · 2024-11-22
> **[2024/11/22 AOE] Detailed Rebuttal (4/4)**
>
> > **Q2-3. Only use the current token feature information to make decision.**
>
> First, we want to clarify that Dynamic-LLaVA evaluates each current token's features to determine whether its activations which through $W_K$ and $W_V$ should be added to the KV cache. It does not use the information of the current token to determine the activations of all previous tokens show be discarded or not. Therefore, making decisions solely based on the activations of the current token is both intuitive and feasible. Our experimental results also partially validate this approach.
>
> **It is noteworthy that this method of making decisions based solely on the features of the current token enables independence from past KV cache usage. Consequently, this allows for accelerated MLLMs during prefill and decoding without KV cache.**
>
>
> > **Q2-4. Analysis of the wrongly evicted tokens.**
>
> *We present visualizations of the decisions made by Dynamic-LLaVA's predictors for images and output text in Fig. 6, Fig. 7, and Fig. 8.* As shown in these figures, Dynamic-LLaVA primarily discards background and redundant content in the vision context while retaining the main subjects comprehensively. For output texts, the discard decisions mostly involve articles, prepositions, and suffixes of long words. We believe that the decisions made by Dynamic-LLaVA are reasonable in most cases.
>
>
>
> > **Q3. Applicability for Group Query Attention.**
>
> We believe this is entirely feasible. Allow us to illustrate using PyTorch dimensions as an example. In the computation of the Group Query Attention, the dimensions of $Q,K,V$ are typically [$B$, $N$, $H_{num}$, $H_{dim}$]. The grouping mechanism occurs along the third dimension, i.e., $H_{num}$. Dynamic-LLaVA reduces the size of the second dimension, $N$, which improves computation efficiency without affecting the computation of the Group Query Attention operation.
>
>
>
> > **Q4. Long-text context scenarios during the prefill stage.**
>
> Good question! In the table below, we present the results for sparsifying the text context during the prefill stage. We add a predictor for sparsifying text tokens, with an architecture identical to that of the output predictor. The sparsification ratio for text tokens is set to 30%, and we use exactly all same settings as Dynamic-LLaVA.
>
> | Method                                                       | POPE | VQA$^{v2}$ | GQA  |
> | ------------------------------------------------------------ | ---- | ---------- | ---- |
> | Dynamic-LLaVA-7B$_{I/T}$                                     | 85.9 | 77.8       | 61.3 |
> | Dynamic-LLaVA-7B$_{I/T}$+30% text context sparsification in prefill | 85.1 | 75.3       | 60.2 |
>
> **It can be observed that sparsifying only 30% of the text context leads to a significant performance drop.** We believe this is because MLLMs rely on the instruction (prompt) to generate responses, and sparsifying the text context may result in critical content being omitted from the question, leading to incorrect answers. We consider this a direction worth exploring in future work.
>
> *We discuss and present the results in detail in Appendix A.3.3 and Tab. 9 of the revised paper.*
>
>
>
> > **Reference**
>
> [1] Sharegpt4v: Improving large multi-modal models with better captions. ECCV2024.
>
> [2] H2O: Heavy-Hitter Oracle for Efficient Generative Inference of Large Language Models. NeurIPS2023.
>
> [3] Flashattention: Fast and memory-efficient exact attention with io-awareness. NeurIPS2022.
>
> [4] FlashAttention-2: Faster Attention with Better Parallelism and Work Partitioning. ICLR2024.

---

> > ### Comment · Reviewer_UyMq · 2024-11-27
> > **Post-rebuttal comments**
> >
> > The authors have addressed my most concerns and I will raise my score.

---

> > > ### Author Response · Authors · 2024-11-27
> > >
> > > Dear Reviewer UyMq,
> > >
> > > Thank you again for your invaluable time and effort on our paper. Thank you very much for approving our work!
> > >
> > > Sincerely Yours,
> > >
> > > The Authors

---

> ### Author Response · Authors · 2024-11-25
> **Any additional concerns or questions**
>
> Dear Reviewer UyMq,
>
> Thank you once again for reviewing our paper and providing valuable feedback. We have carefully considered your suggestions and made multiple revisions to enhance the clarity, depth, and contribution of the paper. Your constructive insights and feedback have played a significant role in the process of improving our paper.
>
> We sincerely hope you will continue to engage in the discussion. Should you have further questions or concerns, we are more than willing to provide additional explanations or supporting materials. Your insights are critical to refining our research and ensuring its relevance and impact.
>
> Furthermore, we hope these revisions and clarifications will encourage you to reassess your evaluation, as these updates directly address your constructive comments. If you have any additional questions or concerns, please feel free to reach out to us. We are committed to ensuring that all issues are thoroughly addressed.
>
> Sincerely Yours,
>
> The Authors

---

### Official Review · Reviewer_LdRV · 2024-11-04

**Soundness:** 2
**Presentation:** 3
**Contribution:** 2
**Rating:** 6
**Confidence:** 4

**Summary:**

In this work, the authors propose Dynamic-LLaVA, a novel approach to enhance both prefilling and generation efficiency in Multimodal Large Language Models (MLLMs). The modified model employs two predictors: an image predictor that dynamically identifies and drops less significant image tokens during prefilling, and an output predictor that filters out non-essential output text tokens. When training these two predictors, the authors propose an end-to-end sparsification training pipeline, which added a mask to the original softmax attention to keep a fixed number of image tokens and generated text tokens. Additionally, the argmax is replaced by Gumbel-Softmax during backward to avoid the gradient flow problem. Extensive experiments show the effectiveness and efficiency of the proposed method.

**Strengths:**

1. This work pioneers the reduction of KV cache length during long text generation for MLLMs, whereas previous research has focused solely on reducing prefilled vision tokens.
2. The end-to-end sparsification training can be viewed as a novel way to enhance the token sparsity of MLLMs.
3. The paper proposed extensive experiments to show the effectiveness of the proposed method.

**Weaknesses:**

1. While Dynamic-LLaVA demonstrates improvements over previous approaches like FastV and PruMerge+, it requires additional post-training, which introduces extra computational overhead. In table 3, it seems that the performance of Dynmaic-LLaVA and FastV are even comparable. Therefore, for real-world deployments where computational efficiency is crucial, implementing existing methods may be more practical. However, this raises an important question: the work would benefit from a clearer justification for why a trainable approach is necessary and what specific advantages it offers over more lightweight solutions.

2. For the output reduction, as we only need to determine whether preserving the current generated token, this pipeline is the same as some work for LLM, such as StreamingLLM [1], H2O [2]. However, the author only compare the Random and Structure strategies. Therefore, I think this is a well-defined problem for LLMs, and the current experimental results cannot convince me.

3. The methodology section of this work requires improvement to enhance clarity and comprehension. It is currently challenging for readers to grasp the entire training and inference pipeline quickly. Additionally, several design choices related to the training process lack sufficient explanation, which may leave readers puzzled about the rationale behind them.

4. Furthermore, the name "Dynamic-LLaVA" may lead to misconceptions about the method. It implies that the system dynamically drops the KV cache at each layer; however, it should be clarified that the cache is dropped permanently and cannot be recovered in subsequent layers.

5. The training process is not aligned with the inference process for the output predictor. The former one can see all generated tokens and decide the most important ones, while the latter one can only decide whether to drop the current one based on previous context.

[1] Xiao, Guangxuan, et al. "Efficient streaming language models with attention sinks." arXiv preprint arXiv:2309.17453 (2023).
[2] Zhang, Zhenyu, et al. "H2o: Heavy-hitter oracle for efficient generative inference of large language models." Advances in Neural Information Processing Systems 36 (2024).

**Questions:**

See weaknesses.

---

> ### Author Response · Authors · 2024-11-22
> **[2024/11/22 AOE] Detailed Rebuttal (1/7)**
>
> Dear Reviewer LdRV,
>
> Thank you very much for your valuable suggestions! We have conducted extensive experiments and revised our paper accordingly. We also apologise for some misunderstandings you may have had about our paper. We have tried to address these issues in the following rebuttal, and have added further clarifications to the paper to avoid potential misunderstandings.
>
> Sincerely Yours,
>
> The Authors
>
>
>
> > **W1-1. Dynamic-LLaVA is a training-required method.**
>
> Currently, not all MLLM token reduction methods are training-free (i.e., they cannot be directly applied to MLLMs without training). For instance, methods such as IVTP [1], PruMerge+ [2], VoCo-LLaMA [3], and TRIM [4] are training-required methods.
>
> Additionally, it is important to note that PruMerge+ [2], which you mentioned, **also requires additional post-training**. For LLaVA-1.5, both Dynamic-LLaVA$_{I/T}$ and PruMerge+ [2] undergo an extra one epoch of instruct-following finetuning for MLLMs. However, **Dynamic-LLaVA$_{I/T}$ outperforms PruMerge+ in 10 out of 12 valid results across LLaVA-1.5-7B and LLaVA-1.5-13B (see Tab. 1 in the main paper), while also offering additional decoding inference benefits.**
>
> We present comparison results for FastV, which undergo one additional epoch of sparsification training, maintaining the same hyperparameter settings as Dynamic-LLaVA. The "*" denotes FastV that has undergone additional sparsification training. *Detailed results and settings can be found in Tab. 1 of the revised paper and Appendix A.4.2.*
>
> |                             | Free           | Token （prefill）$\downarrow$ | TFLOPs（prefill）$\downarrow$ | Token （decoding）$\downarrow$ | VQA$^{v2}$    | GQA           | SQA           | VQA$^{T}$     | POPE          | MME               | MMB           | SEED          |
> | --------------------------- | -------------- | ----------------------------- | ----------------------------- | ------------------------------ | ------------- | ------------- | ------------- | ------------- | ------------- | ----------------- | ------------- | ------------- |
> | ***Dynamic-LLaVA$_{I/T}$*** | ***&#10005;*** | ***20%***                     | ***25%***                     | ***50%***                      | ***77.9***    | ***61.3***    | ***68.6***    | ***56.5***    | ***85.9***    | ***1501.0***      | ***64.1***    | ***65.0***    |
> | * *(ECCV24) FastV [5]*       | *✕*            | *25%*                         | *32%*                         | *100%*                         | *74.2 (-3.7)* | *56.6 (-4.7)* | *64.0 (-4.6)* | *54.3 (-2.2)* | *82.7 (-3.2)* | *1292.2 (-208.8)* | *58.6 (-5.5)* | *58.4 (-6.6)* |
>
> **It is evident that, after undergoing the same additional sparsification training, the performance of Dynamic-LLaVA$_{I/T}$ significantly surpasses that of FastV.**

---

> > ### Author Response · Authors · 2024-11-22
> > **[2024/11/22 AOE] Detailed Rebuttal (2/7)**
> >
> > > **W1-2. Does Dynamic-LLaVA show no performance improvement?**
> >
> > We present the performance of current SoTA and recent token reduction methods (2024/3->2024/10). The results in the table are based on evaluations conducted with LLaVA-1.5-7B, using the performance of Dynamic-LLaVA_I/T as the base. We report the performance differences of other methods relative to ours. The "Free" column indicates whether a method is training-free.
> >
> > |                             | Free           | Token （prefill）$\downarrow$ | TFLOPs（prefill）$\downarrow$ | Token （decoding）$\downarrow$ | VQA$^{v2}$    | GQA           | SQA           | VQA$^{T}$     | POPE          | MME               | MMB           | SEED          |
> > | --------------------------- | -------------- | ----------------------------- | ----------------------------- | ------------------------------ | ------------- | ------------- | ------------- | ------------- | ------------- | ----------------- | ------------- | ------------- |
> > | ***Dynamic-LLaVA$_{I/T}$*** | ***&#10005;*** | ***20%***                     | ***25%***                     | ***50%***                      | ***77.9***    | ***61.3***    | ***68.6***    | ***56.5***    | ***85.9***    | ***1501.0***      | ***64.1***    | ***65.0***    |
> > | * *(ECCV24) FastV [5]*       | *✕*            | *25%*                         | *32%*                         | *100%*                         | *74.2 (-3.7)* | *56.6 (-4.7)* | *64.0 (-4.6)* | *54.3 (-2.2)* | *82.7 (-3.2)* | *1292.2 (-208.8)* | *58.6 (-5.5)* | *58.4 (-6.6)* |
> > | (ECCV24) IVTP [1]           | ✕              | -                             | 53%                           | 100%                           | 77.8 (-0.1)   | 60.4 (-0.9)   | 67.8 (-0.8)   | 58.2 (+1.7)   | 85.7 (-0.2)   | -                 | 66.1 (+2.0)   | 56.4 (-8.6)   |
> > | PruMerge+ [2]               | ✕              | 25%                           | 25%                           | 100%                           | 76.8 (-1.1)   | -             | 68.3 (-0.3)   | 57.1 (+0.6)   | 84.0 (-1.9)   | 1462.4 (-38.6)    | 64.9 (+0.8)   | -             |
> > | VoCo-LLaMA [3]              | ✕              | 22%                           | 22%                           | 100%                           | 76.9 (-1.0)   | 59.8 (-1.5)   | -             | -             | -             | -                 | 61.0 (-3.1)   | 59.1 (-5.9)   |
> > | TRIM [4]                    | &#10005;       | 21%                           | -                             | 100%                           | 76.4 (-1.5)   | 61.4 (+0.1)   | 48.1 (-20.5)  | 53.7 (-2.8)   | 85.3 (-0.6)   | 1461.3 (-39.7)    | 67.4 (+3.3)   | -             |
> > | (ECCV24) FastV [5]          | ✓              | 25%                           | 32%                           | 100%                           | 75.1 (-2.8)   | 57.5 (-3.8)   | 68.7 (+0.1)   | 56.2 (-0.3)   | 81.0 (-4.9)   | 1458.9 (-42.1)    | 63.5 (-0.6)   | 62.8 (-2.2)   |
> > | HiRED [6]                   | ✓              | 20%                           | 20%                           | 100%                           | 74.7 (-3.2)   | -             | 66.4 (-2.2)   | 44.2 (-12.3)  | -             | -                 | -             | -             |
> > | SparseVLM [7]               | ✓              | 22%                           | -                             | 100%                           | 73.8 (-4.1)   | 56.0 (-5.3)   | 67.1 (-1.5)   | 54.9 (-1.6)   | 80.5 (-5.4)   | 1696 (+195)       | 60.0 (-4.1)   | -             |
> >
> > Our proposed Dynamic-LLaVA$_{I/T}$ achieves the best performance in 42 out of 50 valid results across eight commonly used benchmarks. **Therefore, even when compared to the latest MLLM token reduction methods, the performance improvements offered by our method are sufficiently significant.**
> >
> > For the generation ability benchmarks, it is important to note that Dynamic-LLaVA_{I/T} achieves comparable PPL and METEOR scores to FastV [5], indicating similar levels of fluency and quality in generated text [8, 9, 10]. Maintaining normal text generation ability is a fundamental requirement for an MLLM. This suggests that, while some generated tokens are discarded, the impact on Dynamic-LLaVA_{I/T}'s ability to generate coherent text is minimal. However, this similarity in scores does not imply equivalent overall performance (*see Tab. 1, where Dynamic-LLaVA$_{I/T}$ outperforms FastV in 14 out of 16 results on the LLaVA-1.5 experiments*). Additionally, Dynamic-LLaVA$_{I/T}$ further reduces 50% decoding computation (without KV cache) and GPU memory overhead (with KV cache) compared to FastV [5].  This means that, compared to FastV, **Dynamic-LLaVA$_{I/T}$, while having similar generation ability, also offers higher prefill inference benefits, additional decoding inference benefits, and stronger vision understanding performance.**

---

> > > ### Author Response · Authors · 2024-11-22
> > > **[2024/11/22 AOE] Detailed Rebuttal (3/7)**
> > >
> > > > **W1-3. The advantages of Dynamic-LLaVA compared to existing token reduction methods.**
> > >
> > > Here we summarize the advantages of Dynamic-LLaVA compared to existing token reduction methods, including both training-free and training-required methods:
> > >
> > > 1. **Efficiency Advantages**: Dynamic-LLaVA reduces prefill computation consumption and offers additional benefits during the decoding stage, including further computation cost reduction (without KV cache) and GPU memory savings (with KV cache).
> > > 2. **Performance Advantages**: As shown in ''Response to W1-2'', Dynamic-LLaVA significantly outperforms existing token reduction methods in terms of performance.
> > >
> > > **Therefore, even though Dynamic-LLaVA is a training-required method, the approach we propose has substantial contributions and advantages.**
> > >
> > >
> > >
> > > > **W2-1. Is the output token reduction pipeline of Dynamic-LLaVA when decoding with KV cache the same as the LLM KV cache compression pipeline?**
> > >
> > > Dynamic-LLaVA differs from the typical pipeline of LLM KV cache compression methods, such as H2O [11].
> > >
> > > When Dynamic-LLaVA is used in the case of decoding with KV cache, it can indeed be regarded as a form of **''online KV cache compression''**.  Dynamic-LLaVA evaluates each current token's features to determine whether its activations which throughed $W_K$ and $W_V$ should be added to KV cache, rather than removing entries from the past cache (*detailed in Eq. 6 of the main paper*). KV cache compression methods, however, often rely on past generated KV cache. E.g., H2O [11] uses the current $Q$ with the past KV cache to compute attention scores and decide which past cache activations to discard, rather than directly deciding to retain or discard activations of the current token.
> > >
> > > The design of "online KV cache compression" enables Dynamic-LLaVA to **significantly accelerate MLLMs' inference even in scenarios of decoding without KV cache, a feat that is challenging to achieve with traditional LLM KV cache compression methods**.
> > >
> > >
> > > > **W2-2. The results of applying LLM KV cache compression methods to MLLMs.**
> > >
> > > We conduct experiments of H2O[11] on vision understanding and generation ability benchmarks for MLLMs. Note that the scenarios we compared are the generation process of MLLMs with KV cache.
> > >
> > > Firstly, concerning the vision understanding task, the table below shows the performance of H2O and our method. Please note that the "*" denotes enhancements we made to H2O, where $k$ represents the layer from which the reduction of KV cache begins, and $r$ indicates the keep rate of KV cache in each of the prefill and decoding stages. *Detailed results and settings can be found in Tab. 1 of the revised paper and Appendix A.4.2.*
> > >
> > > |                                         | KV Cache (prefill) $\downarrow$ | TFLOPs（prefill）$\downarrow$ | VQA$^{v2}$  | GQA         | SQA          | VQA$^{T}$    | POPE         | MME             | MMB         | SEED         |
> > > | --------------------------------------- | ------------------------------- | ----------------------------- | ----------- | ----------- | ------------ | ------------ | ------------ | --------------- | ----------- | ------------ |
> > > | ***Dynamic-LLaVA-7B$_{I/T}$***          | ***25%***                       | ***25%***                     | ***77.9***  | ***61.3***  | ***68.6***   | ***56.5***   | ***85.9***   | ***1501.0***    | ***64.1***  | ***65.0***   |
> > > | LLaVA-1.5-7B+H2O$_{r=0.5}$ [11]          | 50%                             | 100%                          | $-$         | $-$         | $-$          | 11.6 (-44.9) | 50.5 (-35.4) | 500.5 (-1000.5) | $-$         | $-$          |
> > > | *LLaVA-1.5-7B+H2O$_{k=10, r=0.5}$ [11]  | 67%                             | 100%                          | 77.9 (-0.0) | 61.0 (-0.3) | 41.9 (-26.7) | 55.9 (-0.6)  | 86.9 (+1.0)  | 1458.4 (-42.6)  | 1.4 (-62.7) | 26.8 (-38.2) |
> > > | ***Dynamic-LLaVA-13B$_{I/T}$***         | ***25%***                       | ***25%***                     | 78.8        | 62.5        | 72.4         | 59.6         | 86.5         | 1563.3          | 66.9        | 66.5         |
> > > | *LLaVA-1.5-13B+H2O$_{k=10, r=0.5}$ [11] | 67%                             | 100%                          | 78.9 (+0.1) | 62.3 (-0.2) | 48.5 (-23.9) | 57.5 (-2.1)  | 87.3 (+0.8)  | 1448.3 (-115.0) | 4.7 (-62.2) | 32.3 (-34.2) |

---

> > ### Comment · Reviewer_UyMq · 2024-11-26
> > **FastV does not require additional traning**
> >
> > Thank you for the authors' response. I'd like to clarify that FastV [1] is a plug-and-play method for LVLMs and does not require model training. Additionally, the TFLOP reduction rate during the prefill stage depends on various factors, including model configuration and hyperparameters. These factors include the specific layers pruned, the total number of model layers, and whether multi-head or group-query attention is used (see Equation 5 in Section 4.2 [1]). These settings should be clearly specified.
> >
> > [1] An Image is Worth 1/2 Tokens After Layer 2: Plug-and-Play Inference Acceleration for Large Vision-Language Models. ECCV2024.

---

> ### Author Response · Authors · 2024-11-22
> **[2024/11/22 AOE] Detailed Rebuttal (4/7)**
>
> We found that even discarding only 50% of the prefill KV cache, directly applying H2O to an MLLM leads to severe performance degradation (see the second row of the table above). *This result is similar to those reported in ''Table 5(g) of the work [5]'' concerning the application of StreamingLLM [12] on MLLMs, indicating that directly applying LLM KV cache compression methods to MLLMs can lead to a sharp decline in performance.* Thus, we enhance H2O to start reducing KV cache after the $10$-th layer (Dynamic-LLaVA starts sparsifying after the $2$-nd layer). **Despite retaining less KV cache (25% vs. 67%), Dynamic-LLaVA still achieves better performance. Moreover, Dynamic-LLaVA additionally reduces the inference overhead of the MLLM by 75%.**
>
> For the generation ability task, the table below shows the performance of H2O and our method. ''FastV-7B [5]+H2O$_{r=0.5}$ [11]'' indicates that the prefill stage is sparsified by FastV [5], and the KV cache during the decoding stage is reduced by H2O.
>
> |                                  | KV Cache (prefill) $\downarrow$ | TFLOPs（prefill）$\downarrow$ | KV Cache (decoding) $\downarrow$ | PPL of LVIS-VQA (single-round) | METEOR of LVIS-VQA (single-round) | PPL of LVIS-VQA (multi-round) | METEOR of LVIS-VQA (multi-round) | PPL of SharedGPT4V-VQA (single-round) |
> | -------------------------------- | ------------------------------- | ----------------------------- | -------------------------------- | ------------------------------ | --------------------------------- | ----------------------------- | -------------------------------- | ------------------------------------- |
> | ***Dynamic-LLaVA-7B$_{I/T}$***   | ***25%***                       | ***25%***                     | ***50%***                        | ***4.90***                     | ***0.3108***                      | ***3.17***                    | ***0.4251***                     | ***3.14***                            |
> | LLaVA-1.5-7B+H2O$_{r=0.5}$ [11]  | 50%                             | 100%                          | 50%                              | 78.95                          | 0.0388                            | 42.60                         | 0.0892                           | 25.54                                 |
> | FastV-7B [5]+H2O$_{ r=0.5}$ [11] | 33%                             | 100%                          | 50%                              | 5.62                           | 0.3071                            | 3.57                          | 0.4049                           | 3.56                                  |
> | ***Dynamic-LLaVA-13B$_{I/T}$***  | ***25%***                       | ***25%***                     | ***50%***                        | ***4.76***                     | ***0.3151***                      | ***3.07***                    | ***0.4243***                     | ***3.10***                            |
> | LLaVA-1.5-13B+H2O$_{r=0.5}$ [11] | 50%                             | 100%                          | 50%                              | 41.44                          | 0.0428                            | 18.62                         | 0.0493                           | 10.51                                 |
> | FastV-13B [5]+H2O$_{r=0.5}$ [11] | 33%                             | 100%                          | 50%                              | 5.30                           | 0.3151                            | 3.45                          | 0.4127                           | 3.41                                  |
>
> Applying H2O directly in the generation process of MLLMs leads to a significant degeneration in the fluency and quality of the generated text. Even when H2O is used solely during the decoding stage of MLLMs, Dynamic-LLaVA still able to generate slightly more fluent and higher-quality text while using less KV cache throughout the entire generation process. Consequently, compared to H2O, **Dynamic-LLaVA has a slightly better generation ability with higher prefill inference benefits, additional decoding inference benefits (when decoding without KV cache), and stronger vision understanding performance.**
>
> *Detailed results and settings regarding the performance of H2O can be found in Tab. 1, Tab. 3, and Appendix A.4.2 of the revised paper.*

---

> > ### Author Response · Authors · 2024-11-22
> > **[2024/11/22 AOE] Detailed Rebuttal (5/7)**
> >
> > > **W2-3. The specific differences and advantages of Dynamic-LLaVA compared to LLM KV cache compression methods.**
> >
> > We summarize the specific differences and advantages of Dynamic-LLaVA compared to LLM KV cache compression methods as follows:
> >
> > 1. **Pipeline Differences**: Dynamic-LLaVA, when decoding with KV cache, can be regarded as a form of "online KV cache compression." It directly determines, based on the features of the current token, whether the activations generated by subsequent MLLM layers should be added to the KV cache. In contrast, LLM KV cache compression methods typically rely on subsets of previously generated KV cache to facilitate the selection of critical activations.
> > 2. **Efficiency Advantages:** Dynamic-LLaVA introduces a tailored sparsification inference scheme specific to various inference modes, i.e., prefill, decoding with KV cache, and decoding without KV cache. Only in the case of decoding with KV cache can it be regarded as a form of "online KV cache compression". For prefill and decoding without KV cache, where KV cache activations are not involved, Dynamic-LLaVA can still significantly accelerate MLLM's computation, a benefit that KV cache compression methods cannot achieve.
> > 3. **Performance Advantages:** As shown in ''Response to W2-2'', Dynamic-LLaVA demonstrates significant performance advantages over the LLM KV cache compression method H2O when applied to MLLMs.
> >
> > In the revised paper, we include a more detailed comparison with LLM KV cache compression methods and discuss the computational benefits in the prefill and decoding without KV cache scenarios. This includes the importance of prefill stage inference acceleration for MLLMs [1-7], the GPU memory overhead advantage [13] and potential applications for decoding without KV cache, such as the widely used speculative decoding strategy [14, 15, 16]. *Detailed discussion can be found in Appendix A.3.2.*
> >
> >
> >
> > > **W3-1. ''Method'' section should be improved to enhance clarity and comprehension.**
> >
> > Thank you for your suggestions. We have revised the ''Method'' section of the paper and added more concise descriptions along with visualized workflow diagrams (Fig. 3 and Fig. 4) to facilitate readers' understanding of our approach. The changes to the paper are highlighted in blue.
> >
> >
> >
> > > **W3-2. Design choices for training process.**
> >
> > Sorry for the confusion. Below, we provide some of the design principles:
> >
> > 1. Why MaskedSoftmax?
> >
> >    We provide the motivation behind this design in L311-316. i.e., ''Given the binary masks of the token sets, we need to use these masks to isolate the influence of non-essential tokens on the important tokens during the LLM training computation. One native idea is to directly set the values of the unnecessary tokens to zero vectors. However, this method introduces a problem: when we sparsify output text tokens during the parallel training of LLM, discarding the value of an output text token will prevent it from autoregressively predicting the next output text token, making it impossible to compute the loss.''
> >    Furthermore, we evaluate the effect on the MaskedSoftmax operation in the table below.
> >
> >    |                                            | POPE | VQAv2 | GQA  |
> >    | ------------------------------------------ | ---- | ----- | ---- |
> >    | Dynamic-LLaVA-7B$_{I/T}$ (Ours)            | 85.9 | 77.8  | 61.3 |
> >    | Dynamic-LLaVA-7B$_{I/T}$ w/o MaskedSoftmax | 84.5 | 76.7  | 59.8 |
> >
> >    **It is evident that directly setting the values of unnecessary tokens to zero vectors leads to a significant performance degradation.**
> >
> >    *Detailed results and discussion can be found in Tab. 7 and Appendix A.2 of the revised paper.*
> >
> > 2. Why Gumbel-Softmax+STE?
> >
> >    We provide the motivation behind this design in L329-332. i.e., ''In addition to the issues mentioned above, we have the gradient flow problem in the backward propagation of training. The $\operatorname{argmax}(\cdot)$ operation we performed to obtain $\mathcal{M}^{I}$ and $\mathcal{M}^{OT}$ is non-differentiable and impedes end-to-end training.''
> >
> > 3. $r^I$, $r^{OT}$, $Len^{OT}$, $\lambda$.
> >
> >    *The experiments regarding the selection of these parameters are shown in Tab. 5 and Tab. 6.*

---

> > > ### Author Response · Authors · 2024-11-22
> > > **[2024/11/22 AOE] Detailed Rebuttal (6/7)**
> > >
> > > > **W4-1. Why is our framework named "Dynamic-LLaVA"?**
> > >
> > > The "Dynamic" in Dynamic-LLaVA reflects its use of output predictor to dynamically assess the importance of each token. This dynamic aspect is reflected in making token-level decisions for each sample. Such the "dynamical decision" concept aligns with similar definitions across various fields [17, 18, 19], which is why we named our method "Dynamic-LLaVA". *In the revised version of our paper, we have included additional explanations in the ''Method'' section to avoid potential misunderstandings.*
> > >
> > >
> > >
> > > >  **W4-2. The working mechanism of Dynamic-LLaVA during decoding with KV cache.**
> > >
> > > It is important to clarify that Dynamic-LLaVA does not delete past KV activations during decoding with KV cache. Instead, it determines whether to add the activations of the current token to the MLLM's KV cache based on the token's features. The decision regarding the activations of the current output text token is shared across all subsequent layers beyond the $l$-th layer. *To improve clarity, we have included specific explanations in the ''Method'' section of the revised paper.*
> > >
> > >
> > >
> > > > **W5. Whether the training process of Dynamic-LLaVA is not aligned with the inference process for the output predictor?**
> > >
> > > The form of masking and predictors for Dynamic-LLaVA  maintains uniformity between training and inference. Specifically, during training, the presence of the causal attention mask [8] ensures that each token focuses solely on prior context information, while the use of MLP in the output predictor $P^I$ ensures that decisions are based solely on its own features, consistent with the process used during decoding. *We have included this description in the 'Method''section of the revised paper to avoid misunderstandings.*

---

> > > > ### Author Response · Authors · 2024-11-22
> > > > **[2024/11/22 AOE] Detailed Rebuttal (7/7)**
> > > >
> > > > > **Reference**
> > > >
> > > > [1] IVTP: Instruction-guided Visual Token Pruning for Large Vision-Language Models. ECCV2024.
> > > >
> > > > [2] Llava-prumerge: Adaptive token reduction for efficient large multimodal models. Arxiv2403.
> > > >
> > > > [3] VoCo-LLaMA: Towards Vision Compression with Large Language Models. Arxiv2406.
> > > >
> > > > [4] Less is More: A Simple yet Effective Token Reduction Method for Efficient Multi-modal LLMs. Arxiv2409.
> > > >
> > > > [5] An Image is Worth 1/2 Tokens After Layer 2: Plug-and-Play Inference Acceleration for Large Vision-Language Models. ECCV2024.
> > > >
> > > > [6] HiRED: Attention-Guided Token Dropping for Efficient Inference of High-Resolution Vision-Language Models in Resource-Constrained Environments. Arxiv2408.
> > > >
> > > > [7] SparseVLM: Visual Token Sparsification for Efficient Vision-Language Model Inference. Arxiv2410.
> > > >
> > > > [8] Language models are unsupervised multitask learners. OpenAI blog.
> > > >
> > > > [9] Perplexity—a measure of the difficulty of speech recognition tasks. The Journal of the Acoustical Society of America.
> > > >
> > > > [10] Meteor: An automatic metric for mt evaluation with improved correlation with human judgments. ACL Workshop 2005.
> > > >
> > > > [11] H2O: Heavy-Hitter Oracle for Efficient Generative Inference of Large Language Models. NeurIPS2023.
> > > >
> > > > [12] Efficient streaming language models with attention sinks. Arxiv2309.
> > > >
> > > > [13] Fast inference from transformers via speculative decoding. ICML2023.
> > > >
> > > > [14] Accelerating large language model decoding with speculative sampling. Arxiv2302.
> > > >
> > > > [15] Online speculative decoding. Arxiv2310.
> > > >
> > > > [16] Improved Baselines with Visual Instruction Tuning. CVPR2024.
> > > >
> > > > [17] SkipNet: Learning Dynamic Routing in Convolutional Networks. ECCV2018.
> > > >
> > > > [18] Learning Dynamic Routing for Semantic Segmentation. CVPR2020.
> > > >
> > > > [19] Dynamic Deep Neural Networks: Optimizing Accuracy-Efficiency Trade-Offs by Selective Execution. AAAI2018.

---

> ### Author Response · Authors · 2024-11-25
> **Any additional concerns or questions**
>
> Dear Reviewer LdRV,
>
> Thank you once again for reviewing our paper and providing valuable feedback. We have carefully considered your suggestions and made multiple revisions to enhance the clarity, depth, and contribution of the paper. Your constructive insights and feedback have played a significant role in the process of improving our paper.
>
> We sincerely hope you will continue to engage in the discussion. Should you have further questions or concerns, we are more than willing to provide additional explanations or supporting materials. Your insights are critical to refining our research and ensuring its relevance and impact.
>
> Furthermore, we hope these revisions and clarifications will encourage you to reassess your evaluation, as these updates directly address your constructive comments. If you have any additional questions or concerns, please feel free to reach out to us. We are committed to ensuring that all issues are thoroughly addressed.
>
> Sincerely Yours,
>
> The Authors

---

> ### Author Response · Authors · 2024-11-27
> **[2024/11/27 AOE] Additional Clarification**
>
> Dear Reviewer UyMq,
>
> Thank you for your thorough review of our paper and rebuttal! We are honored that our rebuttal has addressed most of your concerns and that you have subsequently increased your score. Below, we provide responses to the remaining concerns you may have.
>
> Sincerely Yours,
>
> The Authors
>
>
> > **Rationale for adding additional training results for FastV [1].**
>
> We included additional training results for FastV in the ''Response to W1-1'' for Reviewer LdRV to provide a comparative analysis with the SoTA method FastV, which apply the same sparse training as Dynamic-LLaVA. This comparison aims to demonstrate the superior performance of our proposed method. As shown in the ''Response to W1-1'' for Reviewer LdRV, Dynamic-LLaVA exhibits significant improvement compared to both the untrained and trained versions of FastV.
>
>
>
> > **Settings we used for FastV**
>
> In both the paper and rebuttal, we have kept the settings for FastV consistent. Specifically, we set K=3 and R=0.75, as specified in ''Tab. 1 of FastV [1]''. This corresponds to the factors you mentioned (specific layers pruned, the total number of model layers, and whether multi-head or group-query attention is used), which are as follows:
>
> - **Specific layers pruned**: The first three decoder layers use all tokens for computation, while layers beyond the third use only 25% of the tokens.
> - **Total number of model layers**: LLaVA-1.5-7B has 32 decoder layers, while LLaVA-1.5-13B has 40 layers.
> - **Multi-head or group-query attention**: We use multi-head attention (consistent with the official LLaVA-1.5 implementation).
>
> *For the calculation of FLOPs, the detailed process can be found in Tab. 10 and Appendix A.4.5.*
>
>
>
> > **Fairness of the settings for FastV**
>
> The settings K=3 and R=0.75 that we used are a powerful configuration for FastV (likely the optimal one), as recommended in Tab. 1 of the FastV paper [1].
>
> Additionally, we explored the performance of FastV reported in other papers (LLaVA-1.5-7B).
>
> |                          | TFLOPs（prefill）$\downarrow$ | VQA$^{v2}$ | GQA        | SQA        | VQA$^{T}$  | POPE       | MME          | MMB        | SEED       |
> | ------------------------ | ----------------------------- | ---------- | ---------- | ---------- | ---------- | ---------- | ------------ | ---------- | ---------- |
> | FastV in SparseVLM [2]   | 33%                           | 67.1       | 52.7       | 67.3       | 52.5       | 64.8       | -            | 61.2       | -          |
> | ***FastV in our paper*** | ***32%***                     | ***75.1*** | ***57.5*** | ***68.7*** | ***56.2*** | ***81.0*** | ***1458.9*** | ***63.5*** | ***62.8*** |
>
> We ensure that the settings we used are fair. You can obtain the results we reported by using our Anonymous GitHub code ([here](https://anonymous.4open.science/r/ICLR2025_submission4573_full_codes-0F42/LLaVA_fastv/llava/model/language_model/fastv_kvcache.py#L178)) to set ratio = 0.25 (equivalent to modifying the official FastV GitHub repository ([here](https://github.com/pkunlp-icler/FastV/blob/main/src/FastV/lmms-eval/fastv_kvcache.py#L120))).
>
> [1] An Image is Worth 1/2 Tokens After Layer 2: Plug-and-Play Inference Acceleration for Large Vision-Language Models. ECCV2024.
>
> [2] SparseVLM: Visual Token Sparsification for Efficient Vision-Language Model Inference. Arxiv2410.

---

> ### Author Response · Authors · 2024-11-30
>
> Dear Reviewer LdRV,
>
> Thank you again for your time and effort dedicated to reviewing our paper. Please let us know if our responses have addressed your concerns. If you have remaining concerns, please don't hestitate to let us know and we are happy to address them. If your concerns have been addressed, we respectfully ask you to consider raising the score.
>
> Sincerely Yours,
>
> The Authors

---

> ### Author Response · Authors · 2024-12-02
>
> Dear Reviewer LdRV,
>
> As the discussion deadline is approaching, we would like to kindly follow up to confirm whether our responses have adequately addressed your concerns. **We sincerely hope you will participate in the discussion.**
>
> Sincerely Yours,
>
> The Authors

---

> > ### Comment · Reviewer_LdRV · 2024-12-03
> > **Response to the rebuttal**
> >
> > Dear Authors,
> >
> > Thank you for your rebuttal, it has resolved most of my concern. I have raised my score.
> >
> > Best,
> > Reviewer LdRV

---

> > > ### Author Response · Authors · 2024-12-03
> > >
> > > Dear Reviewer LdRV,
> > >
> > > Thank you again for your invaluable time and effort on our paper. Thank you very much for approving our work!
> > >
> > > Sincerely Yours,
> > >
> > > The Authors

---

### Author Response · Authors · 2024-11-14
**The Preliminary Rebuttal (1/3)**

Dear Reviewers,

Thank you very much for your effort on our paper.

We are preparing a detailed rebuttal to address the valuable feedback provided by the reviewers. Currently, we have observed that some reviewers have already adjusted their scores and comments before our complete rebuttal. We have decided to release a preliminary rebuttal for several key and commonly raised concerns.  *Please note that this is not the final rebuttal.* A point-by-point response to all reviewers' comments, along with a revised version of the paper, will be submitted subsequently.

Sincerely Yours,

The Authors


> **Comparison with token reduction methods.**

We present the performance of current SoTA and recent token reduction methods. The results in the table are based on evaluations conducted with LLaVA-1.5-7B, using the performance of Dynamic-LLaVA$_{I/T}$ as the base. We report the performance differences of other methods relative to ours. The 'Free' column indicates whether a method is training-free (i.e., can be applied directly to MLLMs without training).

|                             | Free           | Token$\downarrow$ | TFLOPs$\downarrow$ | VQA$^{v2}$  | GQA         | SQA          | VQA$^{T}$    | POPE        | MME            | MMB         | SEED        |
| --------------------------- | -------------- | ----------------- | ------------------ | ----------- | ----------- | ------------ | ------------ | ----------- | -------------- | ----------- | ----------- |
| ***Dynamic-LLaVA$_{I/T}$*** | ***&#10005;*** | ***20%***         | ***25%***          | ***77.9***  | ***61.3***  | ***68.6***   | ***56.5***   | ***85.9***  | ***1501.0***   | ***64.1***  | ***65.0***  |
| (ECCV24) IVTP [1]           | ✕              | -                 | 53%                | 77.8 (-0.1) | 60.4 (-0.9) | 67.8 (-0.8)  | 58.2 (+1.7)  | 85.7 (-0.2) | -              | 66.1 (+2.0) | 56.4 (-8.6) |
| PruMerge+ [2]               | ✕              | 25%               | 25%                | 76.8 (-1.1) | -           | 68.3 (-0.3)  | 57.1 (+0.6)  | 84.0 (-1.9) | 1462.4 (-38.6) | 64.9 (+0.8) | -           |
| VoCo-LLaMA [3]              | ✕              | 22%               | 22%                | 76.9 (-1.0) | 59.8 (-1.5) | -            | -            | -           | -              | 61.0 (-3.1) | 59.1 (-5.9) |
| TRIM [4]                    | &#10005;       | 21%               | -                  | 76.4 (-1.5) | 61.4 (+0.1) | 48.1 (-20.5) | 53.7 (-2.8)  | 85.3 (-0.6) | 1461.3 (-39.7) | 67.4 (+3.3) | -           |
| (ECCV24) FastV [5]          | ✓              | 25%               | 32%                | 75.1 (-2.8) | 57.5 (-3.8) | 68.7 (+0.1)  | 56.2 (-0.3)  | 81.0 (-4.9) | 1458.9 (-42.1) | 63.5 (-0.6) | 62.8 (-2.2) |
| HiRED [6]                   | ✓              | 20%               | 20%                | 74.7 (-3.2) | -           | 66.4 (-2.2)  | 44.2 (-12.3) | -           | -              | -           | -           |
| SparseVLM [7]               | ✓              | 22%               | -                  | 73.8 (-4.1) | 56.0 (-5.3) | 67.1 (-1.5)  | 54.9 (-1.6)  | 80.5 (-5.4) | 1696 (+195)    | 60.0 (-4.1) | -           |

Our proposed Dynamic-LLaVA_I/T achieves the best performance in 34 out of 42 valid results across eight commonly used benchmarks. It is worth noting that, similar to our method, some other approaches, such as PruMerge+ [2], also require additional post-training.  For LLaVA-1.5, both Dynamic-LLaVA_I/T and PruMerge+[2] undergo one epoch of post-training. However, Dynamic-LLaVA_I/T outperforms PruMerge+ in 4 out of 6 results when LLaVA-1.5-7B is used as the baseline. Moreover, for LLaVA-1.5-13B, Dynamic-LLaVA_I/T consistently surpasses PruMerge+ across all benchmarks (see Tab.1 in the main paper). Compared to other token reduction methods, Dynamic-LLaVA_I/T not only reduces 80% of image tokens in the prefill stage but also decreases 50% of output text tokens during decoding, consistently outperforming both training-free and training-required methods in most cases.

For the generation ability benchmarks, it is important to note that Dynamic-LLaVA_I/T achieves comparable PPL and METEOR scores to FastV [5], indicating similar levels of fluency and quality in generated text [8, 9, 10]. This suggests that, while some generated tokens are discarded, the impact on MLLM's ability to generate coherent text is minimal. However, this similarity in scores does not imply equivalent overall performance (see Tab. 1 in the main paper, where Dynamic-LLaVA_I/T outperforms FastV in 14 out of 16 results on the LLaVA-1.5 experiments).  Additionally, Dynamic-LLaVA_I/T further reduces 50% decoding computation (without KV cache) and GPU memory overhead (with KV cache) compared to FastV [5].

---

> ### Author Response · Authors · 2024-11-14
> **The Preliminary Rebuttal (2/3)**
>
> > **Distinctions from LLMs' KV Cache Compression Methods.**
>
> The proposed Dynamic-LLaVA framework differs fundamentally from LLMs' KV cache compression methods in the following ways:
>
> 1. Dynamic-LLaVA introduces a tailored sparsification inference scheme specific to various inference modes, i.e., prefill, decoding with KV cache, and decoding without KV cache. Only in the case of decoding with KV cache can it be regarded as a form of "online KV cache compression". For prefill and decoding without KV cache, where KV cache activations are not involved, Dynamic-LLaVA can still significantly accelerate MLLM's computation, a benefit that KV cache compression methods cannot achieve. We have noted the reviewers' focus on the decoding with KV cache scenario and the classification of our method as a KV cache compression approach. In the revised paper, we will include a more detailed comparison with LLM KV cache compression methods and discuss the computational benefits in the prefill and decoding without KV cache scenarios. This includes the importance of prefill-stage inference acceleration for MLLMs [5] and potential applications for decoding without KV cache, such as the widely used speculative decoding strategy [11, 12].
> 2.   Considering the complete generation process of MLLMs with KV cache, Dynamic-LLaVA achieves computation and GPU memory savings in the prefill stage, as well as GPU memory savings during decoding (disregarding computational acceleration from KV cache reduction for extremely long sequences [13]). In contrast, typical KV cache compression methods generally offer only GPU memory savings during decoding. This distinction arises because Dynamic-LLaVA evaluates each current token's features to determine whether its activations through $W_K$ and $W_V$ should be added to the KV cache, rather than removing entries from the past cache (detailed in Eq. 6 of the main paper). KV cache compression methods, however, often rely on past generated KV cache. E.g., H2O [13] uses the current Q with the past KV cache in decoding to compute attention scores and decide which cache activations to discard, rather than directly deciding to retain or discard activations for the current token. We will expand upon this discussion in the upcoming revision of the paper.
> 3. Dynamic-LLaVA is an MLLM inference acceleration framework that considers the distinct properties of different modalities and incorporates tailored sparsification strategies accordingly. As noted in [5], “While these methods have boosted the inference efficiency of LLMs, they are designed for text-only language models, and whether their effectiveness can be transferred to LVLMs remain under-explored.”. Thus, the direct applicability of LLM KV cache compression methods to MLLMs is yet to be verified. In our detailed rebuttal, we will provide results on the application of LLM KV cache compression methods in MLLMs.
>
> > **Predictor mechanism.**
>
> The predictor used in Dynamic-LLaVA is applied only once, after the l-th decoder layer of the MLLM. Specifically, once the predictor makes a decision on tokens, this decision is shared across all subsequent layers. For instance, in both the prefill and decoding without KV cache stages, after the predictor reduces the token set $S_l$ to $S_l^*$, all subsequent layers use this reduced token set $S_l^*$ for computation. In decoding with KV cache, the decision of whether to add the current token to the KV cache is likewise propagated to subsequent layers. Following [5], we set l=2, meaning that token reduction occurs after the 2nd layer. Ablation studies on the position of l are provided in Tab. 6 of the main paper.
>
> As for the additional inference overhead introduced by the predictor, we first present theoretical FLOP values. For LLaVA-1.5-13B, the predictor's FLOPs in the prefill stage are under 0.01T, compared to the MLLM’s 19.6T FLOPs. In Table 4 of the main paper, we also report latency measurements on a real GPU for prefill and decoding without KV cache stages. During the decoding with KV cache phase, the predictor's computational overhead is similarly minimal and can be considered negligible. We will provide further details in our comprehensive rebuttal and the revised paper.

---

> ### Author Response · Authors · 2024-11-14
> **The Preliminary Rebuttal (3/3)**
>
> > **Baselines and benchmarks.**
>
> We primarily use LLaVA-1.5 [14] as our main baseline, employing the same 12 benchmarks utilized in LLaVA-1.5. This choice is motivated by the fact that most works in the field [1-7] validate their results on these benchmarks for LLaVA-1.5, allowing for a fair comparison by adopting the same settings. To evaluate the generation ability of MLLMs, we use the LVIS-VQA single- and multi-round benchmarks. Our goal is to assess whether Dynamic-LLaVA, while reducing output text tokens, significantly impacts the generation ability of MLLMs. The results indicate that our method enhances inference efficiency with minimal impact on generation performance. In the rebuttal, we will supplement our results with additional baselines and benchmarks.
>
> > **Writing.**
>
> We recognize that excessive notation may hinder broader reader comprehension. In the revised paper, we are incorporating more intuitive visualizations and adding simplified descriptions to improve the readability.
>
> [1] IVTP: Instruction-guided Visual Token Pruning for Large Vision-Language Models. ECCV2024.
>
> [2] Llava-prumerge: Adaptive token reduction for efficient large multimodal models. Arxiv2403.
>
> [3] VoCo-LLaMA: Towards Vision Compression with Large Language Models. Arxiv2406.
>
> [4] Less is More: A Simple yet Effective Token Reduction Method for Efficient Multi-modal LLMs. Arxiv2409.
>
> [5] An Image is Worth 1/2 Tokens After Layer 2: Plug-and-Play Inference Acceleration for Large Vision-Language Models. ECCV2024.
>
> [6] HiRED: Attention-Guided Token Dropping for Efficient Inference of High-Resolution Vision-Language Models in Resource-Constrained Environments. Arxiv2408.
>
> [7] SparseVLM: Visual Token Sparsification for Efficient Vision-Language Model Inference. Arxiv2410.
>
> [8] Language models are unsupervised multitask learners. OpenAI blog.
>
> [9] Perplexity—a measure of the difficulty of speech recognition tasks. The Journal of the Acoustical Society of America.
>
> [10] Meteor: An automatic metric for mt evaluation with improved correlation with human judgments. ACL Workshop 2005.
>
> [11] Fast inference from transformers via speculative decoding. ICML2023.
>
> [12] Online speculative decoding. Arxiv2310.
>
> [13] H2O: Heavy-Hitter Oracle for Efficient Generative Inference of Large Language Models. NeurIPS2024.
>
> [14] Improved Baselines with Visual Instruction Tuning. CVPR2024.

---

### Author Response · Authors · 2024-11-22
**[2024/11/22 AOE] Summary**

Dear Reviewers,

We thank all reviewers for your insightful and constructive comments! We have submitted the point-by-point responses to all reviewers' comments, along with a revised version of the paper **(with blue text)**. If there is any other question, please feel free to let us know and we will try our best to provide satisfactory answers! ***If your concerns have been addressed, we respectfully ask the reviewer to consider raising the score.***

Please allow us to highlight the contributions of our work. **Dynamic-LLaVA is the first MLLM acceleration framework that simultaneously sparsifies both vision and language contexts while integrating inference efficiency optimization across different MLLM inference modes into a unified framework.** This design enables Dynamic-LLaVA to achieve additional inference efficiency throughout the entire generation process, surpassing both MLLM token reduction methods (additional inference efficiency in decoding with/without KV cache) and LLM KV cache compression methods (additional inference efficiency in prefill and decoding without KV cache). Moreover, Dynamic-LLaVA demonstrates significant performance advantages. Therefore, **we believe that our work provides a clear and substantial contribution to the research community.**

Menwhile, we provide a summary of the specific revisions to the paper below.

Sincerely Yours,

The Authors

> **Additional results.**

- [Reviewer LdRV] Additional sparsification training for FastV (Tab. 1 and Appendix A.4.2).
- [Reviewer LdRV] Evaluate LLM KV cache compression methods in MLLM vision understanding and generation ability benchmasks (Tab. 1, Tab. 3 and Appendix A.4.2).
- [Reviewer LdRV] Evaluate the effect on the MaskedSoftmax operation (Tab. 7 and Appendix A.2).
- [Reviewers UyMq, FzrR] Evaluation of the text context sparsification in prefill (Tab. 9 and Appendix A.3.3).
- [Reviewer FzrR] Evaluation of vision-centric vision understanding benchmarks (Tab. 1 and Tab. 5).
- [Reviewer FzrR] Evaluation of complex generation ability benchmarks (Tab. 3 and Appendix A.4.4).

> **Additional details and discussions.**

- [Reviewer LdRV] Comparison with more SoTA vision context sparsification methods (Tab. 11 and Appendix A.5.1).
- [Reviewers LdRV, UyMq] Inferency efficiency discussion of Dynamic-LLaVA and MLLM token reduction methods (Appendix A3.1).
- [Reviewers LdRV, UyMq] Detailed discussion of Dynamic-LLaVA and LLM KV cache compression methods (Fig. 4, Sec. 2.2 and Appendix A.3.2).
- [Reviewer UyMq] Benchmark statistics and inference benefit discussion (Tab. 8 and Appendix A.3.1).
- [Reviewer UyMq] Details of additional training time (Tab. 15 and Appendix A.5.3).
- [Reviewer UyMq] Details of the lantency when decoding with KV cache (Tab. 14 and Appendix A.5.2).

>  **Writing.**

- Add more explanations for the predictor mechanism (Sec. 3.3.1, Sec.3.3.2 and Fig. 2).
- Add more explanations for the end-to-end sparsification training (Sec. 3.3.3, Sec. A.2 and Fig. 3).
- Add more intuitive figures (Fig. 3 and Fig. 4).
- Move the ''Parallel Sparsification Inference Optimization'' section focused on engineering optimizations to the Appendix (Appendix A.1).

> **Code.**

The implementationsof Dynamic-LLaVA and other methods (e.g., FastV, H2O)  are available at the Anonymous GitHub (https://anonymous.4open.science/r/ICLR2025_submission4573_full_codes-0F42/README.md). We provide the complete codes to facilitate a thorough review of our implementations. Should you require additional details, such as weights or logs, please feel free to contact us. We are committed to ensuring the authenticity and reliability of our results.

---

### Meta-Review · Area_Chair_95wW · 2024-12-19

**Metareview:**

This paper introduces Dynamic-LLaVA, a method for efficient multimodal large language models through dynamic vision-language context sparsification. The framework reduces computation significantly during both the prefill stage and decoding stage, while maintaining performance. Reviewers raised concerns about the necessity of the trainable approach, post-training overhead, and alignment between training and inference. The authors addressed these concerns, leading to scores of 6, 6, 6, 6.

Given the promising results in computational efficiency and memory savings, the AC acknowledges the significant contribution of this work to the field of multimodal large language models, especially in terms of optimizing vision-language context sparsification. With an average score of 6, the paper is deemed competitive for this year's ICLR conference. The AC agrees to accept the paper, with the recommendation to integrate the rebuttal content into the final version.

**Additional Comments On Reviewer Discussion:**

Reviewers raised concerns about the necessity of the trainable approach, post-training overhead, and alignment between training and inference. The authors addressed these concerns, leading to scores of 6, 6, 6, 6.

---

### Decision · Program_Chairs · 2025-01-22

Accept (Poster)